# Reassessment of growth-climate relations indicates the potential for decline across Eurasian boreal larch forests

Wenqing Li [1,2], Rubén D. Manzanedo[3,4], Yuan Jiang [1,5] ✉, Wenqiu Ma[6], Enzai Du [5], Shoudong Zhao[7], Tim Rademacher [3,8], Manyu Dong[1,5], Hui Xu[9], Xinyu Kang[10], Jun Wang[2], Fang Wu[1,11], Xuefeng Cui[11] & Neil Pederson [3]

Larch, a widely distributed tree in boreal Eurasia, is experiencing rapid warming across much of its distribution. A comprehensive assessment of growth on warming is needed to comprehend the potential impact of climate change. Most studies, relying on rigid calendar-based temperature series, have detected monotonic responses at the margins of boreal Eurasia, but not across the region. Here, we developed a method for constructing temporally flexible and physiologically relevant temperature series to reassess growth-temperature relations of larch across boreal Eurasia. Our method appears more effective in assessing the impact of warming on growth than previous methods. Our approach indicates widespread and spatially heterogeneous growth-temperature responses that are driven by local climate. Models quantifying these results project that the negative responses of growth to temperature will spread northward and upward throughout this century. If true, the risks of warming to boreal Eurasia could be more widespread than conveyed from previous works.

Rapid warming in recent decades has profoundly affected and restructured global terrestrial biomes[1–3]. These impacts are especially notable in boreal forests where warming is occurring faster than the global average[4,5]. The circumboreal belt of forests encompasses ~30% of the global forested area[5], accounts for ~20% of the global forest carbon sink[6], and provides critical ecosystem services globally[7]. Previous studies have revealed the responses of boreal forests to warming-induced environmental changes, including changes in forest productivity[8,9], treeline shifts[10,11], redistribution of climatic limitations[12–14], and substantial ecosystem transformations[3,15]. However, our current understanding of the effects of climate changes on boreal forests mainly derives from research focusing on boreal evergreen conifers in Europe and North America (e.g., refs. [16, 17]) or treating boreal forests as a somewhat homogeneous whole biome (e.g., refs. [5, 18]). As a result, the dynamics of some boreal ecosystems, notably, the extensive larch forests in boreal Eurasia are relatively poorly understood.

The dominant larch species in boreal Eurasia, Siberian larch (*Larix sibirica*) and Dahurian larch (*Larix gmelinii*), are distributed over

[1]Beijing Key Laboratory of Traditional Chinese Medicine Protection and Utilization, Faculty of Geographical Science, Beijing Normal University, Beijing 100875, Zhuhai 519087, China. [2]Key Laboratory of Land Consolidation and Rehabilitation, Land Consolidation and Rehabilitation Center, Ministry of Natural Resources, Beijing 100035, China. [3]Harvard Forest, Harvard University, Petersham, MA 01366, USA. [4]Plant Ecology, Institute of Integrative Biology, D-USYS, ETH Zürich, 8006 Zürich, Switzerland. [5]State Key Laboratory of Earth Surface Processes and Resource Ecology, Faculty of Geographical Science, Beijing Normal University, Beijing 100875, China. [6]College of Engineering, China Agricultural University, Beijing 100083, China. [7]State Key Laboratory of Severe Weather, Chinese Academy of Meteorological Sciences, Beijing 100081, China. [8]Institut des Sciences de la Forêt Tempérée, Université du Québec en Outaouais, Ripon J0V 1V0 QC, Canada. [9]Department of Biostatistics and Epidemiology, School of Public Health and Health Sciences, University of Massachusetts Amherst, Amherst, MA 01003, USA. [10]Department of Mathematics and Statistics, Boston University, 111 Cummington Mall, Boston, MA 02215, USA. [11]School of Systems Science, Beijing Normal University, Beijing 100875, China. ✉e-mail: jiangy@bnu.edu.cn

>10 million km² (Fig. 1) and constitute one of the largest global reservoirs of biogenic carbon[19]. The wide distributions would reasonably imply high spatial heterogeneity of growth-climate response patterns[12,20-22]. Prior research has found that low temperature during the growing season is the main limitation to boreal forest productivity[12,23]. Indeed, tree growth in boreal Eurasia has increased under climate warming[24-26]. However, accompanied by modest changes in precipitation, rapid warming has resulted in exacerbated moisture deficits across boreal Eurasia (Supplementary Fig. 1), which may limit the positive effects of warming on trees. Recent studies have reported weakening low-temperature control and projected the tipping point of warming effects from positive to negative on some boreal ecosystems[14,26,27]. These changes appear to have led to growth declines over the drier southern margins of Eurasian boreal forests[28]. Spatially-contrasting responses of forest growth to warming have also been detected at mid-high latitudes over the Northern Hemisphere[12,18,29]. Importantly, IPCC forecasts remarkable increases in temperature across boreal Eurasia throughout the 21st century[30] (Supplementary Part A). The importance of larch species to the ecology and carbon dynamics of boreal Eurasia calls for an improvement in our understanding on their response patterns to climate change.

Large-scale networks of tree-growth data are invaluable for understanding how forest biomes respond to environmental changes[31,32]. The International Tree-Ring Data Bank (ITRDB), with its vast spatial-temporal coverage, offers a unique opportunity for exploring continental- or even global-scale ecological issues[31,33]. Research using ITRDB data has previously suggested that summer temperatures do not influence diameter growth of trees in ~50% of populations located in circumboreal regions (50-67°N)[18]. Another expanded ITRDB-based tree-ring network across boreal Eurasia (>60°N) even found 71.0% of the populations insensitive to temperature, and the sensitive populations are predominantly distributed in

the margins of Eurasian boreal forests[20]. By contrast, studies using satellite-derived tree canopy growth data suggests that warming significantly impacts a large portion of Eurasian boreal forests[26,34]. Clearly, there is an important gap in knowledge between these two approaches over a biome that is important to the global carbon cycle, which could be attributed to the different requirements of wood formation and leaf activity for temperature[35,36].

A potential source for the muted growth-temperature responses in tree-ring-based studies could be linked to the calendar-based approach for calculating growth-temperature correlations. Many studies focus on the variation in tree growth related to climatic variations. However, identifying the climate factors and time periods over which climate affects tree growth has been problematic[37]. General methods for detecting growth-temperature responses typically correlate annual tree growth with temperature series for rigid calendar periods (months, seasons, and/or annual), meaning that choice of time periods is fixed, arbitrary, and might not follow what the physiology of trees require for growth. Since xylem phenology is mainly regulated by temperature cues[38,39], the same calendar period can correspond to very different growth phases from year to year due to the great inter-annual variations in seasonal dynamics of temperature in boreal regions[40-42], therefore, correlating annual growth with temperatures for fixed periods may result in underestimations of growth-temperature response. For trees living under continuous climatic stress or in regions with consistent growing seasons, monthly or seasonal temperature series can be relatively effective to investigate growth-temperature relations. However, for trees living in regions with comparably variable climates, it is not likely appropriate to conclude that trees are insensitive to temperature when climate series are based solely on a human calendar. Boreal growing seasons, especially near the Arctic Circle, are often shorter than two months and their starting and ending dates vary greatly between years[43]. In addition, larches

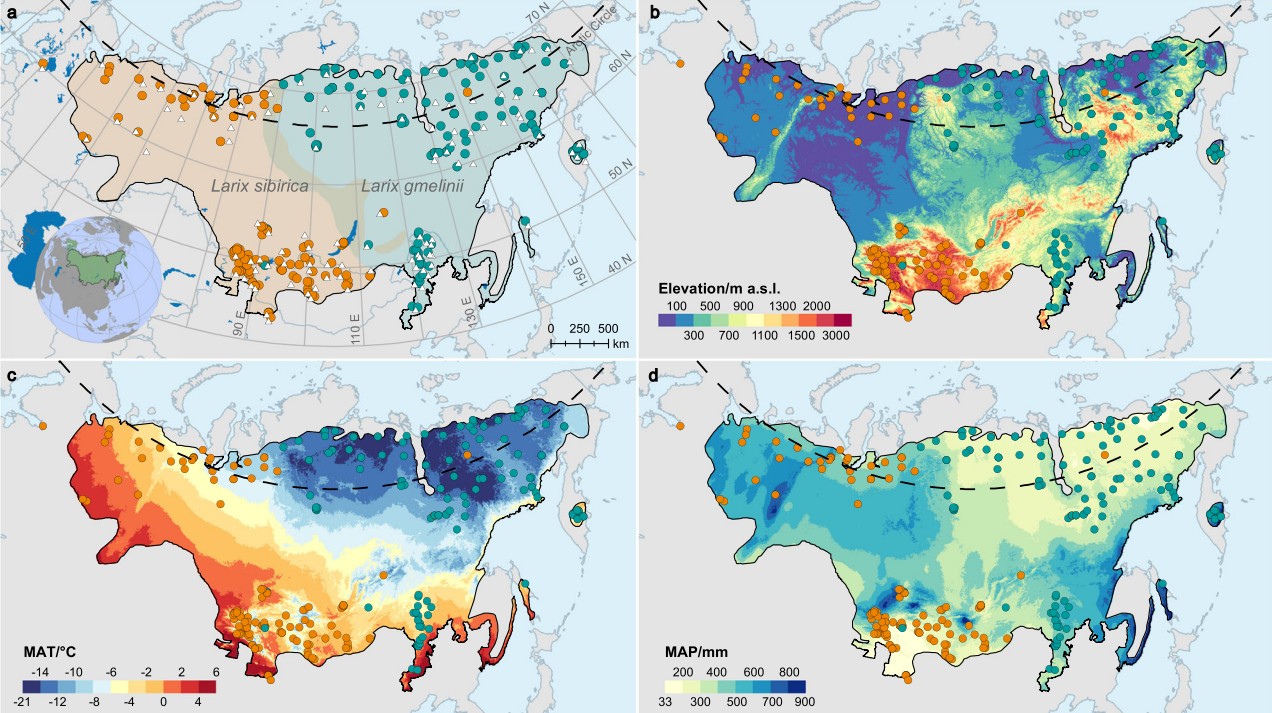

**Fig. 1 | Distribution of Eurasian boreal larch forests and the sampling populations. a** Species distributions of Siberian larch (*Larix sibirica*, orange shadow) and Dahurian larch (*Larix gmelinii*, cyan shadow) digitized and merged from multiple repositories[41,70,71]. Circles represent the tree-ring sampling populations, following the same color scheme as the species distributions; squares represent the

meteorological stations. **b–d** elevation (SRTM DEM v4.1), mean annual temperature (MAT), and precipitation (MAP) during 1970–2000 (WorldClim v2.1) across the distribution of Eurasian boreal larch forests. This figure was created using ArcGIS 10.2 for Desktop (ESRI, Inc).

dominant in boreal Eurasia are deciduous conifers that need to renew their needles at the beginning of each growing season. This trait limits xylem formation in both rate and duration[44], further compressing growth response windows and confusing the detection of response signals using calendar-based approach. Given the lack of consistency between tree-ring-based and satellite-based studies, the potential limitations of calendar-based methods, and the physiological unique-ness of larch, it seems necessary to develop new methods to construct more flexible and physiologically-relevant temperature series. Doing this could both improve the detection of growth responses across these forests and potentially clarify the seemingly inconsistent results shown by different methodological approaches.

Underestimating the responses of boreal forests to projected climate changes may lead to blind optimism and wasted management opportunities. To address current methodological limitations and potential underestimation, we developed the temperature-linked (T-linked) method to construct temperature time series based on movable periods linked to intra-annual temperature variability (see "Methods" and Supplementary Part D for details on how it runs) and used those series to calculate growth-temperature response metrics comparable across the distribution areas of Siberian larch and Dahurian larch at both the individual and population levels. We test whether our results are consistent with those by calendar-based methods. And then, our aims are to (i) explore the spatial coherence in growth-temperature response across boreal larch distributions; (ii) discuss the climatic drivers of the potential spatial heterogeneity of growth-temperature response; and (iii) identify the areas where warming has negative impacts on boreal larch and estimate the future dynamics. To achieve these goals, we compiled an extensive tree-ring network composed of 8544 annual radial growth series of 5089 larch trees and 260 larch populations covering the distributions of boreal larch (Fig. 1). Here, we detect widespread and spatially heterogeneous growth responses to temperature that are driven by local climate. Based on the models quantifying these results, we further project that the negative growth-temperature responses will spread northward and upward throughout this century. Our method refines the detection of growth-temperature responses in boreal Eurasia, and contributes to a better understanding of the potential warming-induced risks these ecosystems may face in the future.

## Results
### Effectiveness of the T-linked method
We compared the T-linked method with two methods that develop temperature series based on fixed calendar periods under the statistical constraint of calculating correlations of the same order of magnitude. Using a moving calendar-based method, each growth series was correlated with 2106 calendar-based temperature series, including commonly-used seasonal and monthly temperature series. Through this way, ~70% of the sampling individuals and populations were identified as sensitive to temperature. Specifically, 29.3% ($n = 1803$) and 32.9% ($n = 1375$) of the single tree growth series showed significant negative responses to calendar-based temperature series from 1960–1990 and 1970–2000, respectively, while 61.1% ($n = 3757$) and 46.0% ($n = 1925$) responded positively (Table 1). Growth-temperature response signals detected using the calendar-based T-linked method (see Methods) were close to those detected using the general moving calendar-based method for both species and both analysis periods (Table 1).

By contrast, we found that >95% of the individual trees and populations were sensitive to temperature when using our more flexible T-linked temperature series approach. We found that the proportions of single tree growth series showing negative temperature sensitivity were 57.3% ($n = 3521$) in 1960–1990 and 68.5% ($n = 2866$) in 1970–2000, while the proportions of positive temperature-sensitive individuals were 86.7% ($n = 5324$) and 81.2% ($n = 3397$) (Table 1).

Analysis at the population level showed the same set of trends. The percentage of populations that responded negatively was 46.9% ($n = 113$) in 1960–1990 and 63.9% ($n = 101$) in 1970–2000, while those responding positively were 82.2% ($n = 198$) and 72.2% ($n = 114$) (Table 1). These percentages were also consistent between species (Table 1). The overwhelming proportion of significant growth-temperature responses detected at both the tree and population levels using our T-linked temperature series demonstrates the effectiveness of T-linked method and highlights the key role of temperature on larch growth across boreal Eurasia.

### Heterogeneity of growth-temperature response pattern
We found significant relationships between the temperature sensitivity of boreal larch and local geographic and climatic conditions (Supplementary Figs. 4–8). After we used partial correlation to control the covarying effects of precipitation and temperature, the four correlation-derived metrics characterizing negative temperature sensitivity were all positively correlated with local temperature and negatively correlated with local precipitation, respectively ($p < 0.01$; Supplementary Fig. 4). Warmer local temperatures and reduced local precipitation resulted in a greater negative impact of temperature on tree growth, manifesting as both wider affecting scopes and longer affecting durations. By contrast, positive temperature sensitivity showed the opposite, but weaker relationships with local climate. For Siberian larch populations, the maximum proportion of positively-responding individuals (MST) and the duration of positive population-level growth response (pDur) were significantly correlated with local climate ($p < 0.01$; Supplementary Fig. 4). For Dahurian larch populations, all four positive temperature sensitivity metrics were negatively correlated with local temperature, while only the durations of positive responses (tDur05, tDur50, and pDur) were positively correlated with local precipitation ($p < 0.01$, Supplementary Fig. 4). The mean coefficient of significant population-level growth-temperature correlations was negatively correlated with local temperature (partial $r_{Siberian\ larch} = -0.57$, $n = 211$ populations; partial $r_{Dahurian\ larch} = -0.50$, $n = 188$ populations; $p < 0.01$) and positively correlated with local precipitation (partial $r = 0.41$ and $0.30$; $p < 0.01$) (Supplementary Fig. 6). Populations with opposing temperature sensitivities were significantly clustered along local climatic conditions (two-sided $t$-test, $p < 0.01$). Interestingly, the latitudinal cluster prevailed over the longitudinal cluster (Supplementary Fig. 7), which reinforces the idea of local temperature in determining the growth-temperature response pattern.

Overall, Eurasian boreal larch populations with high proportion of negative temperature-sensitive trees, significant negative population-level growth-temperature responses, and long durations of growth responses to temperature tended to be located in warm and dry regions, predominantly in the southern parts of the species distributions.

### Local climate drives the heterogeneity of growth response
Tree populations with a negative response to temperature occupied roughly the opposite climate space as those showing positive sensitivity (Supplementary Fig. 8). We divided populations with significant temperature relations into two groups: (1) a positively-responding group composed of populations showing significant positive responses to temperature but no negative responses, and (2) a negatively-responding group composed of populations showing significant negative responses to temperature regardless of showing or not showing positive responses (see "Methods"). These two groups were significantly clustered in the climate space along local mean annual temperature and mean annual precipitation (Fig. 2). From this, we quantified the relationship between the binary grouping and the local climates using logistic regression models.

Our results indicated that the probability of a boreal larch population showing negative growth responses to temperature depended

**Table 1 | Summary of growth series showing temperature sensitivity detected using different temperature series construction methods**

| Species | Time period | Total number | Method[a] | Sensitive pct. (%) | Negative sensitive pct. (%) | Positive sensitive pct. (%) |
|---|---|---|---|---|---|---|
| **Single growth series—individual tree-level growth** | | | | | | |
| *Larix sibirica* | 1960–1990 | 2636 | TL | 2548 (96.7%) | 1448 (54.9%) | 2312 (87.7%) |
| | | | CB | 2047 (77.7%) | 714 (27.1%) | 1675 (63.5%) |
| | | | CT | 2028 (76.9%) | 610 (23.1%) | 1641 (62.3%) |
| | 1970–2000 | 2053 | TL | 1982 (96.5%) | 1432 (69.8%) | 1568 (76.4%) |
| | | | CB | 1424 (69.4%) | 809 (39.4%) | 835 (40.7%) |
| | | | CT | 1427 (69.5%) | 806 (39.3%) | 759 (37.0%) |
| *Larix gmelinii* | 1960–1990 | 3508 | TL | 3361 (95.8%) | 2073 (59.1%) | 3012 (85.9%) |
| | | | CB | 2547 (72.6%) | 1089 (31.0%) | 2082 (59.4%) |
| | | | CT | 2666 (76.0%) | 864 (24.6%) | 2189 (62.4%) |
| | 1970–2000 | 2130 | TL | 2084 (97.8%) | 1434 (67.3%) | 1829 (85.9%) |
| | | | CB | 1393 (65.4%) | 566 (26.6%) | 1090 (51.2%) |
| | | | CT | 1575 (73.9%) | 699 (32.8%) | 1107 (52.0%) |
| **Population chronology—population-level growth** | | | | | | |
| *Larix sibirica* | 1960–1990 | 117 | TL | 113 (96.6%) | 57 (48.7%) | 95 (81.2%) |
| | | | CB | 97 (82.9%) | 35 (29.9%) | 73 (62.4%) |
| | | | CT | 97 (82.9%) | 31 (26.5%) | 71 (60.7%) |
| | 1970–2000 | 94 | TL | 89 (94.6%) | 60 (63.8%) | 64 (68.1%) |
| | | | CB | 64 (68.1%) | 34 (36.2%) | 36 (38.3%) |
| | | | CT | 69 (73.4%) | 38 (40.4%) | 34 (36.2%) |
| *Larix gmelinii* | 1960–1990 | 124 | TL | 116 (93.5%) | 56 (45.2%) | 103 (83.1%) |
| | | | CB | 89 (71.8%) | 41 (33.1%) | 82 (66.1%) |
| | | | CT | 105 (84.7%) | 30 (24.2%) | 94 (75.8%) |
| | 1970–2000 | 64 | TL | 63 (98.4%) | 41 (64.1%) | 50 (78.1%) |
| | | | CB | 42 (65.6%) | 13 (20.3%) | 35 (54.7%) |
| | | | CT | 45 (70.3%) | 18 (28.1%) | 31 (48.4%) |

[a]TL: T-linked method; CB: moving calendar-based method; CT: calendar-based T-linked method.

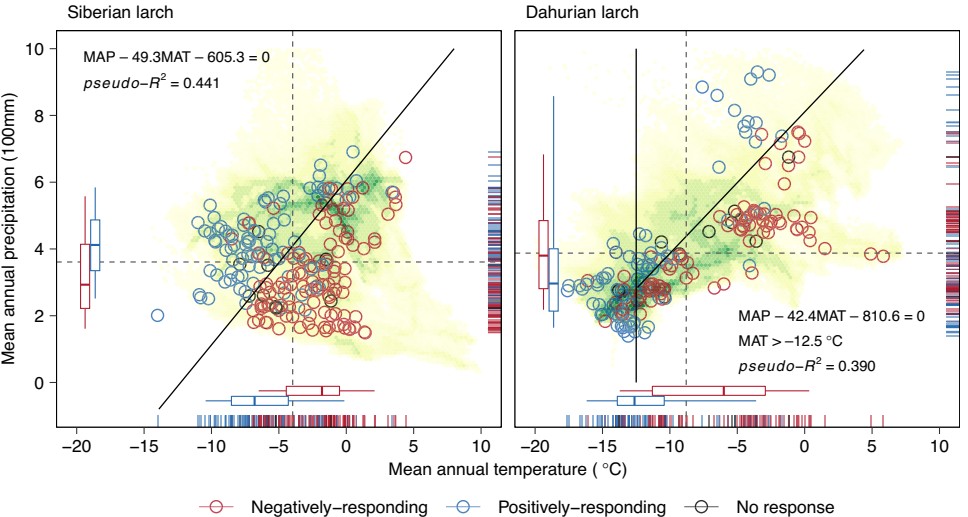

**Fig. 2 | Climate boundaries between the positively-responding and negatively-responding populations with the identification probability threshold of 0.50.** Shadows represent the climate spaces occupied by the two species from 1970 to 2000, while the green-yellow gradient represents high-low cell density. Red circles represent the observed negatively-responding populations, those showing significant negative responses to temperature, regardless of showing or not showing positive responses; blue circles represent the observed positively-responding populations, those showing significant positive responses to temperature but no negative responses. Dashed lines denote the average values of climatic conditions; boxes represent the 25th, 50th and 75th quantiles, and whiskers extend to the 5th and 95th quantiles. Solid lines represent the climate boundaries using the identification probability threshold of 0.50, analysis formulas of the boundaries are noted on each panel (*pseudo-R*$^2$ = 0.441 and 0.390, *n* = 202 and 123 population chronologies, and cross-validation accuracy = 86.16% and 88.15% for Siberian larch and Dahurian larch, respectively). Upper-left and lower-right sides of the boundary represent the estimated positively-responding and negatively-responding climate spaces, respectively.

**Table 2 | Parameter estimates of the logistic regression models**

| Species | MAT ($w_1$, °C) | MAP ($w_2$, mm) | Intercept ($w_3$) |
|---|---|---|---|
| Siberian larch | −0.5974 | 0.01212 | −7.3368 |
| | $P = 1/(1 + e^{(-0.5974*MAT+0.01212*MAP-7.3368)})$ | | |
| Dahurian larch | −0.7363 | 0.01737 | −14.0796 |
| | $P = 1/(1 + e^{(-0.7363*MAT+0.01737*MAP-14.0796)})$ | | |

MAT and MAP represent mean annual temperature and precipitation, respectively; all parameter estimates reached the 0.001 significance level using Wald test. See Supplementary Table 3 for the statistics of the models.

on the balance between local mean annual temperature and precipitation (Table 2). The slopes of the climate boundaries between the positively-responding populations and the negatively-responding populations were close between species, 49.3 mm °C$^{-1}$ for Siberian larch (McFadden's *pseudo*-R$^2$ = 0.4415, $p < 0.001$, $n = 202$) and 42.4 mm °C$^{-1}$ for Dahurian larch (McFadden's *pseudo*-R$^2$ = 0.3904, $p < 0.001$, $n = 123$) (Table 2, Fig. 2; Supplementary Table 2). We did, however, find that Dahurian larch populations with local mean annual temperature below −12.5 °C were typically not negatively impacted by temperature (Fig. 2). The climate boundaries of Siberian larch had lower intercepts than those of Dahurian larch (Fig.2; Supplementary Table 2), indicating a relatively higher level of drought tolerance in Siberian larch. Our 10-fold cross-validation analysis indicated that the estimated models were robust and of fairly-high accuracy and quality (Siberian larch: observed accuracy = 0.862 ± 0.064, Cohen's $\kappa$ = 0.713 ± 0.132; Dahurian larch: observed accuracy = 0.885 ± 0.056, Cohen's $\kappa$ = 0.706 ± 0.131).

### Baseline and projected negatively-responding regions

The baseline probability of the larch populations showing negative growth-temperature responses decreased with latitude and elevation (Fig. 3a–c). Different identification probability thresholds resulted in slightly different, but qualitatively consistent spatial distributions of the positively- and negatively-responding populations (Fig. 3a–c). Estimations using the loose probability threshold of 0.50 suggested that >50% of the distributions of Siberian larch and Dahurian larch could be increasingly stressed at high temperature during portions of the growing season from 1970–2000 (Fig. 3d, e). Even using conservative estimations at the strict probability threshold of 0.95 identified 12.7–19.1% of the species distributions as negatively-responding. Elevation was also an important factor to consider. As expected, populations showing negative growth-temperature responses tended to be located at lower elevations. The effects of elevation seemed stronger in the southern portion of species distributions (Fig. 3f, g). For example, for Siberian larch, the 0.50 iso-probability line of showing negative responses rose from <500 m a.s.l. on average in the north to >2000 m a.s.l. in the south.

According to our projections, the negatively-responding regions were likely to expand northward and upward as anthropogenic climate change continues to evolve (Fig. 4; Supplementary Fig. 9). In contrast, positively-responding regions were likely to recede. These patterns were highly consistent among the climate projections (Supplementary Fig. 10). Even the moderate estimations under SSP2-45 predicted noticeable expansion of the negatively-responding regions throughout this century (Fig. 4). Pessimistic estimations under the worst-case SSP5-85 suggested that >75% of the species distributions would be negatively affected by high temperature during portions of the growing season by 2100 (Fig. 4). Only under the optimistic SSP1-26 would the expanding of negatively-responding region be reduced after 2040 (Supplementary Fig. 9). Even so, there is a good chance it would have already substantially expanded compared with baseline 1970–2000. It should be noted that the spatial trend of increasing

negatively-responding probability is consistent with the spatial trend of decreasing tree density (Supplementary Fig. 11).

## Discussion

Despite the wide distribution and great ecological relevance of Eurasian boreal forests[19], they have received much less scientific attention than boreal forests in western Europe and North America. Previous work correlating tree growth with temperature series based on rigid calendar periods detected a universal response that was almost exclusively set in the margins of the Eurasian boreal forest with continuous severe heat or moisture deficits[18,20,23]. By contrast, our more temporally-flexible and physiologically-relevant T-linked method revealed a much more widespread and universal growth response to temperature across the Eurasian boreal larch forest. We suspect that the deciduous trait of larch, together with the short and highly variable boreal growing seasons, are likely to have diluted the footprints of climate variations in radial growth. This may explain why calendar-based methods work well for evergreen conifers, such as spruce, fir, and pine[22,45], but less so for larch. It might also be why the rigid calendar approach is relatively effective for margins of Eurasian boreal forests[28], but not so for the entire distribution[20]. We expect our method could inform studies focusing on physiological mechanisms of when and how temperature affects growth and identifying critical temperature thresholds for tree growth. While there are potentially serious issues with conducting such a large number of calculations, we tested the risk of potential spurious correlations using the T-linked method through a fair comparison to calendar-based methods and suggested that the effectiveness of our method was not attributed to the increased number of calculations. Furthermore, the cross-validation analyses we conducted demonstrated fairly-high accuracy and robustness of our estimated models, suggesting limited influences of spurious correlations on the main results.

Our results revealed clear spatial heterogeneity in growth-temperature response pattern across Eurasian boreal larch forests. Populations with opposing growth-temperature responses were clustered both spatially and climatically. We further confirmed the key role of local climate played in determining how a larch population responds to warming, thereby identifying local climate as the main driving force for the spatial distribution of heterogenous growth-temperature responses. Higher local precipitation likely allows for higher increases in local temperature before trees showed negative responses to temperature. At the continental scale, broadly-varying climatic conditions were able to accurately discriminate different growth-temperature response patterns[12,46]. Models quantifying the relationship between response pattern and local climates demonstrate that populations with climatic conditions beyond the estimated climate boundary are likely to be only somewhat negatively affected by higher temperatures during portions of the growing season, even though the net effects of warming on these populations are positive.

Rapid warming has led to radical changes in abiotic conditions across global boreal regions[5], affecting forest growth through multiple pathways, including rapid thawing of the permafrost, changed nutrient availability, and drastically altered hydrothermal conditions[41,47,48]. Increasing soil temperature is expected to promote nitrogen mineralization through stimulating the humus decomposition[47], consequently alleviating the prevailing nitrogen limitation in boreal forests[49]. Warming also extends the growing season[29] and ensures the snowpack meltwater for moisture demand during the growing season[40]. Taken together, increasing temperature is expected to enhance boreal forest productivity[25,26] and advance boreal forest into Arctic tundra[11,50]. However, previous studies indicate that such beneficial effects may be transient[13,47]. Our findings also challenge the idea of a monolithic positive response pattern to warming across boreal forests[24,25] and warn of the risks behind the warming-induced benefits over Eurasian boreal forests.

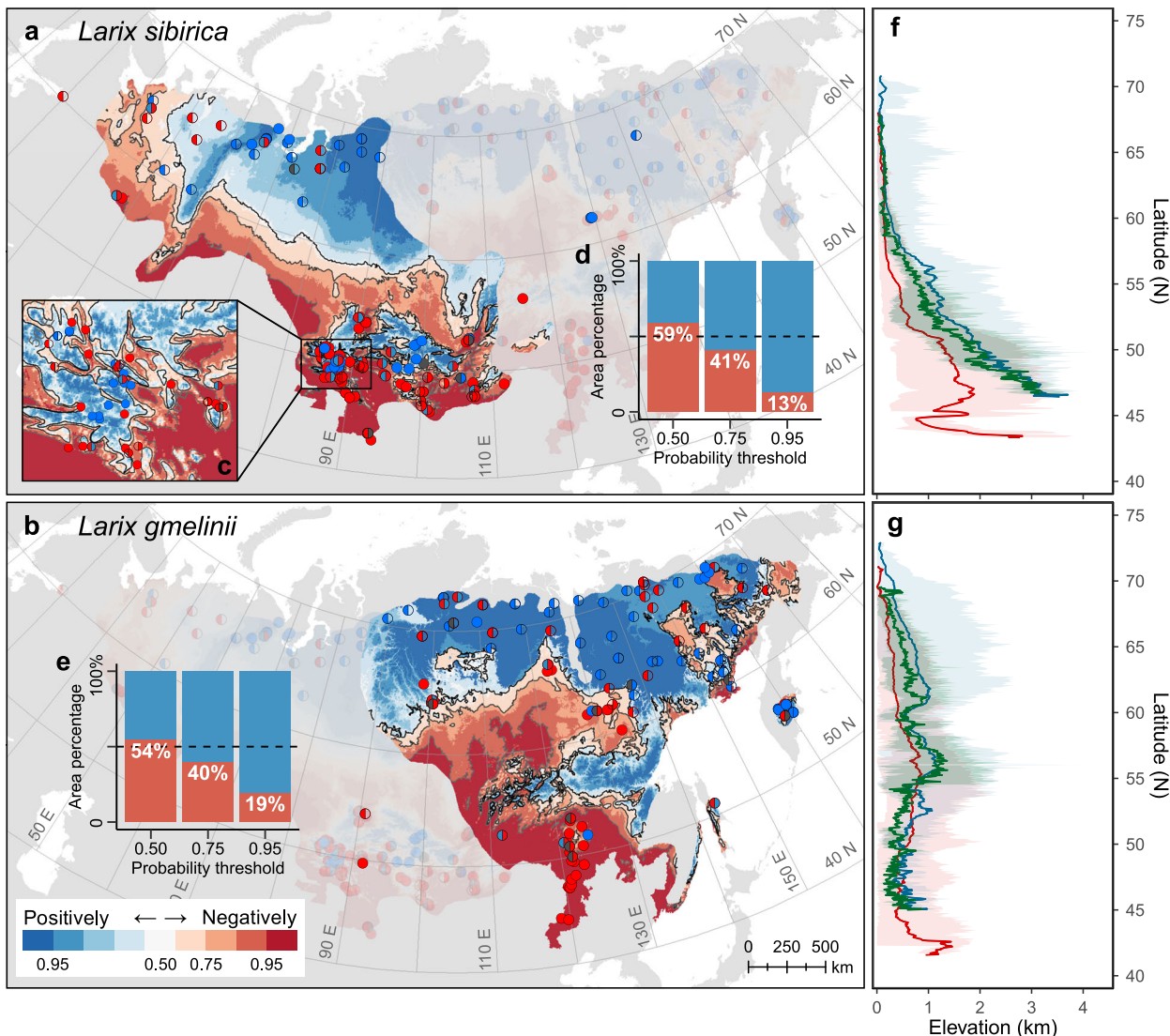

**Fig. 3 | Baseline (1979–2000) probability of showing negative growth-temperature responses across Eurasian boreal larch forests. a**, **b** Circles represent the sampling populations. The left and right halves of each circle are *colored* according to the population-level growth-temperature response patterns of the represented population during 1960–1990 and 1970–2000, respectively, where red and blue represent negatively-responding and positively-responding, dark gray represents no significant response, and no color represents lack of growth data or climate data. Red-white-blue gradient across the species distributions represents the estimated probability of showing negative responses decreasing from 1 to 0, species-specific probability functions were labeled on each plot; gray lines represent the 0.50, 0.75, and 0.95 iso-probability lines. Inset **c** shows dense sampling populations in the west Altai Mountains. Panels **a**–**c** were created using ArcGIS 10.2 for Desktop (ESRI, Inc). Insets **d**, **e** display the area percentage histograms of positively-responding and negatively-responding regions in species distribution identified by three probability thresholds. **f**, **g** Elevation profiles of the 0.50 iso-probability line (green), and the 0.50 probability-identified positively(blue)/negatively(red)-responding regions at a latitudinal resolution of 2.5′. Shadows and lines represent the maximum-to-minimum ranges and mean values of elevation, respectively.

Increasing temperature has amplified the drought stress on some formerly cold-limited forests in mid-latitudes or mid-elevations to the point of becoming a constraint on growth[12]. Our study expands this conclusion to a large proportion of Eurasian boreal larch forests. We report widespread negative growth-temperature responses and forecast a more continuous increase in negative temperature sensitivity throughout the 21st century. Eurasian boreal larch forests are projected to experience a further increase of 5.1 °C in annual temperature and 48 mm in annual precipitation by 2100 under the moderate SSP2-45 (Supplementary Table 1). Since 1 °C warming is accompanied by only 9.4 mm increase in precipitation, much lower than the slopes of the estimated climate boundaries, our models project that a large number of larch populations will cross the boundary to show negative responses to the increasing temperature.

This gap can be greater under the worst-case SSP5-85, where MAP/MAT increase ratio is expected to be 8.4 mm °C⁻¹ (Supplementary Table 1). There are substantial uncertainties in extrapolating the models estimated based on current growth-temperature relations to climate projections. However, the consistent qualitative spatial trends revealed here suggests manager and policymakers need to consider the possibility of emerging negative impacts of warming across much more of the Eurasian boreal forests than previously estimated.

Response patterns of the sampling populations and the estimated models suggest that the larch populations most likely to consistently benefit from warmer climate were generally located at higher latitudes or elevations. However, most larch populations are distributed at relatively low elevations (Fig. 1b), highlighting the increasingly

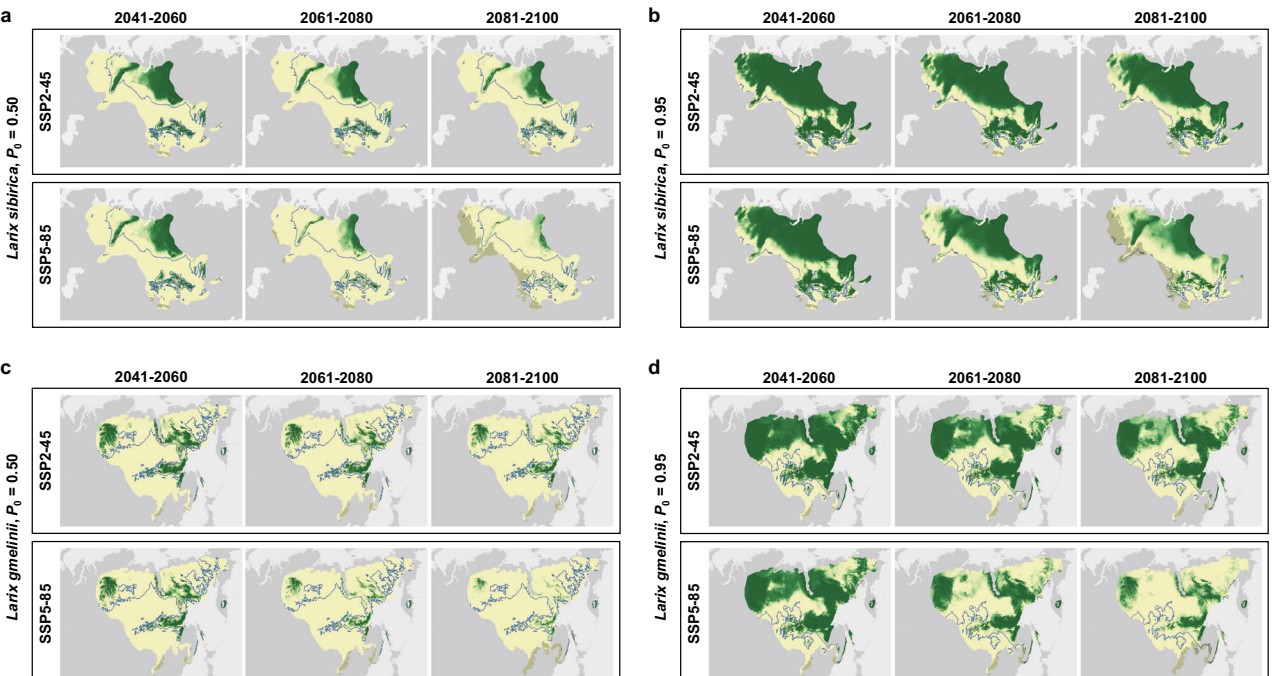

**Fig. 4 | Projected distributions of the positively-responding and negatively-responding regions under Shared Socio-economic Pathways (SSP) 2-45 and 5-85. a–d** Results of Siberian larch and Dahurian larch identified by *probability* thresholds ($P_0$) of 0.50 and 0.95, respectively. Light yellow to dark green represents the decreasing proportion of the climate projections from 25 GCMs that identified the negatively-responding regions under corresponding projection scenarios; blue lines represent the baseline (1970–2000) boundaries between the positively-responding and negatively-responding regions with corresponding probability thresholds. Gray shadows represent the distribution areas where projected climatic conditions falling outside the baseline climate space of boreal larch. This figure was created using ArcGIS 10.2 for Desktop (ESRI, Inc).

negative impacts of increasing temperature. It is also important to notice that tree density in the south of boreal Eurasia is much higher than that in the north[51]. Research over the last decade indicates that trees in dense forests are more vulnerable to drought stress[52–54]. If this holds true in boreal Eurasia and should the warming trend continue, or even accelerate, the southern parts of Eurasian boreal larch forests may be at a major risk of reducing productivity or even increasing mortality. Boreal forest communities in central Asia, western Europe, and western Canada have reported increased mortality associated with changing climate[2,28,55,56].

If further warming translates into a sharp reduction of populations that consistently benefit from increasing temperature, this may trigger substantial declines in forest productivity and carbon sink processed across much of boreal Eurasia[27], in line with the satellite-based estimates of vegetation productivity[8,9,14]. Even forests in the subarctic regions, which are usually assumed to respond rather homogeneously and positively to warming[24–26], may suffer under warmer climate as an increasing proportion of individuals and populations showing negative responses to temperature. Although cold temperature will most likely remain the main climatic limitation to most of Eurasian boreal forests for a long time, we clearly found emerging negative effects and diminishing warming-induced benefits, indicating the upcoming 'temperature tipping point'[57]. This shift could be attributed to increased evapotranspiration demand exceeding available moisture as temperature increases[5,48]. Recent studies using satellite observations and simulated vegetation proxies also have reported weakening low-temperature control, emerging negative impacts of warming[26,27], and future reversal of warming-enhanced vegetation productivity in boreal regions[14]. From the perspective of biogeography, the northward and upward emergence of negative growth-temperature responses may pronounce the modern formation of the Arctic-alpine disjunct distribution, which occurred in many taxonomic floras after the last ice age[58].

Interactions among nutrients, climate, and forest growth are complex and nuanced[11,47]. Further research is needed to estimate and verify the long-term implications of extensive changing climate sensitivity in boreal Eurasia, including but not limited to mortality and demographic trends. Understanding of Eurasian boreal larch forests can be further improved by increasing the density and spatial-temporal coverage of tree-ring collections[33]. Central Siberia is noticeably bereft of populations for study. The existing tree-ring network also could be oversensitive to climate due to the current sampling bias[31,59], though see ref. [60] as a counterpoint that it might not be an overestimate. Such lack of spatial representation and potential climate sensitivity bias may result in contrasting response patterns detected with the actual situation. Complementing tree-ring data with other multi-scale ecological information from forest inventories or remotely-sensed observations will allow us to better understand the relevance and consequences of the growth-temperature response patterns for the real functioning of the forests[31]. Incorporating the 'legacy effects' of climate[61], something not entirely incorporated in vegetation models, but omnipresent in trees, and comprehensive climatic variables such as vapor pressure deficit[50] into analysis is also necessary for fully assessing the impacts of warming on forests. Introducing these factors is the natural next step to understand the consequences of rapid warming for the future of Eurasian boreal forests.

Considering the intra- and inter-species differences in growth responses to temperature is fundamental to improving estimation of forest carbon storage and forecast of species distribution shifts[62,63]. Developing growth-temperature response metrics that are comparable across wide distribution is the key to achieve this goal. Explicitly introducing intra- and inter-annual temperature variabilities in correlation analyses has the potential to largely improve our ability to conduct more comparable and ecologically-meaningful research at large spatial scales. Our temporally-flexible and physiologically-relevant method better captures the highly variable growth responses, making

it a key aspect of understanding how boreal forests respond to changing climate, and appears to bridge the discrepancy between the temperature sensitivity reported in previous tree-ring research and that obtained from satellite-derived data. Our study also indicates that using calendar-based approaches likely underestimate the temperature sensitivity of boreal forests. This level of poor estimation likely has important consequences for the conservation and survival of boreal ecosystems under rapidly changing climate.

## Methods

### Tree-ring network

Our tree-ring network includes annual radial growth series from 131 Siberian larch populations and 129 Dahurian larch populations across boreal Eurasia. These data were collected from three sources: (i) ITRDB, 193 populations (7271 raw tree-ring width series from 4605 trees), 27 of which were contributed by us; (ii) fieldwork during the 2010s, three Siberian larch populations and 17 Dahurian larch populations (1273 raw tree-ring width series from 484 trees) from the Altai Mountains in northwest China and the Greater Khingan Range in northeast China, respectively, with the aim of expanding the data coverage to include the southernmost distributions of the two species; and (iii) previous research, specifically 47 population tree-ring chronologies (established based on 2785 core samples from 1826 trees) with accurate geographic coordinates and suitable temporal coverage digitalized from published literature for further spatially enriching our network. Of the data collected through our fieldwork, 25–50 mature trees were selected in each population and cored 2–4 times per tree at breast height using increment borers. Tree-core samples were preprocessed, visually cross-dated, and measured to the nearest 0.001 mm using LINTAB5 to obtain raw tree-ring width series. We used COFECHA to ensure the quality of cross-dating[64]. Our tree-ring network contains a total of 8544 raw ring-width series, of which 4274 are Siberian larch while the remaining 4270 are Dahurian larch.

To remove the low-frequency trends related to tree age or size, while accentuating the high-frequency climatic signals[65], each raw ring-width series was standardized by fitting a cubic smoothing spline with a 50% frequency cut-off at 30 years to it and then dividing it by the fitted curve to represent annual radial growth. We originally conducted a similar analysis using a 67% spline. Differences between the two approaches in the overall results were negligible. A small subset of growth series with obvious 'end effects' from calculating ratios were detrended by subtracting the fitted values from raw ring widths, while using a data-adaptive power transformation to stabilize the variance prior to detrending[66]. The resulting dimensionless growth series after detrending were then prewhitened to remove the autocorrelation by fitting an autoregressive model. The standardized and prewhitened growth series from the same tree were averaged into individual tree chronologies to create tree-level radial growth, while all growth series from the same population were averaged into population chronologies using bi-weight robust mean[67] to create population-level radial growth. That is, we conducted analyses on both individual trees and populations. Our identification of sampling trees for some ITRDB data is speculative due to the complex and various naming conventions the contributors adopted. Therefore, we compared the correlations of the single standardized growth series and individual tree chronologies with temperature, as well as the correlation-derived metrics. The consistency of these results (Supplementary Fig. 12, Supplementary Table 4) suggests that the single growth series can be a good substitute for individual tree chronology to represent tree-level radial growth. In the Main text, we used single growth series to represent tree-level radial growth for relevant analyses, and provided the correlation results calculated from individual tree chronologies in Supplementary Part C for reference. To test whether our subsequent analyses were robust to the choices of detrending methods, we tested three commonly-used detrending methods, and with or without

prewhitening, and found only subtle but no essential changes in our main results (Supplementary Fig. 13). The above procedures were conducted using the package *dplR* (v1.7.4)[68] in R platform[69].

The ITRDB data and the published chronologies compiled into our network were collected or established primarily for the original purposes of dendroclimatic reconstructions and investigations of growth-climate responses (Supplementary Data 1). Therefore, most of these data were collected following the classic dendrochronological sampling design that increases the climate signals (Supplementary Data 1). Such sampling bias could make the network oversensitive to climate, thus creating the climate sensitivity bias[31,59]. Integrating and cross-validating the existing tree-ring data with extensive unbiased sampling is the most effective way to avoid the potential bias[31,33], while is currently not feasible for boreal Eurasia. However, Nehrbass-Ahles et al.[60] report that investigating growth-climate relationships is largely unaffected by the sampling design. The climate sensitivity bias is also found not to exist when comparing ITRDB data with other independent unbiased reference networks[12,16]. Moreover, our tree-ring network evenly covers both the distribution areas and climatic spaces of the two larch species, showing similar climatic probability density distributions to the species distribution areas (Fig. 1; Supplementary Fig. 2). Given these reasons, we consider our tree-ring network to be suitable for our current analyses and research purposes. The shapefiles of the larch species distributions were digitized and merged from multiple repositories[41,70,71].

### Climate datasets

For the growth-temperature response analysis, we used the continuous daily mean temperature records from 110 meteorological stations (Fig. 1, Supplementary Data 2) within the study area from the Global Historical Climatology Network (GHCN)[72] and the China Meteorological Data Service Center, matched with the sampling populations according to the principle of proximity.

We used the gridded climate layers of mean annual temperature and mean annual precipitation at a spatial resolution of 2.5′ from WorldClim[73] to characterize the near current and future climate features across the distributions of the two larch species. WorldClim provides gridded average climate data for two 30-yr periods, 1960–1990 and 1970–2000, as well as downscaled CMIP6 gridded climate projections from 25 general circulation models (GCMs, Supplementary Table 5) over four projection periods (2021–2040, 2041–2060, 2061–2080, and 2081–2100) and under four Shared Socio-economic Pathways (SSP1-26, SSP2-45, SSP3-70, and SSP5-85). The baseline period was 1970–2000. We used Climatic Research Unit Time-Series version 4.05 (CRU TS 4.05)[74] monthly gridded climate dataset at a spatial resolution of 0.5° to display the 1960–2020 climate trends across the Northern Hemisphere.

### Constructing temperature series using the T-linked method

Most growth-temperature response analyses correlate annual growth series with temperature time series for rigid calendar periods (months, seasons, and/or annual). For forests under continuous climatic stress or in regions with consistent growing seasons, using monthly or seasonal temperature series is likely a simple, yet effective approach to investigate growth-temperature relations. To refine such analyses, many studies have also developed bi-weekly or even daily temperature series (e.g., refs. 75, 76). Another approach, sliding correlation analysis, constructs temperature series through sliding a fixed-length window synchronously over the daily temperature series of all years, and correlates growth series with all these stepwise temperature series to finely detect growth responses (e.g., refs. 77, 78). These methods are all designed to improve the detection of growth-temperature response by increasing the resolution and flexibility of correlation analysis. The temperature series used in these methods, however, are still based on fixed periods linked to human calendars.

We developed a method to construct temperature series linked to intra-annual temperature variability that is more physiologically-relevant and effective in capturing comparably short and variable growth-temperature response windows, particularly for boreal larch forests. This so-called 'temperature-linked' or 'T-linked' method constructs temperature series based on movable periods as determined by two movable dates translated from defined temperature values, rather than fixed calendar periods. Our approach frees the temperatures to be correlated with tree growth from the constraints of human calendars. Five consecutive days is a widely-used validated threshold in meteorological research to determine a stable climate stage or process[79]. Therefore, the temperature values are translated into dates following the five-day pass rule in T-linked method. For each temperature value, the last day of the first five consecutive days in a year with daily mean temperatures all at or above this value is identified as its anchoring date for that year. Following this rule, we can use two independent temperature values with their anchoring dates in each year as start and end points, respectively, to determine a period in each year when trees are more likely developing xylem in their stems, and then to construct a temperature series by averaging the daily mean temperatures over these T-linked periods. The date translated from the same temperature value varies from year to year, as does the period determined by the same pair of temperature values. This flexibility implies that the correlation window also varies from year to year, capturing the inter-annual variations in temperature that boreal forests experience.

Satellite-derived data suggests that the thermal limits for the start and end of the growing season across boreal Eurasia can be as low as 0 °C[40,43]. To ensure coverage of our analyses throughout the growing season, the selection range of temperature value was set between 0 °C and the mean annual maximum daily mean temperature record of each meteorological station. Furthermore, considering the quasi-bilateral symmetry of intra-annual temperature variability in a continental climate, the final selection range is defined as a self-symmetrical set of integral multiples of 0.5 °C that starts from 0 °C to the highest value and then returns to 0 °C. The translations of the temperature values before the highest value are conducted from the first day of the year, while those after the highest value are conducted in reverse order from the last day of the year. Any two values within the selection range can be paired to construct a T-linked temperature series through the above processing, thereby a batch of temperature series can be constructed for each meteorological station for flexible growth-temperature correlation analysis. See Supplementary Part D for greater details on the T-linked method.

### Growth-temperature response analyses
The WorldClim climate layers at a spatial resolution of 2.5′ used to characterize long-term local climatic conditions are aggregated over two periods: 1960–1990 and 1970–2000. In terms of growth data, specifically, 258 of the 260 populations were sampled in or after 1990, 68.5% of the populations north of 60°N were sampled before 1995, and 77.9% of the populations south of 60°N were sampled after 1999 (Supplementary Fig. 14). Based on these, we conducted growth-temperature correlation analyses in parallel for both 1960–1990 and 1970–2000 to maximize the matching of growth data and local climatic conditions.

We excluded the growth series shorter than 25 years during the analysis period. In total, 6144 and 4183 single standardized growth series (3,638 and 2446 individual tree chronologies) were retained for correlation analysis during 1960–1990 and 1970–2000, respectively. At the population level, 117 and 94 Siberian larch population chronologies were used for correlation analysis during 1960–1990 and 1970–2000, respectively, compared with 124 and 64 for Dahurian larch. We calculated Pearson's correlations between each growth series (single growth series, individual tree chronology, and population chronology) and all T-linked temperature series developed at the nearest meteorological station for each population. Growth series (tree-ring chronologies) showing significant correlations ($p < 0.05$) with ≥3 adjacent T-linked temperature series (i.e., ≥3 adjacent colored tiles in Supplementary Fig. 16) were considered sensitive to temperature and, hypothetically, indicating that temperature affected the radial growth of corresponding trees or populations.

### Metrics to measure growth-temperature correlations
We calculated the mean coefficient of all significant correlations (Pearson's $r$, $p < 0.05$) between each population chronology and the T-linked temperature series from the nearest meteorological station. We further established four metrics based on the tree- and population-level growth-temperature correlation results to measure the size of temperature effects on tree growth and to characterize the structure of growth-temperature response pattern for each population from two dimensions: (i) affecting scope, i.e., what is the proportion of tree individuals in a population that show a significant relation to temperature; and (ii) affecting duration, i.e., how long the growth response to temperature lasts. Each metric was calculated separately from negative and positive growth-temperature correlations to characterize negative and positive temperature sensitivity of each population, respectively.

As the time period used to construct the same T-linked temperature series varies inter-annually, we calculated the mean length (in the number of days) of the varying T-linked periods over corresponding analysis period for each T-linked temperature series. We also calculated the percentage of single growth series (individual tree chronologies) in each population showing significant negative or positive correlations with each T-linked temperature series from the nearest meteorological station. The maximum value of these percentages was used as the first metric, that is the maximum proportion of individuals in a population that show negative or positive sensitivity to temperature (MST, %), representing the maximum scope of negative or positive temperature effects on each population. We then ranked all T-linked temperature series correlated with the same population in descending order of the percentage of single growth series (individual tree chronologies) showing significant negative or positive correlation with them, and averaged the mean lengths of the temperature series ranked top 5% and 50% as metrics tDur05 and tDur50 (days), respectively. Another metric was derived from the population-level correlations, corresponding to the average mean lengths of all T-linked temperature series with which the population chronology was significantly negatively or positively correlated (pDur, days). We suspect that the last three metrics can be interpreted as a way of describing the mean duration of negative or positive growth responses to temperature for a population.

We divided the 30-yr population chronologies sensitive to temperature into two groups based on whether they showed negative responses to temperature: (i) positively-responding group, composed of chronologies showing significant positive responses to temperature but no negative responses; and (ii) negatively-responding group, composed of those showing significant negative responses to temperature, regardless of showing or not showing positive responses. The positively-responding group refers to populations where growth was expected to consistently benefit from increasing temperature, while the negatively-responding group refers to populations where growth would decline with high temperatures during portions of the growing season, regardless of the net effects of warming on growth. We located all 30-yr population chronologies in climate space based on their local climatic conditions over the corresponding time periods to demonstrate the climatic differentiation of negative and positive population-level temperature sensitivity.

## Validation of the effectiveness of the T-linked method

To validate the effectiveness and advantages of the T-linked method, we compared our method with two other methods that construct temperature series based on fixed periods. Comparisons were performed under the condition of calculating correlations of the same order of magnitude to demonstrate that the significant growth-temperature correlations detected by the T-linked method were not simply due to the increased number of correlation calculations.

(i) moving calendar-based method. We constructed the calendar-based temperature series using 36 time windows fixed to the calendar with variable widths from 15 days to 120 days at intervals of 3 days. We run each of these fixed windows over daily mean temperature series between April 1st (DOY 91) and September 30th (DOY 273), and averaged the daily mean temperatures within each window year by year to construct the temperature series. As a result, each growth series was correlated with 2106 calendar-based temperature series, a number that exceeds the average number of T-linked temperature series correlated with each growth series, which is 1548. The commonly-used seasonal, bimonthly, monthly, and semi-monthly temperature series were all included in the analysis through this way.

(ii) calendar-based T-linked method. We averaged the inter-annually varying start and end dates of the T-linked periods used to construct each T-linked temperature series, and then averaged the daily mean temperatures between the two mean dates year by year to construct a calendar-based T-linked temperature series. Each growth series was then correlated with all the calendar-based T-linked temperature series that corresponded on a one-to-one basis to the normal T-linked temperature series.

## Statistical analyses

We calculated the partial Pearson correlations (Pearson's $r$) between the above-mentioned correlation-derived metrics and local climatic conditions (MAT and MAP) to reveal how local climates affect the growth-temperature response pattern after the covarying effects of MAT and MAP being controlled for. We used two-sided Student's $t$-tests to assess the significance in the differences between the negative and positive temperature-sensitive populations regarding latitude, longitude, and local climatic conditions.

We built species-specific logistic regression models using climate coordinates (MAT and MAP) as independent variables to estimate the probability of showing negatively growth-temperature responses for a boreal larch population (Eq. 1). The groupings of temperature-sensitive 30-yr population chronologies and their corresponding climatic conditions were used as observations for parameter estimation. We calculated McFadden's *pseudo-$R^2$* for each logistic model[80]. Given a probability threshold, the sigmoid logistic functions (Eq. 1) can then be transformed to linear functions (Eq. 2), that is, the climate boundary that distinguishes between positively-responding and negatively-responding populations.

$$P = \frac{1}{1 + e^{w_1 * MAT + w_2 * MAP + w_3}} \qquad (1)$$

$$w_1 * MAT + w_2 * MAP + w_3 = \ln\left(\frac{1}{P_0} - 1\right) \qquad (2)$$

where, $w_1$, $w_2$, $w_3$, the parameters to be estimated; $P$, the probability of showing negative responses; $P_0$, the given probability threshold.

We tested the quality and accuracy of the estimated logistic models using 10-fold cross-validation analysis[81]. The observations for parameter estimation were divided into 10 subsets that were roughly of equal size. Nine of these subsets were used as training group to estimate a cross-validated model while the 10th subset was used for testing. We calculated the classification accuracy and the Cohen's kappa ($\kappa$) of the cross-validated model applied to the test group to measure the classification performance. This procedure was executed 10 times, with each subset taking turns as the test group. The resulting 10 classification accuracies and 10 Cohen's $\kappa$ were averaged as estimates of the overall predictive performance of the final model.

Using the calculated species-specific probability functions (Eq. 1), we estimated the baseline (1970–2000) probability of showing negative growth-temperature responses across the species distributions. Each species' distribution was divided into positively-responding and negatively-responding regions by given identification probability thresholds (Eq. 2). To investigate how the two types of regions may change in the future, we applied the climate boundary functions to the climate projections from 25 GCMs over three time periods (2041–2060, 2061–2080, and 2081–2100) under four SSPs (SSP1-26, SSP2-45, SSP3-70, and SSP5-85). To prevent the estimations from depending on our choice of probability threshold and to accommodate the potential climate sensitivity bias of our tree-ring network, we compared the results of using three gradient thresholds ($P_0 = 0.50$, 0.75, and 0.95) to identify the two types of regions for both baseline and future periods. We then calculated the proportions of GCMs that identified each 2.5′ grid cell as negatively-responding using the three probability thresholds, respectively. In this study, we mainly discussed the negatively-responding regions identified by more than 75% ($n \geq 18$) of GCMs for the credibility of our projections. The cross-combination of the three probability thresholds and the SSPs constructs the gradient scenarios for projecting the two types of regions (Supplementary Fig. 15). We acknowledge that the thresholds for regions where larch populations showing and not showing negative growth-temperature responses are not strict boundaries, but potential areas where corresponding growth outcomes would be most likely to occur under future warming and that in the areas near the thresholds would have greater uncertainty. Considering the uncertainties in extrapolating the models to climate projections falling outside the data used to fit the models, we marked the areas where projected climatic conditions beyond the 1970–2000 baseline climate space of boreal larch. These areas were limited and concentrated in the southern margins of species distributions (Fig. 4; Supplementary Fig. 6).

The above analyses and calculations were all conducted in R platform[69], using the packages *caret* (v6.0-93)[82], *data.table* (v1.14.6)[83], *dplyr* (v1.0.10)[84], *ggplot2* (v3.4.0.9000)[85], *ggpubr* (v0.5.0)[86], *gtools* (v3.9.4)[87], *maptools* (v1.1-6)[88], *raster* (v3.6-13)[89], and *rgdal* (v1.6-4)[90].

### Reporting summary

Further information on research design is available in the Nature Portfolio Reporting Summary linked to this article.

## Data availability

The ITRDB tree-ring width data was obtained from https://www.ncei.noaa.gov/products/paleoclimatology/tree-ring. The daily mean temperature records from the meteorological stations operated by the Global Historical Climatology Network and the China Meteorological Data Service Center were obtained from https://www.ncei.noaa.gov/cdo-web/search?datasetid=GHCND and http://data.cma.cn. The shapefiles of species distributions we digitized, all tree-ring data we used, and the daily mean temperature records are available via https://doi.org/10.6084/m9.figshare.23296082. The Worldclim datasets were obtained from https://www.worldclim.org. The CRU TS 4.05 dataset was obtained from https://www.uea.ac.uk/groups-and-centres/climatic-research-unit.

## Code availability

We used R software for statistical analyses, computations, and visualization. The code of T-linked method and other necessary codes used for data analysis and visualization are available via https://doi.org/10.6084/m9.figshare.23296082.

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

## Acknowledgements

We acknowledge all the contributors who shared the tree-ring data of boreal larch species to the ITRDB. We thank the Harvard Forest for the financial support and for providing an intellectually stimulating environment to develop this study. This study is supported by the National Natural Science Foundation of China (42171049 and 41630750) and the National Key Research and Development Program of China (2018YFA0606101). Wenqiu Ma is supported by the National Natural Science Foundation of China (42001199). Shoudong Zhao is supported by the National Natural Science Foundation of China (42101138) and Basic Research Fund of CAMS (2021Y006).

## Author contributions

W.L., Y.J., and N.P. conceived the study. W.L. and R.D.M. performed the analyses. W.L. drafted the manuscript. R.D.M., N.P., Y.J., T.R., E.D., W.M., and J.W. contributed to the editing and writing of the manuscript. T.R., E.D., S.Z., J.W., X.C., H.X., and X.K. provided constructive comments and indispensable guidance for the study. W.L., Y.J., E.D., S.Z., N.P., M.D., and F.W. conducted fieldwork and sample processing.

## Competing interests

The authors declare no competing interests.
