## [Peer Review File · Nature Communications]

REVIEWER COMMENTS

Reviewer #1 (Remarks to the Author):

Dear authors,

I read your manuscript submission to Nature Communications, "Reassessment of growth-climate relations indicates the potential for decline over Eurasian boreal larch forests," with great interest. The manuscript describes a spatially extensive and sophisticated investigation into the growth-climate relationships of boreal Eurasian larch (2 species) using an insightful and novel method to aggregate climate data. Rather than rely on monthly aggregates of climate data (which are of fixed length and period year-to-year), the authors define potentially physiologically relevant periods using daily temperature thresholds (which may vary in length and period year-to-year). The authors propose that by using this method, they can obtain much more robust and accurate estimates of growth-temperature relationships. Indeed, using >8,500 larch tree-ring series, they found that many more trees and populations show significant correlations with their temperature time series (>90%) vs. the monthly aggregated series (~50-60%) in each of the climatic periods they investigated (1960-1990 & 1970-2000). In order to gain insight into possible mechanisms driving the growth-temperature relationships, the authors next described the climate spaces, in terms of mean annual temperature and mean annual precipitation, which were more associated with positive and negative growth-temperature relationships. Finally, the authors use the results of this last analysis and climate projections from a suite of GCMs to predict how areas suitable for or detrimental to larch growth may shift over their study region in the coming century. The authors clearly did an impressive amount of work.

I think that this study has the potential to greatly influence future work on tree growth-climate relationships globally. Tree rings are unmatched in ecological science for their temporal extent, resolution, and accuracy and therefore represent one of the best tools we have to answer critical questions about how trees and forests have responded and will respond to changes in climate. Tree rings themselves and assessing their relationship to climate variables present several challenges. This paper represents a major advance in the latter challenge.

There are, however, several issues with the paper in its current form that I believe require major revisions before it is suitable for publication in Nature Communications. Given the obvious level of skill and sophistication that the authors possess, I believe all of my suggestions are achievable. The first major issue is that the method they used to process (detrending & standardization) their tree ring data is inappropriate for their goals. Some percentage of the Pearson correlations with climate will be invalid as a result. The suggestions I made should fix these issues. The choice of tree ring detrending methods can be enormously consequential and can confound subsequent analyses. I think the authors have not adequately considered these pitfalls, instead relying on software default methods. The second major

issue I have is with the formal analysis to project areas that will be suitable for or detrimental to larch growth (“warming-benefitted” and “warming-threatened”) based on their results and GCM predictions. This analysis consumes a major portion of the manuscript, yet it appears to have a weak premise: that the nature of growth-temperature relationships during 1970-2000 will carry forward through 2100. The authors themselves report that there were changes in the proportions of trees recording positive and negative correlations with climate from 1960-1990 to 1970-2000. I think that devoting so much of the manuscript to an analysis that relies on extrapolation (and could very likely be wrong) risks turning a potentially very impactful piece of work into a slick but disposable article.

Instead, I suggest that the authors delete the projection analysis and instead focus on improved explanation of their novel t-linked method (including better tests of its efficacy) and a more clear development/description of their 4 growth-climate metrics (replacing or adding figures too). I think they should retain their analysis of the climate space associated with positive and negative responses to temperature - this is very useful in providing insights as to what the underlying mechanisms are that produce the varying responses to climate. I think you should expand on these mechanisms/hypotheses in the discussion. In addition, a very informative additional set of analyses would be to consider how their conclusions about the mechanisms driving growth-climate relationships would have been different for traditional rigid monthly climate time series vs their more sophisticated time series.

I catalogue the bulk of my detailed suggestions and critiques in the line-specific comments below.

Line-specific comments for the main text:

Line 2: If you keep this title, it should read “across Eurasian...” rather than “over Eurasian...”.

Line 24: The abstract is poorly-written and difficult to interpret. I request that you substantially revise it in your revisions.

For the most part the other parts of the paper are very well written.

Lines 25-26: Unclear where this 30% figure comes from. It is not cited in the introduction. Is it derived from the Isaev et al 1995 paper? Are there not more recent efforts to estimate species or genus-specific cover? Either clearly cite this in the intro or walk the reader through (in a sentence) how you derived it.

Lines 28-30: Revise this sentence for clarity. What do you mean by universal? Positive responses to warming? Negative responses to warming? Both? If both, then the responses are not universal. Also, since you have mentioned larch already I presume that the previous studies are on larch growth-climate relationships specifically? Perhaps you can move the larch focus to after this sentence - I think most growth-climate relationships in trees globally (not just in boreal larch) use monthly data.

Maybe something like this would be more clear:

“Most studies thus far have relied on monthly-aggregated climate data to assess tree growth relationships with climate, detecting monotonic responses to warming at boreal forest margins but no clear responses in vast interior regions. However, such rigid calendar-based aggregates may not adequately capture physiologically-relevant periods that vary interannually.”

Lines 32-33: You should say “individual tree-ring series.” Some of those series represent radii from the same trees. Chronologies refer to aggregates of many tree series.

Also, a more clear and accurate sentence would be:

“...8,566 tree-ring series aggregated to 260 population chronologies.” Either way, wow! Very impressive sample size.

Line 34: “across the distribution of boreal Eurasian larch”

Line 35-37: I don't know what this sentence means. Please revise for clarity.

Line 48: I am thankful that you kept the more standard in-text citation style for this stage of submission - it makes it easier to keep track of where your ideas come from. I'm guessing the editor will want the Nature numbered style in the next round of revision, however. *[Editor's note: since the reviewer finds it helpful, for the time being you may keep the current reference style]*

Line 65: Great maps!

Lines 91-96: One explanation for a mis-match between ITRDB-based analyses and satellite-based ones is that ITRDB data do not derive from any kind of objective sampling design. In fact, ITRDB records may have been selected in attempt to maximize certain climate signals for the purpose of dendroclimatological reconstructions. This means that ITRDB records may not be suitable for making general inferences across regions. I don't see that you have really grappled with this potential issue in your paper.

See (DeRose et al., 2017, Building the forest inventory and analysis tree-ring data set, Journal of Forestry 115(4), pp283-291) for a detailed discussion of this issue.

It seems to me that at minimum you would need to know something about how these data were collected (ITRDB links the papers) and filter out those that were not appropriate to your study. Alternatively you could limit your inferences based on lack of knowledge of what kind of sample design your data comes from.

Lines 105-106: This is a really great point and very compelling!

Lines 108-110: "High intra- and inter-annual..." This sentence is unclear, please revise.

Lines 125-126: You might describe these in more detail here if you can fit in the text. I think you need to describe them in more detail elsewhere (current explanations are not adequate).

Line 133: This is misleading because you trimmed down this dataset to 8,566 series. Really, you should report the number of trees, not measurement series. Trees and populations are your units of analysis, correct?

Line 139: Orange, not organ?

Line 141: The grey squares are hard to see. Maybe black-outlined squares with white fill would be more clear?

Lines 146-147: You need to report what you did with the monthly series somewhere in the main text or in the supplement - I didn't see any description of your methods for the monthly data anywhere. What was the data source (ideal comparison would be to aggregate your daily station data to monthly means)? What correlations did you perform, just individual months or aggregates of 2-12 months? I hope you did both.

The key here is that the comparison is as fair as possible - this is a central question in your study and we can't take it on faith that the comparison was set up in the best way possible.

It is also important to keep in mind that there are a few plausible reasons why you would find many more sig. correlations with your method vs. monthly data (assuming you treated the monthly data as I have described above). One is your conclusion that it is due to capturing the year-to-year variability in relevant temperature periods. This makes sense to me too. However, another explanation is that you have simply computed many more discrete intra-annual periods: something on the order of 3000 possible periods within a year for your t-linked method (do I have that \pm right?) vs. ~ 78 possible contiguous month combinations. Given these ballpark numbers alone, it is no surprise that you would find more sig correlations. A real test of your hypothesis that more sig correlations was due to capturing the same temperature period in each year (regardless of what dates it covers) would be to compute fixed year-to-year periods using the daily data. You could constrain the rigid daily periods to ≥ 10 days, etc. to match your variable t-linked periods (e.g., 5 April to 10 May for all years 1970-2000). This would generate a number of possible series that is in the same order of magnitude as your t-linked series. My guess is that year-to-year rigid daily series will yield more significant correlations than with rigid monthly data. If you are correct that the year-to-year flexibility is the key, then you should still get more sig correlations with the t-linked series.

Line 148: Report the sample sizes with the percentages.

Lines 149-150: Why not 1980-2010 as well? This would bring your analysis closer to the relevant recent time period.

Line 154: Are you sure these are individual tree tree-ring series? It is common practice to extract multiple cores from a single tree.

Also, report sample sizes whenever you report these percentages because n is not equal through all these comparisons.

Line 154: The change in proportion of sensitive trees is an indication that growth-climate relationships are indeed unstable over time. Because you essentially only have 2 data points here, why not add another for 3? 1980-2010. The number of larch populations dwindles as you move forward into time, but it seems like the most recent possible period is too important to skip. One thing you could do beyond simply calculating percentages of chronologies with sig. correlations is focus on the trees/populations that cover all 3 periods and report how their growth-temperature relationships have changed through time. You could get a much better sense of this by advancing your 30-year moving windows by 5-year (or even 1-year) instead of 10-year steps. Then you could plot time on the x-axis and 30-year correlation coefs on the y. I don't think this would be too much additional work if you are already calculating t-linked series for 1960-2010. The insight gained would be really valuable. Perhaps this would aid you in your interpretation of warming-threatened and warming-benefits trees/populations/areas.

Lines 164-165: It is difficult to assess this claim without yet understanding the method (I needed to read the supplement to begin to understand it). What do you mean by “temperature variations”?

After reading the Supplement file, I agree with the first part of the sentence (this is an effective approach over monthly climate series).

Line 167: You need to describe what you did with the monthly data too - at least briefly in the methods and in detail in the supplement. In order for your methods to be comparable, I hope that you examined both individual months and aggregates of months (i.e., seasons).

Lines 172-173: What does this mean? Also, incorrect grammar.

Lines 173-175: we don't know what “the four” are at this point, so maybe just say “four metrics” and give a description of what they are. I think the lack of description of these metrics and how they are derived is a major weakness of this paper as it is currently. At minimum, a whole supplement section should be devoted to explaining these metrics. They are critical as you don't report actual correlation coefficients (I realize this may be impractical in the main text due to the complexity of your method).

Line 177: “...both wider affecting scopes and longer affecting durations.” I don't understand this.

Lines 189-190: Grammar.

Line 198 (Fig. 2): This figure is important, but I'm not sure it is illustrative enough to warrant inclusion in the main text. Panel b should definitely move to the supplement. Panel a could also go to the supplement, or you could probably express the same data in a more effective way by plotting each of your metrics (y-axis) against MAP and MAT on the axes. You could then print the correlation coefficients within the plots. This would be many more panels, but it would be a lot easier to interpret.

Line 266: I am not convinced that this analysis is valid. The analysis is sophisticated, but you are essentially extrapolating growth-temperature relationships from 1970-2000 into the future. Perhaps you are capturing some of the mechanisms involved in the growth-temperature relationships by using MAT and MAP in your model - is this what you have assumed? I think this is clever but would like to see the authors address this concern explicitly here or in the methods.

Line 392: “which report” instead of “where report”

Line 439: Did you compute tree means for your samples and those from the ITRDB? If not, you should!

Line 443: remind us what the analysis periods are

Lines 445-448: Good choice using the raw ring width series from ITRDB (and making your own chronologies) rather than ready-made chronologies. Also, it is great that you tell the reader what methods you used to do this. However, there are still some potentially major issues with the method you chose. This is a crucially important decision in your analyses - perhaps the most important decision.

You are correct that the main reason to detrend/standardize tree ring series is to remove the signals (noise in your case) that are due to endogenous biological processes related to wood production. These signals manifest in 2 main ways: trends in the magnitude of ring widths (e.g., neg. exp decline or hughershoff-type decline) and trends in the variance of ring widths (series are typically heteroscedastic). There are many ways to handle this, each with its own pitfalls. The typical methods describe and remove lower-frequency (user-specified) trends and stabilize the variance. The “art” of detrending is really to choose the method(s) that minimize the pitfalls for your particular analysis - there is no universally perfect detrending method. Most of the “conservative” detrending options fit some kind of curve to the raw ring widths, then take the ratio of the raw ring widths:fitted values. The ratio stabilizes variance, but can also introduce artifacts, particularly at the ends of the series (or anywhere when the curve doesn’t fit the data well). See the 2 papers below for an overview on these issues.

Your ultimate goal here is to have tree-level and population-level series/chronologies that reflect reality and are statistically sound for your 30-year correlation periods. Especially because you are comparing differences in relationships between 1960-1990 and 1970-2000, it is critical that the variance is stabilized over time and that there are not artifacts from the detrending method at any frequency. Both of these could confound your analysis.

The default dplR “Spline” option is unlikely to accomplish what you hope it would (remove age/size trends) in all of these series. In fact, it may add some artifacts of its own. Using variable wavelength splines on series of different lengths doesn’t make sense to me. For example, a $2/3 \times \text{series length}$ wave length spline gives 100 years for a 150-year series and 33.3 years for a 50-year series. These are very different levels of stiffness, so you are removing signals of varying frequencies for each series. This seems like a bad way to go about this. Your 30 year correlation periods could easily be influenced by artifacts from your detrending method.

The authors of dplR warn against adopting detrending methods without careful inspection in the description file for `detrend.series()` : “Automating detrending and not evaluating each series individually is folly.” I agree with this, although it presents a challenge when you have 8.5k series! Perhaps you could look at a subsample of series individually, or better yet, make 260 plots (one for each population) and plot all tree-ring series contained within with thin lines. These will be noisy plots, but any weird end effects or other outliers should be obvious. There really is no substitute for looking at plots of the data. You will have to do some of this.

I can't tell you what method will be the best for your tree ring data, but perhaps I can give you some direction. I would 1st try a constant 30-year spline fit (will remove all frequencies lower than that, which you can't assess anyway) and use ratios (`difference = FALSE`) to stabilize the variance. Take a look at some plots to make sure there aren't weird end effects from computing ratios. The 30-year spline should be flexible enough to fit most series well and avoid end effects. If not, consider Cook & Peters (1997) power transforming & residual method with the 30-year spline (transform is done before curve fitting to stabilize variance, then curve is fit, then transformed rw series are subtracted from the fitted values). Which ever method you end up with, please also describe in the methods section why you chose this method. I think also you need to compute AR residual series (“prewhitening”) after detrending and before aggregating to chronologies - this will remove any autocorrelation in the series that could make your Pearson correlations invalid. Note: if you use Cook & Peter's method, you will have to add a constant (say 1 or the original series means) to the transformed residual series before prewhitening.

See

Coulthard et al., 2020, The limits of freely-available tree-ring chronologies, *Quaternary Science Reviews* 234

Cook & Peters, 1997, Calculating unbiased tree-ring indices for the study of climatic and environmental change, *The Holocene* 7(3)

Line 470: Nowhere in here do you report how you prepared the data and performed analyses for the rigid monthly climate series that you show results for in Table 1. Did you examine individual months as well as aggregates of months?

Line 474: “yet-effective” is an inappropriate use of hyphenation. Correct would be: “...a simple, yet effective, approach...”

Line 483: Well, there are multiple human calendars. Most of the climate datasets use the western Gregorian calendar. Either way, they are arbitrary relative to tree physiology. You could say “linked to human calendars”.

Lines 503-504: This must be temperature at the start of green up, right? Good check on temperature limits of growing seasons.

Line 518: You might specify the WorldClim layers as 30-year climate normals. Give the resolution of WorldClim pixels here too.

Lines 532-533: Are Pearson correlations appropriate here? Given the nature of the tree ring data (and your choice of detrending method), it seems the IID assumption will not be met here. Maybe the climate and (detrended) tree ring series are \pm identically distributed, but with autocorrelation present in the tree ring series (maybe the climate series too) observations are not independent. Probably the best solution is to compute AR residual series of your detrended tree ring series (aka "prewhitening"). This will remove all but the high-frequency (interannual) variation in the series. This should be fine - even desirable - in your case because you are only doing correlations for 30-year periods. See my detailed comments above on detrending/ standardizing your tree ring series.

Lines 542-545: This is fantastic. I've noticed a tendency recently for researchers to lump significant positive and negative correlations together into one metric of sensitivity - obscuring crucial ecological differences.

I was pleased to read this, then confused when in the next sentence you appear to combine both negative and positive correlations into a single metric. You probably could reduce this confusion by substituting "negative/positive" with "negative or positive".

I also think you could do a better job clearly explaining your metrics - it is still hard to understand such multi-layered metrics intuitively. The simple stuff (the actual correlation coefficients) is buried fairly deep here.

Line 545: "maximum proportion" Is this the maximum proportion out of all the different temperature series (with different L & R values)? This is not clear as written.

Line 550: not clear what the "average mean" is. Please explain in more detail.

Lines 559-561: This doesn't make sense to me, how can a chronology show both negative and positive correlations within the same 30-year period? Would this be for different temperature series? You should clarify.

Lines 620-623: I hope that you will strongly consider archiving your code with an online database that is freely available to anyone interested. You have developed a great method that others should be able to play around with (and perhaps improve).

Line-specific comments for the supplementary information:

Line 14 (Fig. S1): I think this figure would be more helpful if it was just your study region (like figure 1) instead of all of the northern hemisphere.

Line 31: experience would be a better word than "suffer"

Line 110: There are a lot fewer populations represented in the 1970-2000 period than in 1960-1990 (as you report in the text). If you are accepting diminishing sample sizes here, why not cover the next 30-year period (1980-2010) too? It might give you some insight into how growth-climate relationships are changing over time.

Line 127-140: This is fantastic. I am really glad you have taken on this task! I have one challenge to your approach: wouldn't heat sums (e.g., GDD) be better than means? Also, heat sums might simplify the analyses.

Line 144: "Creatively" is not a word you generally want to read in the methods section of a scientific paper. I recommend you simply delete this word.

Line 201: Did you consider computing heat sums instead of means?

Line 221 (table S6): typo at 1981 length: 35 days not 350 days

Line 224: For this figure and S13, S14, S15, add some thin grey grid lines to help readers track Temperature Value 1. This looks like ggplot2 to me. If so, you can add grid lines with: `theme(panel.grid =`

element_line()). You will probably want to make sure that your white geom_tiles are actually filled (not blank) so that they cover the grid lines.

Line 231: This is a helpful graphical representation of your process. I think you still need to show us more, however. Really we need to see some time series plots of the temperature series that you generated. Maybe at least some of the series that have significant pos and negative correlations with your population chronologies?

Line 238: This figure is a great opportunity to show some actual correlation coefficients between chronologies and climate series - which do not appear anywhere in your main text or in the supplement. You could keep the red and blue themes but use white-blue and white-red gradients to express the values of negative and positive correlation coefficients.

Reviewer #2 (Remarks to the Author):

Summary: This manuscript presents a new method for assessing temperature sensitivity of tree growth with temporally flexible temperature variables. They apply this method to two species of Eurasian boreal larch and use it to determine which populations and proportions of their ranges are expected to benefit from warming and are threatened by warming. Although I feel the authors need to clarify some terminology and further discuss this new method, I think this is an important advancement in the field. The methodology is mostly sound (though I have a few concerns I would like addressed) and with a few small tweaks I think the authors do a good job interpreting and discussing their results. I look forward to seeing this method further refined and integrated into statistical software to make it more widely available.

Major comments:

I think the t-linked method is novel and very helpful for moving towards more physiologically relevant climate indicators. The authors did an excellent job explaining the method and I appreciated the additional detail and figures in the supplement. However, how do we think about the potential for spurious correlations from running so many correlation analyses? With monthly values, this is already concerning (though partly accounted for in some approaches). How should we think about it here and what concerns should we have?

How do these t-linked estimates compare to other estimates of growing season length or heat accumulation rather than merely monthly temperatures? Some analysis of how T-linked estimates improve upon heating degree days or growing degree days, for example, seems like it would clarify how much better this method is than other methods that attempt to avoid calendar months.

I am concerned (and a little unclear) about what “warming threatened” means and how it should be interpreted. As I understand it, warming threatened means that at some point during the growing season, you identified a cluster of t-linked values that were correlated with declines in detrended ring widths regardless of how small the effect was or the nature of that cluster of t-linked values (spring, summer, long, or short). Is this correct? I think this should be made more clear in the main text. Related to this, if that is at least close to correct, the manuscript reads as if “warming threatened” (which really means growth declines somewhat with warm temperatures) is driving a perception of mortality or unsuitable habitat when looking at the maps and projections. Also, your work is on detrended growth which is often only capturing about 20% of the actual width of the ring. I suggest further clarifying (or redefining) what it means to be warming threatened and potentially setting a higher bar for what constitutes as “threatened” or just avoiding that term and sticking more to the data about correlations and growth.

I am a little concerned about the use of area in some of these figures (ex figures 5 and 6) when we know the distribution of larch is not continuous across this area. Are there no good models available for larch realized distributions within this space that you can rely on? Or at the very least, compare with your area-based estimates?

Minor comments:

Line 37 and 41: I am not sure “expand” and “retreat” are the best terms here. This phrasing invokes the population moving instead of existing populations having a change in their growth-climate relationships.

Line 139 “Orange shadow”

Figure 1. Panel D seems to have site ID labels?

Figures often have scientific names but text refers to common names. Consider revising for consistency and readability.

Line 229: Is “high moisture deficits” the correct way to phrase this? Isn’t it really showing the proportional area that is more generally threatened by warming? And we presume it is moisture deficit that is driving it?

Figure 5. The caption seems to say the green represents the warming threatened regions but it actually looks like the opposite? If the scale is the same as Figure 6, just add that scale.

Figures 5 and 6 are pretty repetitive and I think show more detail than is needed for the main text. It is hard to pull out interesting patterns other than “decreasing proportion of green” and “a lot vs. a little” between the different scenarios. Consider reducing the data presented here and perhaps combining figures 5 and 6.

Line 310-312: rephrase. Are you arguing they are more ecologically relevant than temperate forests?

Line 345: clarify where this idea of monolithic response comes from and what you mean? I think instead your work points to a more intuitive and physiologically relevant response (once you better articulate the physiological relevance, see below).

Line 399: this paragraph could be removed or should provide more concrete next steps for “complimenting radial growth” and relevant citations. For example, what type of models are needed? Here you use detrended radial growth – is that the way forward or do we need process based models etc?

Line 420-424: I would argue your method as implications beyond just boreal larch populations and think you should broaden this statement here.

Line 454-457: unclear how these met station data were used to generate climate data for each population. I see it later on line 533 but a half sentence here would help too.

Line 557-567: It is also quite common to have lagged effects: climate from one year impacting growth the following year. Why was that not considered in this study?

Line 557-564: Is there no effect of cold temperatures that should be considered?

The use of red and blue seems counterintuitive to me. Often red indicates threatened situations but here you use them opposite. Perhaps a different color scheme makes more sense?

At many points you refer to the t-linked method as “physiologically-informed” however that physiology was never discussed beyond just leaf phenology and short growing seasons. Is that all you mean? If so, this phrase seems to be a bit of a stretch.

With the development of this new t-linked approach, why are you not also considering a more robust precipitation metric? There are obviously water balance and drought indices that do this but could your new method provide an alternative way to think about this?

Reviewer #3 (Remarks to the Author):

Summary

This manuscript focused on the boreal larch forests of Eurasia. In these forests, the relationships between tree growth and climate were studied. To carry out this analysis, the authors developed a new method for correlating tree growth with growing season temperature. These new correlations revealed that a much higher proportion of larch forests may be responding negatively to growing season temperature than was previously thought. Using the observed correlations, the authors projected how larch forest tree growth would respond to future climate change. A main conclusion was that increased temperatures pose a significant and overlooked risk to Eurasian larch forests.

I think this is an important study with potentially strong impact. My opinion is that larch forests are probably understudied, and this work should stimulate interest and inspire new work. The T-based method is novel and interesting, though I think it requires better motivation and more validation, as described below.

Motivation for the Study

I reacted with skepticism to the argument in ll. 99-120, which is one of the major motivations for this study. It would help if the authors could do more to convince me that “the same calendar period can correspond to very different growth phases from year to year”. What calendar period is typically used? What growth phases are the authors talking about? Can the authors provide a specific example to illustrate the point?

The authors justify their study, in part, with the sentence on ll. 67-69: “The wide distributions would imply high spatial heterogeneity of growth responses...”. More justification for this statement would be helpful. The authors do cite a book, but I don’t have access to that book, so it is difficult for me to evaluate the authors’ claim.

Further justification for the study comes in ll. 91-96, where the authors cite a contrast between tree ring and satellite data. I think this contrast is overplayed. The tree ring datasets provide information on tree diameter growth. The satellite data definitely do NOT provide information on tree diameter growth. For example, Berner and Goetz study vegetation greenness. Thus, I don't see how direct comparisons and contrasts are possible.

Validation and Assessment of Results

Taking the results at face value, the T-linked approach seems to be an important advance over the conventional calendar-linked approach. However, I do worry about the possibility of spurious correlations. As I understand it, the T-linked approach involves the determination of two dates based on two temperature values (ll. 487-489). The temperature in this correlation window is then correlated to growth, and statistical significance is assessed. This procedure is carried out many times, for many correlation windows. Because many correlation windows are tested, it seems inevitable to me that false positives will creep in to the analysis. To make a more convincing case that the analysis is robust, the authors might consider doing some cross-validation or other sort of out-of-sample testing. I think that such out-of-sample testing is especially important given that the authors are extrapolating their correlations to projected climates (Fig. 5).

Another issue with the analysis pertains to the effect sizes. The Results showed the fraction of trees and populations with either a statistically significant temperature sensitivity (Table 1). However, I did not see the authors report the actual effect sizes anywhere. This absence is important. One could get statistically significant correlations ($p < 0.05$) even if the correlation coefficient itself is very small. Thus, there is the possibility of obtaining statistically significant but biologically meaningless correlations. To convince me this is not the case, it would be helpful to provide information on the correlation coefficients obtained from correlations between tree diameter growth and correlation window temperature.

Presentation

Lines 121-122 strike me as an unhelpful generalization. What specifically do the authors mean? Are there references to back up this point?

I struggled with the data presented in Table 1 and the related discussion in the text. For example, the first line in Table 1 lists the "Sensitive chron. pct." as 57.6%. I further expected that all trees showing sensitivity would either have a positive sensitivity or a negative sensitivity. That is, I expected that "Sensitive chron. pct." = "Positive sensitive chron pct" + "Negative sensitive chron pct". But this is clearly not the case. The authors should better explain what is going on here.

Figure 2b: The various p-thresholds are confusing. Why not choose a single threshold? Or, even better, report the exact p value in each case.

Comparison of the 1960-1990 and 1970-2000 results is not that compelling because those two time periods mostly overlap. Why not compare 1960-1980 and 1980-2000?

Mechanisms and Caveats

The Discussion section of the paper struck me as narrow. I felt like the authors could have done more to discuss the mechanisms for the changes and the caveats to their analysis. For example, this study analyzed MAP but not VPD. I think this choice could be controversial. A number of recent studies argue for the importance of VPD in controlling boreal tree growth and mortality. Also, the correlations between temperature and tree growth may now be controlled (or could possibly be controlled in the future) by many mechanisms, like permafrost loss, increases in nitrogen mineralization, etc. I was surprised to see no discussion of these points. Neither was there much discussion of the mechanisms behind the observed north-south variation in the correlations. How might these spatial gradients be impacted by gradients in stand density, soils, etc.? Also, the climate projections were presented with

few caveats. It is not at all clear to me that climate models are able to correctly simulate the observed interannual variation in boreal Eurasian climate, which is the underlying motivation for this study.

REVIEWER COMMENTS

Reviewer #1 (Remarks to the Author):

Dear authors,

I read your manuscript submission to Nature Communications, "Reassessment of growth-climate relations indicates the potential for decline over Eurasian boreal larch forests," with great interest. The manuscript describes a spatially extensive and sophisticated investigation into the growth-climate relationships of boreal Eurasian larch (2 species) using an insightful and novel method to aggregate climate data. Rather than rely on monthly aggregates of climate data (which are of fixed length and period year-to-year), the authors define potentially physiologically relevant periods using daily temperature thresholds (which may vary in length and period year-to-year). The authors propose that by using this method, they can obtain much more robust and accurate estimates of growth-temperature relationships. Indeed, using >8,500 larch tree-ring series, they found that many more trees and populations show significant correlations with their temperature time series (>90%) vs. the monthly aggregated series (~50-60%) in each of the climatic periods they investigated (1960-1990 & 1970-2000). In order to gain insight into possible mechanisms driving the growth-temperature relationships, the authors next described the climate spaces, in terms of mean annual temperature and mean annual precipitation, which were more associated with positive and negative growth-temperature relationships. Finally, the authors use the results of this last analysis and climate projections from a suite of GCMs to predict how areas suitable for or detrimental to larch growth may shift over their study region in the coming century. The authors clearly did an impressive amount of work.

Thank you for your comments. We truly appreciate them.

I think that this study has the potential to greatly influence future work on tree growth-climate relationships globally. Tree rings are unmatched in ecological science for their temporal extent, resolution, and accuracy and therefore represent one of the best tools we have to answer critical questions about how trees and forests have responded and will respond to changes in climate. Tree rings themselves and assessing their relationship to climate variables present several challenges. This paper represents a major advance in the latter challenge.

There are, however, several issues with the paper in its current form that I believe require major revisions before it is suitable for publication in Nature Communications. Given the obvious level of skill and sophistication that the authors possess, I believe all of my suggestions are achievable. The first major issue is that the method they used to process (detrending & standardization) their tree ring data is inappropriate for their goals. Some percentage of the Pearson correlations with climate will be invalid as a result. The suggestions I made should fix these issues. The choice of tree ring detrending methods can be enormously consequential and can confound subsequent analyses. I think the authors have not adequately considered these pitfalls, instead relying on software default methods. The second major issue I have is with the formal analysis to project areas that will be suitable for or detrimental to larch growth ("warming-benefitted" and "warming-

threatened”) based on their results and GCM predictions. This analysis consumes a major portion of the manuscript, yet it appears to have a weak premise: that the nature of growth-temperature relationships during 1970-2000 will carry forward through 2100. The authors themselves report that there were changes in the proportions of trees recording positive and negative correlations with climate from 1960-1990 to 1970-2000. I think that devoting so much of the manuscript to an analysis that relies on extrapolation (and could very likely be wrong) risks turning a potentially very impactful piece of work into a slick but disposable article.

Instead, I suggest that the authors delete the projection analysis and instead focus on improved explanation of their novel t-linked method (including better tests of its efficacy) and a more clear development/description of their 4 growth-climate metrics (replacing or adding figures too). I think they should retain their analysis of the climate space associated with positive and negative responses to temperature - this is very useful in providing insights as to what the underlying mechanisms are that produce the varying responses to climate. I think you should expand on these mechanisms/hypotheses in the discussion. In addition, a very informative additional set of analyses would be to consider how their conclusions about the mechanisms driving growth-climate relationships would have been different for traditional rigid monthly climate time series vs their more sophisticated time series.

I catalogue the bulk of my detailed suggestions and critiques in the line-specific comments below.

We sincerely thank the reviewer for the constructive and detailed comments for our manuscript, also for the recognition of our work. The reviewer raised several major concerns and a series of specific comments, which are very helpful for improving our work. We have substantially revised our manuscript to address the reviewer’s concerns and made point-by-point responses to the reviewer’s specific comments as below.

Overall, the reviewer’s primary concerns were 1) detrending and standardization of tree-ring data, 2) validation of the T-linked method, 3) uncertainties of extrapolating the models to climate projections. These are excellent points. As you will see below, we did extensive re-analyses based on your suggestions.

To address point 1), we reprocessed the raw tree-ring data following the reviewer’s guidance, and updated all subsequent results. On this basis, we also compared the main results calculated based on the tree-ring data detrended using different methods. We found subtle but no essential changes in our main results regardless of the detrending method. We speculate that this might be related to the fact that we only used fragments of the chronologies in the analysis and avoid much of the end effects issue, which is very real. For most of our data, the middle sections of the growth series we tested here were relatively stable no matter the detrending method. As a result, the set of findings from the reviewer’s suggestions are in line with those from our original set of detrending methods.

To address point 2), we performed additional analyses to test the effectiveness and advantages of the T-linked method over the calendar-based methods under the condition of computing correlations of the same order of magnitude as suggested. We also conducted 10-fold cross-

validation on the logistic models estimated based on the correlation results using T-linked method to test the robust of the models. To address point 3), we have made significant reduction in this part and marked the area where the projection is relatively unreliable.

In response to the major concerns and other detailed comments, we have performed additional analyses and substantially revised the manuscript. These supplementary analyzes and corresponding results have been described and provided in detail below. Overall, the reviewer's comments led to a much-improved manuscript, thank you for that again.

Please find below our point-by-point responses to each of the reviewer's comments.

Line-specific comments for the main text:

Line 2: If you keep this title, it should read "across Eurasian..." rather than "over Eurasian..."

-- Thank you for your suggestion. We have revised the title as suggested.

Line 24: The abstract is poorly-written and difficult to interpret. I request that you substantially revise it in your revisions.

For the most part the other parts of the paper are very well written.

-- We have rewritten the abstract as suggested by the reviewer, absorbing the reviewer's follow-up comments on the abstract, and shortened it to 150 words to meet the journal's requirement. The new revised text reads:

Lines 24-35, Abstract section: "Larch, a widely distributed tree in boreal Eurasia, is experiencing rapid warming across much of its distribution. A comprehensive assessment of growth to warming is needed to comprehend potential impact of climate change. Most studies, relying on rigid calendar-based temperature series, have detected monotonic responses at margins of boreal Eurasia, but not across the region. We developed a novel method for constructing temporally-flexible and physiologically-relevant temperature series to reassess growth-temperature relations of larch across boreal Eurasia. Our method appears more effective in assessing the impact of warming to growth. Our approach indicates a widespread and spatially heterogeneous growth-temperature responses that is driven by local climate. Models quantifying these results project that the negative responses of growth to temperature will spread northward and upward throughout this century. If true, the risks of warming to boreal Eurasia could be more widespread than conveyed from previous works."

Lines 25-26: Unclear where this 30% figure comes from. It is not cited in the introduction. Is it derived from the Isaev et al 1995 paper? Are there not more recent efforts to estimate species or genus-specific cover? Either clearly cite this in the intro or walk the reader through (in a sentence) how you derived it.

-- This percentage was calculated indirectly based on the following two points: (i) "Globally, the

boreal zone covers about 1.890 billion ha in the northern hemisphere, 60% in Russia, 28% in Canada...” (Brandt et al., 2013). (ii) “Larch forest covers the largest area (42.4% of total area of Russian forests)” (Isaev et al., 1995). Therefore, the larch forests in Russia roughly represent 25.44% of global boreal forests, plus the larch forests distributed in Kazakhstan, Mongolia, and Northwest and Northeast China, we estimated that Eurasian boreal larch trees could represent 25-30% (~30%) of global boreal forests. Considering errors and uncertainties in the calculation, we have deleted this percentage in the revision.

References:

- Brandt, J.P., Flannigan, M.D., Maynard, D.G., Thompson, I.D. and Volney, W.J.A., 2013. An introduction to Canada’s boreal zone: ecosystem processes, health, sustainability, and environmental issues. *Environmental Reviews*, 21(4): 207-226.
- Isaev, A., Korovin, G., Zamolodchikov, D., Utkin, A. and Pryaznikov, A., 1995. Carbon stock and deposition in phytomass of the Russian forests. *Water Air Soil Poll*, 82(1-2): 247-256.

Lines 28-30: Revise this sentence for clarity. What do you mean by universal? Positive responses to warming? Negative responses to warming? Both? If both, then the responses are not universal. Also, since you have mentioned larch already I presume that the previous studies are on larch growth-climate relationships specifically? Perhaps you can move the larch focus to after this sentence - I think most growth-climate relationships in trees globally (not just in boreal larch) use monthly data.

Maybe something like this would be more clear:

“Most studies thus far have relied on monthly-aggregated climate data to assess tree growth relationships with climate, detecting monotonic responses to warming at boreal forest margins but no clear responses in vast interior regions. However, such rigid calendar-based aggregates may not adequately capture physiologically-relevant periods that vary interannually.”

-- Thank you for your suggestion. We have modified the text in the revision as suggested, the new text reads: **(Lines 26-28)** “Most studies, relying on rigid calendar-based temperature series, have detected monotonic responses at margins of boreal Eurasia, but not across the region.” We have also rewritten the abstract, please see the revised abstract above.

Lines 32-33: You should say “individual tree-ring series.” Some of those series represent radii from the same trees. Chronologies refer to aggregates of many tree series.

Also, a more clear and accurate sentence would be:

“...8,566 tree-ring series aggregated to 260 population chronologies.” Either way, wow! Very impressive sample size.

-- Thank you. We have modified the text as suggested, please see the revised abstract above.

Line 34: “across the distribution of boreal Eurasian larch”

-- Thank you. We have corrected the text and rewritten the abstract, please see the revised abstract above.

Line 35-37: I don't know what this sentence means. Please revise for clarity.

-- Thank you. We have rewritten the abstract, the sentence has been deleted, please see the revised abstract above.

Line 48: I am thankful that you kept the more standard in-text citation style for this stage of submission - it makes it easier to keep track of where your ideas come from. I'm guessing the editor will want the Nature numbered style in the next round of revision, however. [Editor's note: since the reviewer finds it helpful, for the time being you may keep the current reference style]

-- Thank you. We keep the reference style for your convenience in the revision.

Line 65: Great maps!

-- Thank you.

Lines 91-96: One explanation for a mis-match between ITRDB-based analyses and satellite-based ones is that ITRDB data do not derive from any kind of objective sampling design. In fact, ITRDB records may have been selected in attempt to maximize certain climate signals for the purpose of dendroclimatological reconstructions. This means that ITRDB records may not be suitable for making general inferences across regions. I don't see that you have really grappled with this potential issue in your paper.

See (DeRose et al., 2017, Building the forest inventory and analysis tree-ring data set, Journal of Forestry 115(4), pp283-291) for a detailed discussion of this issue.

It seems to me that at minimum you would need to know something about how these data were collected (ITRDB links the papers) and filter out those that were not appropriate to your study. Alternatively you could limit your inferences based on lack of knowledge of what kind of sample design your data comes from.

-- We agree with the reviewer that the ITRDB data may not be fully representative due to the potential sampling bias, this issue also has been discussed in many previous studies. As suggested by the reviewer, we reviewed the sampling design and original collection purposes of the tree-ring data we used from the information provided by the contributors to ITRDB or the related published papers.

The data we used in this study were collected mainly for two initial purposes, climate reconstructions and investigating growth-climate relations, and most of these data were collected following the classic dendrochronological sampling design that only the selected dominant and undamaged trees (15-30 individuals generally, or more) were sampled to increase the climate signals. Such sampling design may lead to poor representativeness and overestimations of forest productivity and climate change impacts on trees (Babst et al., 2018; DeRose et al., 2017; Klesse, et al., 2018; Zhao, et al., 2018). Also, studies for climate reconstruction are usually conducted in

marginal habitats, which may introduce a ‘macro-site selection’ bias in tree-ring networks (Klesse, et al., 2018).

Despite all these, Nehrbass-Ahles et al. (2014) compared diverse sampling designs and found that investigating climate-growth relationships is largely unaffected by the sampling design and robust to the classic dendrochronological sampling design. Also, in previous studies, the climate sensitivity bias is not found to exist when comparing ITRDB data with other independent unbiased reference networks (Klesse et al., 2018; Babst et al., 2019). Besides, ITRDB data, including the part we used, have been widely used to explore continental- or even global-scale ecological issues (Gao et al., 2022; Zuidema et al., 2022), having demonstrated its reliability and robustness. Moreover, our tree-ring network evenly covers both the distribution areas and climatic spaces of the two larch species, and shows similar climatic probability density distributions with the whole species distribution areas (**Supplementary Fig. 2**, see below), suggesting good representativeness. Taken all, we consider our tree-ring network suitable for our current analyses and research purposes.

Nevertheless, we totally agree that the best way to address this is to conduct extensive unbiased sampling across Eurasian boreal forests to validate the representativeness of the existing tree-ring data (Babst et al., 2018; Zhao et al., 2019). Unfortunately, that is not feasible at the present time with the most comprehensive public data. As a precaution, we introduced gradient probability thresholds (0.5, 0.75 and 0.95) in logistic model to determine whether a larch population shows negative growth-temperature responses instead of using only a 0.5 in most cases using logistic model as classifier. The intent here is to accommodate the potential overestimation of the temperature effects by the ITRDB data.

We provided the sampling design of each sampling population in the supplementary information (**Appendix A**), replotted Supplementary Figure 2 (see below) to add the climatic probability density distributions of our tree-ring network and the larch distribution areas, and added caveats in the Methods and Discussion sections as seen below:

Lines 474-489, Methods section: *“The ITRDB data and the published chronologies compiled into our network were collected or established primarily for the original purposes of dendroclimatic reconstructions and investigations of growth-climate responses (Appendix A). Therefore, most of these data were collected following the classic dendrochronological sampling design that increases the climate signals (Appendix A). Such sampling bias could make the network oversensitive to climate, thus creating the climate sensitivity bias (Babst et al., 2018; Klesse et al., 2018b). Integrating and cross-validating the existing tree-ring data with extensive unbiased sampling is the most effective way to avoid the potential bias (Babst et al., 2018; Zhao et al., 2019), while is currently not feasible for boreal Eurasia. However, Nehrbass-Ahles et al. (2014) report that investigating growth-climate relationships is largely unaffected by the sampling design. The climate sensitivity bias is also found not to exist when comparing ITRDB data with other independent unbiased reference networks (Babst et al., 2019; Klesse et al., 2018a). Moreover, our tree-ring network evenly covers both the distribution areas and climatic spaces of the two larch species, showing similar climatic probability density distributions to the species distribution areas (Fig. 1; Supplementary Fig. 2). Given these reasons, we consider our tree-ring network to be suitable for*

our current analyses and research purposes.”

Lines 673-675, Methods section: “To prevent the estimations from depending on our choice of probability threshold and to accommodate the potential climate sensitivity bias of our tree-ring network, we compared the results of using three gradient thresholds ($P_0 = 0.50, 0.75, \text{ and } 0.95$) to identify the two types of regions for both baseline and future periods.”

Lines 400-405, Discussion section: “The existing tree-ring network also could be oversensitive to climate due to the current sampling bias (Babst et al., 2018; Klesse et al., 2018b), though see Nehrbass-Ahles et al. (2014) as a counterpoint that it might not be an overestimate. Such lack of spatial representation and potential climate sensitivity bias may result in contrasting response patterns detected with the actual situation.”

Supplementary Fig. 2 | Climate spaces occupied by Siberian larch (a) and Dahurian larch (b) during 1970-2000. MAT and MAP represent mean annual temperature and precipitation, respectively. Shadows represent the climate spaces occupied by the two larch species. Gradient color represents the cell density. Straight dashed lines represent the averages of climate parameters, and slant lines represent the fitted lines between MAT and MAP using simple least square regression. Correlation coefficients (Pearson's r) between MAT and MAP are noted on each panel. Sampling populations were located in the climate space based on their long-term local climate conditions. Probability distributions of the climate conditions (MAT and MAP) across the species distributions (red line) and of the sampling populations (black line) estimated by kernel density were provided.

References:

- Gao, S. et al., 2022. An earlier start of the thermal growing season enhances tree growth in cold humid areas but not in dry areas. *Nature Ecology & Evolution*, 6(4): 397-404.
- Babst, F. et al., 2018. When tree rings go global: Challenges and opportunities for retro- and prospective insight. *Quaternary Science Reviews*, 197: 1-20.
- Babst, F. et al., 2019. Twentieth century redistribution in climatic drivers of global tree growth. *Science Advances*, 5(1): eaat4313.

- Dannenber, M.P., Wise, E.K. and Smith, W.K., 2019. Reduced tree growth in the semiarid United States due to asymmetric responses to intensifying precipitation extremes. *Science Advances*, 5(10): eaaw0667.
- DeRose, R.J., Shaw, J.D. and Long, J.N., 2016. Building the Forest Inventory and Analysis Tree-Ring Data Set. *Journal of Forestry*, 115(4): 283-291.
- Klesse, S. et al., 2018. A combined tree ring and vegetation model assessment of European forest growth sensitivity to interannual climate variability. *Global Biogeochemical Cycles*, 32(8): 1226-1240.
- Klesse, S. et al., 2018. Sampling bias overestimates climate change impacts on forest growth in the southwestern United States. *Nature Communications*, 9(1): 5336.
- Nehrbass-Ahles, C. et al., 2014. The influence of sampling design on tree-ring-based quantification of forest growth. *Global Change Biology*, 20(9): 2867-85.
- Zuidema, P.A. et al., 2022. Tropical tree growth driven by dry-season climate variability. *Nature Geoscience*, 15(4): 269-276.

Lines 105-106: This is a really great point and very compelling!

-- Thank you.

Lines 108-110: "High intra- and inter-annual..." This sentence is unclear, please revise.

-- Thank you for your comment. We have deleted this sentence in the revision.

Lines 125-126: You might describe these in more detail here if you can fit in the text. I think you need to describe them in more detail elsewhere (current explanations are not adequate).

-- Thank you for your suggestion. We have rewritten the descriptions of the correlation-derived metrics in a more detailed manner as a separate subsection in the Methods section of the main text. The new text reads:

Lines 575-602, Methods section:

"We calculated the mean coefficient of all significant correlations (Pearson's r , $p < 0.05$) between each population chronology and the T-linked temperature series from the nearest meteorological station. We further established four metrics based on the tree- and population-level growth-temperature correlation results to measure the size of temperature effects on tree growth and to characterize the structure of growth-temperature response pattern for each population from two dimensions: (i) affecting scope, i.e., what is the proportion of tree individuals in a population that show a significant relation to temperature; and (ii) affecting duration, i.e., how long the growth response to temperature lasts. Each metric was calculated separately from negative and positive growth-temperature correlations to characterize negative and positive temperature sensitivity of each population, respectively.

As the time period used to construct the same T-linked temperature series varies inter-annually, we calculated the mean length (in the number of days) of the varying T-linked periods over corresponding analysis period for each T-linked temperature series. We also calculated the

percentage of single growth series (individual tree chronologies) in each population showing significant negative or positive correlations with each T-linked temperature series from the nearest meteorological station. The maximum value of these percentages was used as the first metric, that is the maximum proportion of individuals in a population that show negative or positive sensitivity to temperature (MST, %), representing the maximum scope of negative or positive temperature effects on each population. We then ranked all T-linked temperature series correlated with the same population in descending order of the percentage of single growth series (individual tree chronologies) showing significant negative or positive correlation with them, and averaged the mean lengths of the temperature series ranked top 5% and 50% as metrics *tDur05* and *tDur50* (days), respectively. Another metric was derived from the population-level correlations, corresponding to the average mean lengths of all T-linked temperature series with which the population chronology was significantly negatively or positively correlated (*pDur*, days). We suspect that the last three metrics can be interpreted as a way of describing the mean duration of negative or positive growth responses to temperature for a population.”

Line 133: This is misleading because you trimmed down this dataset to 8,566 series. Really, you should report the number of trees, not measurement series. Trees and populations are your units of analysis, correct?

-- Thank you. We have reported the total number of trees in our tree-ring network in the revision as suggested by the reviewer. It is not easy to correctly count the number of sampling trees for each sampling population from the ITRDB because the contributors have adopted complex and highly variable, if not custom, naming conventions for their core samples and trees. Most of these data followed standard naming conventions, like SSTTC (site-tree-core), which can be deciphered accurately, while for another small part (about 20%), we cannot ensure the accurate understanding for their various naming conventions. Frustrating for us, these conventions are not provided to users of the data for many, if not most, data sets.

Nevertheless, under the premise of ensuring the accuracy as much as possible, we have completed the identification and statistics of the sampled trees for the 193 ITRDB sampling populations. We have added this information to Appendix A. We then averaged the standardized single growth series from the same tree into individual tree chronologies.

Hence, we conducted correlation analyses on both the single standardized growth series and the individual tree chronologies and tested the differences between the correlation results. The proportions of single growth series and individual trees shows that the significant positive or negative correlations with temperature in each sampling population were highly consistent (Supplementary Table 4, also see below). The metrics calculated separately based on the correlation results of single standardized growth series and individual tree chronologies with temperature were also highly consistent (Supplementary Fig. 13, also see below). The consistency of these results suggests that single growth series can be a good substitute for individual tree chronology to represent tree-level radial growth. Therefore, in the revised Main text, we used single growth series to represent growth for relevant analyses, and provided the results calculated from individual tree chronologies in Supplementary Information Part C for reference.

Lines 461-469, Methods section: “Our identification of sampling trees for some ITRDB data is speculative due to the complex and various naming conventions the contributors adopted. Therefore, we compared the correlations of the single standardized growth series and individual tree chronologies with temperature, as well as the correlation-derived metrics. The consistency of these results (Supplementary Fig. 12, Supplementary Table. 4) suggests that the single growth series can be a good substitute for individual tree chronology to represent tree-level radial growth. In the Main text, we used single growth series to represent tree-level radial growth for relevant analyses, and provided the correlation results calculated from individual tree chronologies in Supplementary Part C for reference.”

Supplementary Table 4| Comparisons of the correlation results of the single standardized growth series and the individual tree chronology with temperature

Species	Time period	Total number	Method*	Sensitive pct. (%)	Negative sensitive pct. (%)	Positive sensitive pct. (%)
Single standardized growth series						
Larix sibirica	1960-1990	2636	TL	2548 (96.7%)	1448 (54.9%)	2312 (87.7%)
			CB	2047 (77.7%)	714 (27.1%)	1675 (63.5%)
			CT	2028 (76.9%)	610 (23.1%)	1641 (62.3%)
	1970-2000	2053	TL	1982 (96.5%)	1432 (69.8%)	1568 (76.4%)
			CB	1424 (69.4%)	809 (39.4%)	835 (40.7%)
			CT	1427 (69.5%)	806 (39.3%)	759 (37.0%)
Larix gmelinii	1960-1990	3508	TL	3361 (95.8%)	2073 (59.1%)	3012 (85.9%)
			CB	2547 (72.6%)	1089 (31.0%)	2082 (59.4%)
			CT	2666 (76.0%)	864 (24.6%)	2189 (62.4%)
	1970-2000	2130	TL	2084 (97.8%)	1434 (67.3%)	1829 (85.9%)
			CB	1393 (65.4%)	566 (26.6%)	1090 (51.2%)
			CT	1575 (73.9%)	699 (32.8%)	1107 (52.0%)
Individual tree chronology						
Larix sibirica	1960-1990	1823	TL	1754 (96.2%)	972 (53.3%)	1610 (88.3%)
			CB	1451 (79.6%)	482 (26.4%)	1214 (66.6%)
			CT	1444 (79.2%)	393 (21.6%)	1202 (65.9%)
	1970-2000	1452	TL	1399 (96.3%)	948 (65.3%)	1150 (79.2%)
			CB	1040 (71.6%)	559 (38.5%)	666 (45.9%)
			CT	1033 (71.1%)	529 (36.4%)	609 (41.9%)
Larix gmelinii	1960-1990	1815	TL	1759 (96.9%)	1033 (55.8%)	1597 (88.0%)
			CB	1394 (76.8%)	582 (32.1%)	1181 (65.1%)
			CT	1448 (79.8%)	429 (23.6%)	1225 (67.5%)
	1970-2000	994	TL	976 (98.2%)	617 (62.1%)	863 (86.8%)
			CB	679 (68.3%)	272 (27.4%)	563 (56.6%)
			CT	772 (77.7%)	304 (30.6%)	591 (59.5%)

* TL: T-linked method; CB: moving calendar-based method; CT: Calendar-based T-linked method.

Supplementary Fig. 13| Comparisons of the metrics derived from the correlation results of the single standardized growth series and the individual tree chronology with temperature. The plot shows the correlation-derived metrics (MST, tDur05, and tDur50 in rows) and the temperature sensitivity signs (negative and positive in columns). The x-axis and y-axis in each panel represent the metrics calculated based on single standardized growth series (core) and individual tree chronology (tree), respectively, and the Pearson correlation coefficient between them is noted on each panel. Red and blue points represent sampling populations of Siberian larch and Dahurian larch, respectively. MST represents the maximum scope of temperature effects on populations, which is described by the maximum proportion of negative or positive temperature-sensitive single growth series (individual tree chronologies) in a population. tDur05/tDur50 represents the duration of tree-level growth-temperature response, which is described by the average mean number of days of the top 5%/50% T-linked temperature series in descending order of the proportion of single growth series (individual tree chronologies) in the population that are significantly negatively or positively correlated with them.

Line 139: Orange, not organ?

-- Thank you. We have corrected the text in Line 138.

Line 141: The grey squares are hard to see. Maybe black-outlined squares with white fill would be more clear?

-- Thank you for your suggestion. We have replotted this figure as suggested by the reviewer.

Lines 146-147: You need to report what you did with the monthly series somewhere in the main text or in the supplement - I didn't see any description of your methods for the monthly data anywhere. What was the data source (ideal comparison would be to aggregate your daily station data to monthly means)? What correlations did you perform, just individual months or aggregates of 2-12 months? I hope you did both.

The key here is that the comparison is as fair as possible - this is a central question in your study and we can't take it on faith that the comparison was set up in the best way possible.

It is also important to keep in mind that there are a few plausible reasons why you would find many more sig. correlations with your method vs. monthly data (assuming you treated the monthly data as I have described above). One is your conclusion that it is due to capturing the year-to-year variability in relevant temperature periods. This makes sense to me too. However, another explanation is that you have simply computed many more discrete intra-annual periods: something on the order of 3000 possible periods within a year for your *t*-linked method (do I have that \pm right?) vs. ~78 possible contiguous month combinations. Given these ballpark numbers alone, it is no surprise that you would find more sig correlations. A real test of your hypothesis that more sig correlations was due to capturing the same temperature period in each year (regardless of what dates it covers) would be to compute fixed year-to-year periods using the daily data. You could constrain the rigid daily periods to ≥ 10 days, etc. to match your variable *t*-linked periods (e.g., 5 April to 10 May for all years 1970-2000). This would generate a number of possible series that is in the same order of magnitude as your *t*-linked series. My guess is that year-to-year rigid daily series will yield more significant correlations than with rigid monthly data. If you are correct that the year-to-year flexibility is the key, then you should still get more sig correlations with the *t*-linked series.

-- Thank you for your suggestion. We understand the concern related with the high number of correlations. As the reviewer said, the effectiveness and advantage of the *T*-linked method over the calendar-based method have not been tested and verified under the condition of computing correlations of the same order of magnitude. We have supplemented this part of the analysis in the

revision as suggested to further demonstrate that the significant growth-temperature correlations detected by our T-linked method were not simply due to more correlation calculations. We used two different methods to construct the calendar-based temperature series, described as below or in Lines 615-636 in the revision:

(i) We constructed the calendar-based temperature series using 36 time windows fixed to the calendar with variable widths from 15 days to 120 days at intervals of 3 days. To create calendar-based temperature series of the same order of magnitude as the T-linked temperature, we run each of these fixed windows over daily mean temperature series between April 1st (DOY 91) and September 30th (DOY 273), and then averaged the daily mean temperatures within each window year by year to construct the temperature series. As a result, each growth series was correlated with 2,106 calendar-based temperature series, a number that exceeds the average number T-linked temperature series correlated with each growth series, which is 1,548. The commonly-used seasonal, bimonthly, monthly, and semi-monthly temperature series were all included in the analysis through this way. The results showed that although much more calendar-based temperature series were used in correlation analysis, we still detected significantly more temperature-sensitive growth series using the T-linked method (Lines 144-169 and Table 1 as below).

(ii) We averaged the interannually varying start and end dates of the T-linked periods used to construct each T-linked temperature series and then averaged the daily mean temperatures between the two mean dates year by year to construct a calendar-based T-linked temperature series. Each growth series was then correlated with all the calendar-based T-linked temperature series that corresponded on a one-to-one basis to the normal T-linked temperature series. Similarly, temperature-sensitive growth series detected using the calendar-based T-linked temperature series were significantly less than those detected using the normal T-linked temperature series (Lines 144-169 and Table 1 as below).

In addition to the above supplementary analyses, we also conducted cross-validation analysis on the models established based on the correlation results using T-linked method. We performed a 10-fold cross-validation to test the quality and accuracy of the estimated logistic models (Hastie et al., 2001) (Lines 657-665). The observations for parameter estimation were divided into 10 subsets that were roughly of equal size. Nine of these subsets were used as training group to estimate a cross-validated model while the 10th subset was used for testing. The estimated error, which is used to determine classification accuracy for a logistic model, was calculated by applying the cross-validated model to the test group. This procedure was executed 10 times, with each subset taking turns as the test group. The resulting 10 classification accuracies were averaged as an estimate of the overall predictive performance of the final model. The results show that the logistic models we established for both larch species have fairly high cross-validation accuracy and quality (observed accuracy = 0.862 ± 0.064 , Cohen's $\kappa = 0.713 \pm 0.132$ for Siberian larch; observed accuracy = 0.885 ± 0.056 , Cohen's $\kappa = 0.706 \pm 0.131$ for Dahurian larch) (Lines 222-225). The results of 10-fold cross-validation indicated high robustness and accuracy of the models built based on the different subsets of the correlation results using T-linked method. We also found similarly strong validation of the estimated quantitative relationships between growth-temperature response pattern and local

climate conditions are robust and physiologically reasonable.

We hope that the reviewer's concern is addressed with the multiple levels of supplementary analyses that we conducted. We added the supplementary analyses as independent parts in the revision. We also discuss these points now in the discussion section, to clarify and make clear the best use of our method, and contextualize our results, as well as added a caveat in the Discussion section (**Lines 310-316**). The new text reads as below:

Lines 144-169, Results section:

“We compared the T-linked method with two methods that develop temperature series based on fixed calendar periods under the statistical constraint of calculating correlations of the same order of magnitude. Using a moving calendar-based method, each growth series was correlated with 2,106 calendar-based temperature series, including commonly-used seasonal and monthly temperature series. Through this way, ~70% of the sampling individuals and populations were identified as sensitive to temperature. Specifically, 29.3% (n = 1,803) and 32.9% (n = 1,375) of the single tree growth series showed significant negative responses to calendar-based temperature series from 1960-1990 and 1970-2000, respectively, while 61.1% (n = 3,757) and 46.0% (n = 1,925) responded positively (Table 1). Growth-temperature response signals detected using the calendar-based T-linked method (see Methods) were close to those detected using the general moving calendar-based method for both species and both analysis periods (Table 1).

By contrast, we found that >95% of the individual trees and populations were sensitive to temperature when using our more flexible T-linked temperature series approach. We found that the proportions of single tree growth series showing negative temperature sensitivity were 57.3% (n = 3,521) in 1960-1990 and 68.5% (n = 2,866) in 1970-2000, while the proportions of positive temperature-sensitive individuals were 86.7% (n = 5,324) and 81.2% (n = 3,397) (Table 1). Analysis at the population level showed the same set of trends. The percentage of populations that responded negatively was 46.9% (n = 113) in 1960-1990 and 63.9% (n = 101) in 1970-2000, while those responding positively were 82.2% (n = 198) and 72.2% (n = 114) (Table 1). These percentages were also consistent between species (Table 1). The overwhelming proportion of significant growth-temperature responses detected at both the tree and population levels using our new T-linked temperature series demonstrates the effectiveness of T-linked method and highlights the key role of temperature on larch growth across boreal Eurasia.”

Lines 222-225, Results section: *“Our 10-fold cross-validation analysis indicated that the estimated models were robust and of fairly-high accuracy and quality (Siberian larch: observed accuracy = 0.862 ± 0.064 , Cohen's $\kappa = 0.713 \pm 0.132$; Dahurian larch: observed accuracy = 0.885 ± 0.056 , Cohen's $\kappa = 0.706 \pm 0.131$).”*

Lines 310-316, Discussion section: *“While there are potentially serious issues with conducting such a large number of calculations, we tested the risk of potential spurious correlations using the T-linked method through a fair comparison to calendar-based methods and suggested that the effectiveness of our method was not attributed to the increased number of calculations. Furthermore, the cross-validation analyses we conducted demonstrated fairly-high accuracy and robustness of our estimated models, suggesting limited influences of spurious correlations on the*

main results.”

Lines 657-665, Methods section: “We tested the quality and accuracy of the estimated logistic models using 10-fold cross-validation analysis (Hastie et al., 2001). The observations for parameter estimation were divided into 10 subsets that were roughly of equal size. Nine of these subsets were used as training group to estimate a cross-validated model while the 10th subset was used for testing. We calculated the classification accuracy and the Cohen’s kappa (κ) of the cross-validated model applied to the test group to measure the classification performance. This procedure was executed 10 times, with each subset taking turns as the test group. The resulting 10 classification accuracies and 10 Cohen’s κ were averaged as estimates of the overall predictive performance of the final model.”

Table 1| Summary of growth series showing temperature sensitivity detected using different temperature series construction methods

Species	Time period	Total number	Method*	Sensitive pct. (%)	Negative sensitive pct. (%)	Positive sensitive pct. (%)
Single growth series – individual tree-level growth						
Larix sibirica	1960-1990	2636	TL	2548 (96.7%)	1448 (54.9%)	2312 (87.7%)
			CB	2047 (77.7%)	714 (27.1%)	1675 (63.5%)
			CT	2028 (76.9%)	610 (23.1%)	1641 (62.3%)
	1970-2000	2053	TL	1982 (96.5%)	1432 (69.8%)	1568 (76.4%)
			CB	1424 (69.4%)	809 (39.4%)	835 (40.7%)
			CT	1427 (69.5%)	806 (39.3%)	759 (37.0%)
Larix gmelinii	1960-1990	3508	TL	3361 (95.8%)	2073 (59.1%)	3012 (85.9%)
			CB	2547 (72.6%)	1089 (31.0%)	2082 (59.4%)
			CT	2666 (76.0%)	864 (24.6%)	2189 (62.4%)
	1970-2000	2130	TL	2084 (97.8%)	1434 (67.3%)	1829 (85.9%)
			CB	1393 (65.4%)	566 (26.6%)	1090 (51.2%)
			CT	1575 (73.9%)	699 (32.8%)	1107 (52.0%)
Population chronology – population-level growth						
Larix sibirica	1960-1990	117	TL	113 (96.6%)	57 (48.7%)	95 (81.2%)
			CB	97 (82.9%)	35 (29.9%)	73 (62.4%)
			CT	97 (82.9%)	31 (26.5%)	71 (60.7%)
	1970-2000	94	TL	89 (94.6%)	60 (63.8%)	64 (68.1%)
			CB	64 (68.1%)	34 (36.2%)	36 (38.3%)
			CT	69 (73.4%)	38 (40.4%)	34 (36.2%)
Larix gmelinii	1960-1990	124	TL	116 (93.5%)	56 (45.2%)	103 (83.1%)
			CB	89 (71.8%)	41 (33.1%)	82 (66.1%)
			CT	105 (84.7%)	30 (24.2%)	94 (75.8%)
	1970-2000	64	TL	63 (98.4%)	41 (64.1%)	50 (78.1%)
			CB	42 (65.6%)	13 (20.3%)	35 (54.7%)
			CT	45 (70.3%)	18 (28.1%)	31 (48.4%)

* TL: T-linked method; CB: moving calendar-based method; CT: Calendar-based T-linked method.

Line 148: Report the sample sizes with the percentages.

-- Thank you. We have reported the sample sizes in the revision as suggested by the reviewer (Lines 149-165).

Lines 149-150: Why not 1980-2010 as well? This would bring your analysis closer to the relevant recent time period.

-- Thank you for your comment. We have not introduced the third analysis time period of 1980-2010 mainly for the two reasons below:

1) In this study, we determined the local climatic conditions of each sampling population using the WorldClim datasets, which currently provides climate layers at a high spatial resolution of 2.5' that is aggregated over two time periods, 1960-1990 and 1970-2000. The high spatial resolution of this dataset helps us distinguish the local climatic conditions of the intensive sampling populations, which is critical for accurately quantifying the relationship between growth-temperature response pattern and local climate.

2) Of the 260 sampling populations, only 61 were sampled in or after 2010 and are able to fully cover the period of 1980-2010. Moreover, these 61 populations are mainly located in the warmest southern parts of boreal Eurasia (only 11 populations are located north of 60°N), suggesting poor spatial representativeness. Introducing these data would increase the relative weight of the southern distributions of boreal larch. Based on our analyses and this scenario, as well as previous studies, we hypothesize that the results are more likely to be negatively affected by increasing temperature.

Taken altogether, we do not plan on introducing the 1980-2010 time period into our analysis. We do, however, look forward to the update and enrichment of tree-ring data across high latitudes in Eurasia so that we can better study the continuous changes of growth-climate response patterns over time.

Line 154: Are you sure these are individual tree tree-ring series? It is common practice to extract multiple cores from a single tree.

-- Thank you. As described above, we have counted the tree numbers of each sampling population, established the individual tree chronologies, and conducted correlation analysis on individual tree chronologies.

Also, report sample sizes whenever you report these percentages because n is not equal through all these comparisons.

-- Thank you. We have reported the sample sizes in the revision as suggested by the reviewer (Lines 149-165).

Line 154: The change in proportion of sensitive trees is an indication that growth-climate relationships are indeed unstable over time. Because you essentially only have 2 data points here, why not add another for 3? 1980-2010. The number of larch populations dwindles as you move forward into time, but it seems like the most recent possible period is too important to skip. One thing you could do beyond simply calculating percentages of chronologies with sig. correlations is focus on the trees/populations that cover all 3 periods and report how their growth-temperature relationships have changed through time. You could get a much better sense of this by advancing your 30-year moving windows by 5-year (or even 1-year) instead of 10-year steps. Then you could plot time on the x-axis and 30-year correlation coefs on the y. I don't think this would be too much additional work if you are already calculating t-linked series for 1960-2010. The insight gained would be really valuable. Perhaps this would aid you in your interpretation of warming-threatened and warming-benefits trees/populations/areas.

-- Thank you very much for the suggestion. We agree with the reviewer that the comparison between two time periods is not informative enough, so we have reduced the comparison in our revision. The reasons or difficulties why we have not introduced other analysis time periods are explained above.

Nonetheless, we totally agree with the reviewer that it is a very inspired research idea to study the continuous change of the temperature sensitivity of tree growth in time through the sliding the analysis period at certain steps. "The insight gained would be really valuable". – yes! Agree!! In the follow-up analysis, we focus on using the data we directly collected more recently in China. We calculated the response patterns of these data during the 30-year moving windows at one-year steps from 1960-1990 to 1980-2010. We counted the number of populations showing significant negative temperature sensitivity during 30-year window and then plotted these results using the end year of the 30-year window on the x-axis and the number of negatively-responding populations as y-axis (Figure below). We found that the number of negatively-responding populations significantly increased over time (Pearson's $r = 0.664$, $p = 0.001$, $n = 21$). This small-scale attempt proved the feasibility and significance of the reviewer's suggestion. However, as this is not the focus of this project and due to the temporal bias of available data, we only present these limited results here. We thank the reviewer for the inspiration!

Lines 164-165: It is difficult to assess this claim without yet understanding the method (I needed to read the supplement to begin to understand it). What do you mean by “temperature variations”? After reading the Supplement file, I agree with the first part of the sentence (this is an effective approach over monthly climate series).

-- Thank you for your comment. We have restructured this part of the Main text in the revision by moving some content explaining the motivation and principle of the T-linked method from the Supplementary Information to the Main text, and provided some of this nuance in the Main text, so as to make the writing more coherent and more in line with readers' understanding process. The new text reads:

Lines 90-118, Introduction section: *“A potential source for the muted growth-temperature responses in tree-ring-based studies could be linked to the calendar-based approach for calculating growth-temperature correlations. Variation in tree growth is usually related to climatic variation. However, identifying the climate factors and time periods over which climate affects tree growth has been problematic (van de Pol et al., 2016). General methods for detecting growth-temperature responses typically correlate annual tree growth with temperature series for rigid calendar periods (months, seasons, or annual), meaning that choice of time periods is fixed, arbitrary, and might not follow what the physiology of trees require for growth. Since xylem phenology is mainly regulated by temperature cues (Huang et al., 2023; Rossi et al., 2016; Rossi et al., 2008), the same calendar period can correspond to very different growth phases from year to year due to the great interannual variations in seasonal dynamics of temperature in boreal regions (Osawa et al., 2010; Vaganov et al., 1999; Zhang et al., 2018), therefore, correlating annual growth with temperatures for fixed periods may result in underestimations of growth-temperature response. For tree living under continuous climatic stress or in regions with consistent growing seasons, monthly or seasonal temperature series can be relatively effective to investigate growth-temperature relations. However, for trees living in regions with comparably variable climates, it is not likely appropriate to conclude that trees are insensitive to temperature when climate series are based solely on a human calendar. Boreal growing seasons, especially near the Arctic Circle, are often shorter than two months and their starting and ending dates vary greatly between years (Osawa et al., 2010; Seftigen et al., 2018). In addition, larches dominant in boreal Eurasia are deciduous conifers that need to renew their needles at the beginning of each growing season. This trait limits xylem formation in both rate and duration (Rossi et al., 2009), further compressing growth response windows and confusing the detection of response signals using calendar-based approach. Given the lack of consistency between tree-ring-based and satellite-based studies, the potential limitations of calendar-based methods, and the physiological uniqueness of larch, it seems necessary to develop new methods to construct more flexible and physiologically-relevant temperature series. Doing this could both improve the detection of growth responses across these forests and potentially clarify the seemingly inconsistent results shown by different methodological approaches.”*

Line 167: You need to describe what you did with the monthly data too - at least briefly in the methods and in detail in the supplement. In order for your methods to be comparable, I hope that you examined both individual months and aggregates of months (i.e., seasons).

-- Thank you for your comment. As described above, we have used much more calendar-based temperature series in the revision, and described in detail how we performed this part of analysis in the Methods section. (Lines 615-636)

Lines 172-173: What does this mean? Also, incorrect grammar.

-- Thank you for your comment, we have corrected the grammar and rewritten this sentence in the revision (Lines 175-179). Local mean annual temperature and mean annual precipitation (MAT and MAP) show significant positive correlation across the study area. Therefore, to assess the independent effects of MAT and MAP on the correlation-derived metrics, we calculated partial correlations between the metrics and local MAT or MAP, whereby the covarying effects of MAT and MAP on these correlations were controlled for. The new text reads:

Lines 175-179: Results section: “After we used partial correlation to control the covarying effects of precipitation and temperature, the four correlation-derived metrics characterizing negative temperature sensitivity were all positively correlated with local temperature and negatively correlated with local precipitation, respectively ($p < 0.01$; Supplementary Fig. 4).”

Lines 173-175: we don't know what “the four” are at this point, so maybe just say “four metrics” and give a description of what they are. I think the lack of description of these metrics and how they are derived is a major weakness of this paper as it is currently. At minimum, a whole supplement section should be devoted to explaining these metrics. They are critical as you don't report actual correlation coefficients (I realize this may be impractical in the main text due to the complexity of your method).

Line 177: “...both wider affecting scopes and longer affecting durations.” I don't understand this.

-- Thank you for your comment. We have rewritten the descriptions of the correlation-derived metrics in a more detailed manner as a separate subsection in the Methods section of the main text. (Lines 575-602, also as described below)

Essentially, we established four metrics based on the growth-temperature correlation results to measure the size of temperature effects on tree growth and to characterize the structure of growth-temperature response pattern for each population from two dimensions:

(i) Affecting scope, i.e., what is the proportion of individual trees (individual tree chronologies or single standardized growth series) in a population that show a significant relation to temperature. We calculated the percentage of individuals in each population showing significant negative or positive correlations with each T-linked temperature series from the nearest meteorological station. The maximum value of these percentages was used as the first metric, that is, the maximum

proportion of individuals in a population that show negative or positive sensitivity to temperature, representing the maximum scope of negative or positive temperature effects on each population.

(ii) Affecting duration, i.e., how long the growth response to temperature lasts. As the time period used to construct the same T-linked temperature series varies from year to year, we calculated the mean length (in the number of days) of the varying T-linked periods over corresponding analysis period for each T-linked temperature series. We calculated the average mean lengths of the T-linked temperature series which were significantly correlated with both population- and individual-level growth series. We suspect that the metrics established based on this attribute can be interpreted as a way of describing the mean duration of negative or positive responses of growth to temperature for a population.

We used these four correlation-derived metrics to measure the size of temperature effects on tree growth or characterize the structure of growth-temperature response pattern. Our results showed that these metrics were sensitive to local climates of the populations.

Finally, we also calculated the mean coefficient of all significant correlations between each population chronology and the T-linked temperature series from the nearest meteorological station to measure the size of temperature effects on tree growth. The four metrics and the mean correlation coefficient were correlated with the local climates (Lines 174-196).

Lines 189-190: Grammar.

-- Thank you. We have corrected this sentence in the revision. The new text reads:

Lines 194-196, Results section: *“Interestingly, the latitudinal cluster prevailed over the longitudinal cluster (Supplementary Fig. 7), which reinforces the role of local temperature in determining the growth-temperature response pattern.”*

Line 198 (Fig. 2): This figure is important, but I’m not sure it is illustrative enough to warrant inclusion in the main text. Panel b should definitely move to the supplement. Panel a could also go to the supplement, or you could probably express the same data in a more effective way by plotting each of your metrics (y-axis) against MAP and MAT on the axes. You could then print the correlation coefficients within the plots. This would be many more panels, but it would be a lot easier to interpret.

-- Thank you for your comment. As suggested by the reviewer, we have replotted this figure, printed the correlation coefficients on each panel, and moved it to the Supplementary Information as two figures, Supplementary Figs. 4 and 7.

Supplementary Fig. 4 | Partial correlations of the metrics characterizing the temperature sensitivity with the local climatic conditions. MST represents the maximum scope of temperature effects on populations, which is the maximum proportion of negative or positive temperature-sensitive single growth series in a population; tDur05/tDur50 represents the duration of tree-level growth-temperature response, which is the average mean number of days of the top 5%/50% T-linked temperature series in descending order of the proportion of individuals in the population that are significantly negatively or positively correlated with them; pDur represents the duration of population-level growth-temperature response, which is the average mean number of days of all T-linked temperature series with which the population chronology is significantly negatively or positively correlated. The horizontal lines denote the 0.05 significance level; red and blue represent mean annual temperature and precipitation, respectively.

Supplementary Fig. 7 | Probability distributions of climatic factors and geographical coordinates grouped by populations showing positive (red) and negative (blue) temperature sensitivity. The distribution functions were estimated by kernel density; p -values of Student's t -test are noted on each panel; vertical dashed lines represent the average values of each factor.

Line 266: I am not convinced that this analysis is valid. The analysis is sophisticated, but you are essentially extrapolating growth-temperature relationships from 1970-2000 into the future. Perhaps you are capturing some of the mechanisms involved in the growth-temperature relationships by using MAT and MAP in your model - is this what you have assumed? I think this is clever but would like to see the authors address this concern explicitly here or in the methods.

-- Thank you for your comment. In this study, we found that local climatic conditions of larch population played an important role in determining whether it showed negative growth-temperature responses. The populations with and without negative responses to temperature roughly occupied opposite climate spaces.

Nevertheless, we agree with the reviewer that extrapolations outside the range of data used to develop the model are speculative. In some parts of the larch distribution, especially in the south, the projected climatic conditions are outside the current whole climate spaces occupied by boreal larch. These projected climatic conditions do not seem to be within the realm of any boreal larch. It is hard, however, to not acknowledge that this may indeed change the current relationships between local climate and growth-climate response pattern of boreal larch.

To address this concern, we have recalculated the values used to plot Figure 5 (now is **Figure 4**) while marking the areas outside the current climate space of boreal larch. Although the removal of these areas does not change the general results and conclusion we reported here, removing them would increase the relative weight of northern parts, as these reduced areas are mainly located in the warmest southern parts of larch distribution. Therefore, we chose to keep these areas in our projections but we have marked them in **Figure 4** (grey shadow as shown in the below figure) and added significant additional information in the Methods section and caveats in the discussion of the main text.

Lines 357-361, Discussion section: *“There are substantial uncertainties in extrapolating the models estimated based on current growth-temperature relations to climate projections. However, the consistent qualitative spatial trends revealed here suggests manager and policymakers need to consider the possibility of emerging negative impacts of warming across much more of the Eurasian boreal forests than previously estimated.”*

Lines 685-689, Methods section: *“Considering the uncertainties in extrapolating the models to*

climate projections falling outside the data used to fit the models, we marked the areas where projected climatic conditions beyond the 1970-2000 baseline climate space of boreal larch. These areas were limited and concentrated in the southern margins of species distributions (Fig. 4; Supplementary Fig. 6)."

Line 392: "which report" instead of "where report"

-- Thank you. We have rewritten this sentence in the revision. The new text reads:

Lines 388-391, Discussion section: *"Recent studies using satellite observations and simulated vegetation proxies also have reported weakening low-temperature control, emerging negative impacts of warming (Piao et al., 2014; Wang et al., 2018), and future reversal of warming-enhanced vegetation productivity in boreal regions (Zhang et al., 2022)."*

Line 439: Did you compute tree means for your samples and those from the ITRDB? If not, you should!

-- Thank you. As described above, we have deciphered most of the naming conventions and provided the tree number for each population in Appendix A. We have also reported the total number of trees in our tree-ring network in the revision:

Lines 130-133, Introduction section: *"To achieve these goals, we compiled an extensive tree-ring network composed of 8,544 annual radial growth series of 5,089 larch trees and 260 larch populations covering the distributions of boreal larch (Fig. 1)."*

Lines 430-440, Methods section: *"Our tree-ring network includes annual radial growth series from 131 Siberian larch populations and 129 Dahurian larch populations across boreal Eurasia. These data were collected from three sources: (i) ITRDB, 193 populations (7,271 raw tree-ring width series from 4,605 trees), 27 of which were contributed by us; (ii) fieldwork during the 2010s, three Siberian larch populations and 17 Dahurian larch populations (1,273 raw tree-ring width series from 484 trees) from the Altai Mountains in northwest China and the Greater Khingan Range in northeast China, respectively, with the aim of expanding the data coverage to include the southernmost distributions of the two species; and (iii) previous research, specifically 47 population tree-ring chronologies (established based on 2,785 core samples from 1826 trees) with accurate geographic coordinates and suitable temporal coverage digitalized from published literature for further spatially enriching our network."*

Line 443: remind us what the analysis periods are

-- Thank you. We have added the periods in the revision. The new text reads:

Lines 562-565, Methods section: *"We excluded the growth series shorter than 25 years during the analysis period. In total, 6,144 and 4,183 single standardized growth series (3,638 and 2,446 individual tree chronologies) were retained for correlation analysis during 1960-1990 and 1970-*

2000, respectively.”

Lines 445-448: Good choice using the raw ring width series from ITRDB (and making your own chronologies) rather than ready-made chronologies. Also, it is great that you tell the reader what methods you used to do this. However, there are still some potentially major issues with the method you chose. This is a crucially important decision in your analyses - perhaps the most important decision.

You are correct that the main reason to detrend/standardize tree ring series is to remove the signals (noise in your case) that are due to endogenous biological processes related to wood production. These signals manifest in 2 main ways: trends in the magnitude of ring widths (e.g., neg. exp decline or hugershoff-type decline) and trends in the variance of ring widths (series are typically heteroscedastic). There are many ways to handle this, each with its own pitfalls. The typical methods describe and remove lower-frequency (user-specified) trends and stabilize the variance. The “art” of detrending is really to choose the method(s) that minimize the pitfalls for your particular analysis - there is no universally perfect detrending method. Most of the “conservative” detrending options fit some kind of curve to the raw ring widths, then take the ratio of the raw ring widths:fitted values. The ratio stabilizes variance, but can also introduce artifacts, particularly at the ends of the series (or anywhere when the curve doesn’t fit the data well). See the 2 papers below for an overview on these issues.

Your ultimate goal here is to have tree-level and population-level series/chronologies that reflect reality and are statistically sound for your 30-year correlation periods. Especially because you are comparing differences in relationships between 1960-1990 and 1970-2000, it is critical that the variance is stabilized over time and that there are not artifacts from the detrending method at any frequency. Both of these could confound your analysis.

*The default dplR “Spline” option is unlikely to accomplish what you hope it would (remove age/size trends) in all of these series. In fact, it may add some artifacts of its own. Using variable wavelength splines on series of different lengths doesn’t make sense to me. For example, a 2/3*series length wave length spline gives 100 years for a 150-year series and 33.3 years for a 50-year series. These are very different levels of stiffness, so you are removing signals of varying frequencies for each series. This seems like a bad way to go about this. Your 30 year correlation periods could easily be influenced by artifacts from your detrending method.*

The authors of dplR warn against adopting detrending methods without careful inspection in the description file for `detrend.series()` : “Automating detrending and not evaluating each series individually is folly.” I agree with this, although it presents a challenge when you have 8.5k series! Perhaps you could look at a subsample of series individually, or better yet, make 260 plots (one for each population) and plot all tree-ring series contained within with thin lines. These will be noisy plots, but any weird end effects or other outliers should be obvious. There really is no substitute for looking at plots of the data. You will have to do some of this.

I can’t tell you what method will be the best for your tree ring data, but perhaps I can give you some

direction. I would 1st try a constant 30-year spline fit (will remove all frequencies lower than that, which you can't assess anyway) and use ratios (difference = FALSE) to stabilize the variance. Take a look at some plots to make sure there aren't weird end effects from computing ratios. The 30-year spline should be flexible enough to fit most series well and avoid end effects. If not, consider Cook & Peters (1997) power transforming & residual method with the 30-year spline (transform is done before curve fitting to stabilize variance, then curve is fit, then transformed rw series are subtracted from the fitted values). Which ever method you end up with, please also describe in the methods section why you chose this method. I think also you need to compute AR residual series ("prewhitening") after detrending and before aggregating to chronologies - this will remove any autocorrelation in the series that could make your Pearson correlations invalid. Note: if you use Cook & Peter's method, you will have to add a constant (say 1 or the original series means) to the transformed residual series before prewhitening.

See

Coulthard et al., 2020, The limits of freely-available tree-ring chronologies, Quaternary Science Reviews 234

Cook & Peters, 1997, Calculating unbiased tree-ring indices for the study of climatic and environmental change, The Holocene 7(3)

-- Thank you so much for such a detailed and systematic comment. We fully understand the importance of the choice of the detrending method for our research. You made it clear that the detrending method we used before might not be appropriate.

As suggested by the reviewer, we first re-detrended the raw tree-ring width series by fitting a cubic smoothing spline with a 50% frequency cut-off at 30 years and then dividing the ring widths by the fitted values. Then we plotted all single detrended growth series for each population to check for the severe 'end effects' induced by calculating ratios. For those series, we detrended them by subtracting the fitted values from raw ring widths instead of division and used a data-adaptive power transformation to stabilize the variance prior to detrending. Importantly to this concept, most of our analyses and results were not, however, affected by the 'end effects' of most tree-ring data, because we only used the segments of the growth series during 1960-1990 or 1970-2000. These periods do not cover the end part of most tree-ring series. All the resulting dimensionless growth series after detrending were then prewhitened to remove the autocorrelation by fitting an autoregressive model, then we used the standardized and prewhitened growth series to construct the tree-ring chronology for each tree and each population. (Lines 447-461)

Also, to test whether our subsequent analyses were robust to detrending choices and how the choices of detrending methods influence our results, we tested three commonly-used methods of detrending with or without prewhitening. We calculated the main results using growth series detrended following different methods (Lines 469-472), and we found subtle but no essential changes in our main results (Supplementary Fig. 13, see below). We speculated that this might be related to the fact that we only used fragments of the chronologies in the analysis. For most of our data, the different detrending methods appear to be stable for the middle part of the growth series.

Supplementary Fig. 13| Comparison of the main results depending on different detrending method used. We calculated the main results used the growth series detrended by different methods, with and without prewhitening. **a.** 67% length spline function without prewhitening; **b.** 67% length spline function with prewhitening; **c.** negative exponential function without prewhitening; **d.** negative exponential function with prewhitening; **e.** 30-year length spline function without prewhitening. In each panel, the horizontal and vertical dashed lines denote the average values of climatic conditions, the slant dashed line represents the 0.50-probability climate boundary estimated based on the growth series detrended by 30-year length spline function with prewhitening, the slant solid line represents the 0.50-probability climate boundaries estimated based on the growth series detrended by the method labeled. The analysis formulas of the climate boundaries are noted on each panel.

Lines 447-461, Methods section: “To remove the low-frequency trends related to tree age or size, while accentuating the high-frequency climatic signals (Esper et al., 2002), each raw ring-width series was standardized by fitting a cubic smoothing spline with a 50% frequency cut-off at 30 years to it and then dividing it by the fitted curve to represent annual radial growth. We originally conducted a similar analysis using a 67% spline. Differences between the two approaches in the overall results were negligible. A small subset of growth series with obvious ‘end effects’ from calculating ratios were detrended by subtracting the fitted values from raw ring widths, while using a data-adaptive power transformation to stabilize the variance prior to detrending (Cook and Peters, 1997). The resulting dimensionless growth series after detrending were then prewhitened to remove the autocorrelation by fitting an autoregressive model. The standardized and prewhitened

growth series from the same tree were averaged into individual tree chronologies to create tree-level radial growth, while all growth series from the same population were averaged into population chronologies using bi-weight robust mean (Cook, 1985) to create population-level radial growth."

Lines 469-472, Methods section: *"To test whether our subsequent analyses were robust to the choices of detrending methods, we tested three commonly-used detrending methods, and with or without prewhitening, and found only subtle but no essential changes in our main results (Supplementary Fig. 13)."*

Line 470: Nowhere in here do you report how you prepared the data and performed analyses for the rigid monthly climate series that you show results for in Table 1. Did you examine individual months as well as aggregates of months?

-- We calculated monthly temperature series based on daily mean temperatures from meteorological stations. We only examined individual months in the initial submission. As described above, we have used much more calendar-based temperature series through sliding correlation analysis as suggested by the reviewer, and described in detail how we performed this part of analysis in the Methods section. (**Lines 622-630**)

Lines 622-630, Methods section: *"(i) moving calendar-based method. We constructed the calendar-based temperature series using 36 time windows fixed to the calendar with variable widths from 15 days to 120 days at intervals of 3 days. We run each of these fixed windows over daily mean temperature series between April 1st (DOY 91) and September 30th (DOY 273), and averaged the daily mean temperatures within each window year by year to construct the temperature series. As a result, each growth series was correlated with 2,106 calendar-based temperature series, a number that exceeds the average number of T-linked temperature series correlated with each growth series, which is 1,548. The commonly-used seasonal, bimonthly, monthly, and semi-monthly temperature series were all included in the analysis through this way."*

Line 474: "yet-effective" is an inappropriate use of hyphenation. Correct would be: "...a simple, yet effective, approach..."

-- Thank you. We have corrected the text. (**Line 511**)

Line 483: Well, there are multiple human calendars. Most of the climate datasets use the western Gregorian calendar. Either way, they are arbitrary relative to tree physiology. You could say "linked to human calendars".

-- Thank you. We have corrected the text. (**Line 520**)

Lines 503-504: This must be temperature at the start of green up, right? Good check on temperature limits of growing seasons.

-- That's right. This is the lowest temperature that can correspond to the start of green up across

boreal Eurasia based on satellite-derived NDVI data.

Line 518: You might specify the WorldClim layers as 30-year climate normals. Give the resolution of WorldClim pixels here too.

-- Thank you for your suggestion. We have added the spatial resolution of WorldClim here in the revision as suggested. The new text reads:

Lines 555-556, Methods section: *"The WorldClim climate layers at a spatial resolution of 2.5' used to characterize long-term local climatic conditions are aggregated over two periods: 1960-1990 and 1970-2000."*

Lines 532-533: Are Pearson correlations appropriate here? Given the nature of the tree ring data (and your choice of detrending method), it seems the IID assumption will not be met here. Maybe the climate and (detrended) tree-ring series are \pm identically distributed, but with autocorrelation present in the tree-ring series (maybe the climate series too) observations are not independent. Probably the best solution is to compute AR residual series of your detrended tree ring series (aka "prewhitening"). This will remove all but the high-frequency (interannual) variation in the series. This should be fine - even desirable - in your case because you are only doing correlations for 30-year periods. See my detailed comments above on detrending/ standardizing your tree ring series.

-- Thank you. As described above, all standardized dimensionless growth series were prewhitened to remove the autocorrelation by fitting an autoregressive model before we constructed the tree-ring chronologies in the revision.

Lines 542-545: This is fantastic. I've noticed a tendency recently for researchers to lump significant positive and negative correlations together into one metric of sensitivity - obscuring crucial ecological differences.

-- Thank you.

I was pleased to read this, then confused when in the next sentence you appear to combine both negative and positive correlations into a single metric. You probably could reduce this confusion by substituting "negative/positive" with "negative or positive".

-- Thank you. We have corrected the text as suggested. (Lines 585-602)

I also think you could do a better job clearly explaining your metrics - it is still hard to understand such multi-layered metrics intuitively. The simple stuff (the actual correlation coefficients) is buried fairly deep here.

-- We have rewritten the descriptions of the correlation-derived metrics in a more detailed manner as a separate subsection in the Methods section of the main text. (Lines 575-602, also as described above)

Line 545: “maximum proportion” Is this the maximum proportion out of all the different temperature series (with different L & R values)? This is not clear as written.

-- You are right, as described above, we calculated the percentage of individual trees (individual tree chronologies or single standardized growth series) in a population that show a significant negative or positive relation with each T-linked temperature series from the nearest meteorological station. The maximum proportion (MST) is the maximum value out of the percentages of all the T-linked temperature series, i.e., the maximum proportion of individuals in a population that show negative or positive sensitivity to temperature. Here, we simply discussed whether there were significant growth-temperature correlations without distinguishing in which part of the growing season these correlations occurred, which is the potential future research and needs to be supported by more targeted sampling designs.

Line 550: not clear what the “average mean” is. Please explain in more detail.

-- As the time period used to construct the same T-linked temperature series varies interannually, we calculated the mean length (number of days) of the varying T-linked periods over corresponding analysis period for each T-linked temperature series. We averaged the mean lengths (numbers of days) of all or part of the T-linked temperature series that the growth series significantly correlated with according to certain conditions to describe the size of temperature effects on tree growth from the duration of growth response to temperature. This process led us to call it the “average mean”. We have rewritten the descriptions of these metrics in a more detailed manner, as a separate subsection in the Methods section of the main text. The new text reads:

Lines 585-600, Methods section: “*As the time period used to construct the same T-linked temperature series varies inter-annually, we calculated the mean length (in the number of days) of the varying T-linked periods over corresponding analysis period for each T-linked temperature series. We then ranked all T-linked temperature series correlated with the same population in descending order of the percentage of single growth series (individual tree chronologies) showing significant negative or positive correlation with them, and averaged the mean lengths of the temperature series ranked top 5% and 50% as metrics tDur05 and tDur50 (days), respectively. Another metric was derived from the population-level correlations, corresponding to the average mean lengths of all T-linked temperature series with which the population chronology was significantly negatively or positively correlated (pDur, days).*”

Lines 559-561: This doesn’t make sense to me, how can a chronology show both negative and positive correlations within the same 30-year period? Would this be for different temperature series? You should clarify.

-- We correlated each population chronology with all T-linked temperature series from the nearest meteorological station, therefore, it is possible for the same population chronology to show both negative and positive sensitivity to temperature, as shown in **Supplementary Fig. 18** as below.

Tree growth can show opposite responses to temperature at different stages of the growing season.

For example, previous studies have reported that larch species may show opposite responses to temperature during the phase of rebuilding their crowns (usually the early growing season) and during the active period of cambium (usually the middle growing season) (Li et al., 2019; Zhang et al., 2021; Zhou et al., 2021).

Supplementary Fig. 18| Correlation results between population-level growth series and the qualified T-linked temperature time series. The blue-white-red gradient represents correlation coefficients from positive to zero to negative. Label of ‘x’ denotes significant correlations ($p < 0.05$).

References:

Li, W. et al., 2019. Diverse responses of radial growth to climate across the southern part of the Asian boreal forests in northeast China. *Forest Ecology and Management*, 458: 117759.
 Zhang, Y. et al., 2021. Higher plasticity of water uptake in spruce than larch in an alpine habitat of North-Central China. *Agricultural and Forest Meteorology*, 311: 108696.
 Zhou, P. et al., 2021. Radial growth of *Larix sibirica* was more sensitive to climate at low than high altitudes in the Altai Mountains, China. *Agricultural and Forest Meteorology*, 304-305: 108392.

Lines 620-623: I hope that you will strongly consider archiving your code with an online database that is freely available to anyone interested. You have developed a great method that others should be able to play around with (and perhaps improve).

-- Thank you.

We have made our main codes available via <https://doi.org/10.6084/m9.figshare.22590235>.

Line-specific comments for the supplementary information:

Line 14 (Fig. S1): I think this figure would be more helpful if it was just your study region (like figure 1) instead of all of the northern hemisphere.

-- Thank you for your suggestion. We plotted the entire northern hemisphere to compare the climate changes of the study area with other regions of the world, so as to highlight that the warming rate in our study area was higher than the global average during the past decades. In our revision, we have replotted this figure, streamlined the content, and explicitly highlighted the extent of the study area in each panel as below:

Supplementary Fig. 1 | Climate changes across the Northern Hemisphere over 1960-2020. Trends of (a) annual mean temperature ($^{\circ}\text{C dec}^{-1}$), (b) annual total precipitation (mm dec^{-1}) and (c) annual total potential evapotranspiration (mm dec^{-1}) over 1960-2020. The black lines denote the distribution area of Eurasian boreal larch forests. Colored cells represent significant trends ($p < 0.05$), while grey cells represent non-significant trends. Climate data are collected from CRU TS 4.05 at a spatial resolution of 0.5° .

Line 31: experience would be a better word than “suffer”

-- Thank you for your suggestion. We have corrected the text in Line 29.

Line 110: There are a lot fewer populations represented in the 1970-2000 period than in 1960-1990 (as you report in the text). If you are accepting diminishing sample sizes here, why not cover the next 30-year period (1980-2010) too? It might give you some insight into how growth-climate relationships are changing over time.

-- Thank you very much for your suggestion. We agree with the reviewer that the comparison between two time periods is not informative enough, and the sample size dwindles a lot from 1960-1990 to 1970-2000, so we have reduced the comparison between the two periods in the revision. The reasons or difficulties (limitation of the climate data and the bias from poor spatial representativeness) why we have not introduced other analysis time periods have been explained in the responses above. As we move forward into time, the number of larch populations dwindles, and the latitudinal bias increases. We look forward to the update and enrichment of tree-ring data across high latitudes in Eurasia so that we can better study the continuous changes of growth-climate response patterns over time.

Line 127-140: This is fantastic. I am really glad you have taken on this task! I have one challenge to you approach: wouldn't heat sums (e.g., GDD) be better than means? Also, heat sums might simplify the analyses.

Line 201: Did you consider computing heat sums instead of means?

-- Thank you for your comment. The T-linked method was actually inspired by the method of determining thermal growing seasons based on certain temperature thresholds (and then calculating the growing degree days, GDD). We initially considered using accumulated temperatures instead of mean temperatures.

However, the concept of GDD is not applicable to most T-linked periods. The thermal growing seasons determined based on temperature thresholds and the T-linked periods both vary from year to year, but compared with the thermal growing seasons, most of the T-linked periods are much shorter (**Supplementary Fig. 16**). This phenomenon amplifies the proportion of the interannual fluctuations in window length. When calculating accumulated temperatures, the interannual fluctuation range lasting a few days to a little more than ten days has negligible influence on the entire, 100+ day growing seasons. The inter-annual fluctuation of the T-linked period length may cause the mean temperature and the accumulated temperature to represent completely different thermal conditions. The accumulated temperature of a relatively longer, but a colder time period may be greater than that of a relatively shorter, but warmer time period. The inter-annual variation of accumulated temperature may mainly come from the inter-annual variation of the length of the T-linked periods rather than that of the climatic conditions, which is particularly obvious for the relatively short T-linked periods (**Supplementary Fig. 16**).

Nonetheless, we generally agree that using temperature thresholds to determine the thermal growing season and then using a GDD series to correlate with growth series is also an efficient way to avoid the rigid calendar periods (Gao et al., 2022; Vaganov et al., 1999). But we found out the GDD approach might be more appropriate for detecting the overall growth response to temperature during the entire growing season or estimating the forest productivity. The temporal

flexibility is the main advantage of the T-linked method over the GDD.

References:

Gao, S. et al., 2022. An earlier start of the thermal growing season enhances tree growth in cold humid areas but not in dry areas. *Nature Ecology & Evolution*, 6(4): 397-404.

Vaganov, E.A., Hughes, M.K., Kirdyanov, A.V., Schweingruber, F.H. and Silkin, P.P., 1999. Influence of snowfall and melt timing on tree growth in subarctic Eurasia. *Nature*, 400(6740): 149-151.

Line 144: “Creatively” is not a word you generally want to read in the methods section of a scientific paper. I recommend you simply delete this word.

-- Thank you. We have deleted this word in the revision (Lines 196-198).

Line 221 (table S6): typo at 1981 length: 35 days not 350 days

-- Thank you. We have corrected the text in the revision (Line 274, Supplementary Table 7).

Line 224: For this figure and S13, S14, S15, add some thin grey grid lines to help readers track Temperature Value 1. This looks like ggplot2 to me. If so, you can add grid lines with: `theme(panel.grid = element_line())`. You will probably want to make sure that your white `geom_tiles` are actually filled (not blank) so that they cover the grid lines.

-- Thank you for your suggestion. We have replotted these figures as suggested by the reviewer in the revision (Supplementary Figs. 18, 19 and 20). See below:

Supplementary Fig. 18| Correlation results between the population chronology and the qualified T-linked temperature time series. The blue-white-red gradient represents correlation coefficients from positive to zero to negative. Label of ‘x’ denotes significant correlations ($p < 0.05$).

Line 231: This is a helpful graphical representation of your process. I think you still need to show us more, however. Really we need to see some time series plots of the temperature series that you generated. Maybe at least some of the series that have significant pos and negative correlations with your population chronologies?

-- Thank you for your suggestion. We have plotted a new figure (Supplementary Fig. 21) to show the T-linked temperature series in the revision as below:

Supplementary Fig. 21| The qualified T-linked temperature series during 1960-1990 in this example. Red, blue, and grey represent the T-linked temperature series with which the population-level growth series was significantly negatively correlated, significantly positively correlated, and uncorrelated relationships, respectively.

Line 238: This figure is a great opportunity to show some actual correlation coefficients between chronologies and climate series - which do not appear anywhere in your main text or in the supplement. You could keep the red and blue themes but use white-blue and white-red gradients to express the values of negative and positive correlation coefficients.

-- Thank you for your comment. We have replotted this figure (Supplementary Fig. 18, please see above), using blue-white-red gradient to represent correlation coefficients from positive to zero to negative in the revision, as suggested by the reviewer.

Reviewer #2 (Remarks to the Author):

Summary: This manuscript presents a new method for assessing temperature sensitivity of tree growth with temporally flexible temperature variables. They apply this method to two species of Eurasian boreal larch and use it to determine which populations and proportions of their ranges are expected to benefit from warming and are threatened by warming. Although I feel the authors need to clarify some terminology and further discuss this new method, I think this is an important advancement in the field. The methodology is mostly sound (though I have a few concerns I would like addressed) and with a few small tweaks I think the authors do a good job interpreting and discussing their results. I look forward to seeing this method further refined and integrated into statistical software to make it more widely available.

We thank the reviewer for the detailed and constructive comments for our manuscript. We would also like to thank the reviewer for the recognition of our work. We have incorporated the comments and responded to the concerns in this revised version of the manuscript.

Overall, the reviewer's primary concerns were 1) potential spurious correlations within the T-linked method, 2) comparing the T-linked and GDD methods, 3) clarification of terminology, 4) distribution area of boreal larch.

To address point 1), we performed additional analyses to test the effectiveness and advantages of the T-linked method over calendar-based methods and implemented 10-fold cross-validations on the T-linked results to test the method's robustness. For point 2), we constructed GDD series and directly compared their results with those of the T-linked method. For point 3), we have revised the original terminology to be clearer and more unambiguous. To address point 4), we have scaled down the claims in this section and highlighted areas where projections may be less reliable.

Besides these additional analyses, we have substantially revised the manuscript to reflect the reviewers' suggestions, leading to a much-improved manuscript.

Please find below our point-by-point responses to each of the reviewer's comments.

Major comments:

I think the t-linked method is novel and very helpful for moving towards more physiologically relevant climate indicators. The authors did an excellent job explaining the method and I appreciated the additional detail and figures in the supplement. However, how do we think about the potential for spurious correlations from running so many correlation analyses? With monthly values, this is already concerning (though partly accounted for in some approaches). How should we think about it here and what concerns should we have?

-- Thank you for your positive comments. We understand the concern related with the high number of correlations. In the initial submission, we attempted to limit the influence of potential spurious

correlations by raising the identification criteria for temperature sensitivity, that is, only the growth series showing significant correlations with >3 adjacent T-linked temperature series were identified as sensitive to temperature. The probability of finding spurious significant correlations with >3 adjacent T-linked temperature series is much lower, as these are likely randomly distributed. Raising the identification criteria for temperature sensitivity can greatly reduce the influences of the randomly-distributed spurious correlations on our main results.

In the revision, we performed a 10-fold cross-validation to test the quality and accuracy of the estimated logistic models (Hastie et al., 2001) (Lines 657-665). The observations for parameter estimation were divided into 10 subsets that were roughly of equal size. Nine of these subsets were used as training group to estimate a cross-validated model while the 10th subset was used for testing. The estimated error, which is used to determine classification accuracy for a logistic model, was calculated by applying the cross-validated model to the test group. This procedure was executed 10 times, with each subset taking turns as the test group. The resulting 10 classification accuracies were averaged as an estimate of the overall predictive performance of the final model. The results show that the logistic models we established for both larch species have fairly high cross-validation accuracy and quality (observed accuracy = 0.862 ± 0.064 , Cohen's $\kappa = 0.713 \pm 0.132$ for Siberian larch; observed accuracy = 0.885 ± 0.056 , Cohen's $\kappa = 0.706 \pm 0.131$ for Dahurian larch) (Lines 222-225). The spurious correlation should be random and have systematic effects on the results. The results of 10-fold cross-validation indicated high robustness and accuracy of the models built based on the different subsets of the correlation results using T-linked method. We also found similarly strong validation of the estimated quantitative relationships between growth-temperature response pattern and local climate conditions are robust and physiologically reasonable.

Lines 222-225, Results section: *“Our 10-fold cross-validation analysis indicated that the estimated models were robust and of fairly-high accuracy and quality (Siberian larch: observed accuracy = 0.862 ± 0.064 , Cohen's $\kappa = 0.713 \pm 0.132$; Dahurian larch: observed accuracy = 0.885 ± 0.056 , Cohen's $\kappa = 0.706 \pm 0.131$).”*

Lines 657-665, Methods section: *“We tested the quality and accuracy of the estimated logistic models using 10-fold cross-validation analysis (Hastie et al., 2001). The observations for parameter estimation were divided into 10 subsets that were roughly of equal size. Nine of these subsets were used as training group to estimate a cross-validated model while the 10th subset was used for testing. We calculated the classification accuracy and the Cohen's kappa (κ) of the cross-validated model applied to the test group to measure the classification performance. This procedure was executed 10 times, with each subset taking turns as the test group. The resulting 10 classification accuracies and 10 Cohen's κ were averaged as estimates of the overall predictive performance of the final model.”*

Also, as another reviewer said, the effectiveness and advantage of the T-linked method over the calendar-based method have not been tested and verified under the condition of computing correlations of the same order of magnitude. We have supplemented this part of the analysis in the revision to further demonstrate that the significant growth-temperature correlations detected by our

T-linked method were not simply due to more correlation calculations. The spurious correlations should be randomly distributed. If T-linked method would show substantially more significant correlations, this would indicate that the significant growth-temperature correlations detected by the T-linked method were not simply because we calculated more correlations. We used two different methods to construct the calendar-based temperature series, described as below or in Lines 615-636 in the revision:

(i) We constructed the calendar-based temperature series using 36 time windows fixed to the calendar with variable widths from 15 days to 120 days at intervals of 3 days. To create calendar-based temperature series of the same order of magnitude as the T-linked temperature, we run each of these fixed windows over daily mean temperature series between April 1st (DOY 91) and September 30th (DOY 273), and then averaged the daily mean temperatures within each window year by year to construct the temperature series. As a result, each growth series was correlated with 2,106 calendar-based temperature series, a number that exceeds the average number T-linked temperature series correlated with each growth series, which is 1,548. The commonly-used seasonal, bimonthly, monthly, and semi-monthly temperature series were all included in the analysis through this way. The results showed that although much more calendar-based temperature series were used in correlation analysis, we still detected significantly more temperature-sensitive growth series using the T-linked method (Lines 144-169 and Table 1 as below).

(ii) We averaged the interannually varying start and end dates of the T-linked periods used to construct each T-linked temperature series and then averaged the daily mean temperatures between the two mean dates year by year to construct a calendar-based T-linked temperature series. Each growth series was then correlated with all the calendar-based T-linked temperature series that corresponded on a one-to-one basis to the normal T-linked temperature series. Similarly, temperature-sensitive growth series detected using the calendar-based T-linked temperature series were significantly less than those detected using the normal T-linked temperature series (Lines 144-169 and Table 1 as below).

These sensitivity analyses suggest that the T-linked method are robust to spurious correlations. We discuss these points now in the discussion section, to clarify and make clear the best use of our method, and contextualize our results, as well as added a caveat in the Discussion section (Lines 310-316). We hope that the reviewer's concern is addressed with the multiple levels of supplementary analyses that we conducted.

Lines 144-169. Results section:

“We compared the T-linked method with two methods that develop temperature series based on fixed calendar periods under the statistical constraint of calculating correlations of the same order of magnitude. Using a moving calendar-based method, each growth series was correlated with 2,106 calendar-based temperature series, including commonly-used seasonal and monthly temperature series. Through this way, ~70% of the sampling individuals and populations were identified as sensitive to temperature. Specifically, 29.3% ($n = 1,803$) and 32.9% ($n = 1,375$) of the single tree growth series showed significant negative responses to calendar-based temperature

series from 1960-1990 and 1970-2000, respectively, while 61.1% ($n = 3,757$) and 46.0% ($n = 1,925$) responded positively (Table 1). Growth-temperature response signals detected using the calendar-based T-linked method (see Methods) were close to those detected using the general moving calendar-based method for both species and both analysis periods (Table 1).

By contrast, we found that >95% of the individual trees and populations were sensitive to temperature when using our more flexible T-linked temperature series approach. We found that the proportions of single tree growth series showing negative temperature sensitivity were 57.3% ($n = 3,521$) in 1960-1990 and 68.5% ($n = 2,866$) in 1970-2000, while the proportions of positive temperature-sensitive individuals were 86.7% ($n = 5,324$) and 81.2% ($n = 3,397$) (Table 1). Analysis at the population level showed the same set of trends. The percentage of populations that responded negatively was 46.9% ($n = 113$) in 1960-1990 and 63.9% ($n = 101$) in 1970-2000, while those responding positively were 82.2% ($n = 198$) and 72.2% ($n = 114$) (Table 1). These percentages were also consistent between species (Table 1). The overwhelming proportion of significant growth-temperature responses detected at both the tree and population levels using our new T-linked temperature series demonstrates the effectiveness of T-linked method and highlights the key role of temperature on larch growth across boreal Eurasia.”

Lines 310-316. Discussion section: “While there are potentially serious issues with conducting such a large number of calculations, we tested the risk of potential spurious correlations using the T-linked method through a fair comparison to calendar-based methods and suggested that the effectiveness of our method was not attributed to the increased number of calculations. Furthermore, the cross-validation analyses we conducted demonstrated fairly-high accuracy and robustness of our estimated models, suggesting limited influences of spurious correlations on the main results.”

Table 1| Summary of growth series showing temperature sensitivity detected using different temperature series construction methods

Species	Time period	Total number	Method*	Sensitive pct. (%)	Negative sensitive pct. (%)	Positive sensitive pct. (%)
Single growth series – individual tree-level growth						
Larix sibirica	1960-1990	2636	TL	2548 (96.7%)	1448 (54.9%)	2312 (87.7%)
			CB	2047 (77.7%)	714 (27.1%)	1675 (63.5%)
			CT	2028 (76.9%)	610 (23.1%)	1641 (62.3%)
	1970-2000	2053	TL	1982 (96.5%)	1432 (69.8%)	1568 (76.4%)
			CB	1424 (69.4%)	809 (39.4%)	835 (40.7%)
			CT	1427 (69.5%)	806 (39.3%)	759 (37.0%)
Larix gmelinii	1960-1990	3508	TL	3361 (95.8%)	2073 (59.1%)	3012 (85.9%)
			CB	2547 (72.6%)	1089 (31.0%)	2082 (59.4%)
			CT	2666 (76.0%)	864 (24.6%)	2189 (62.4%)
	1970-2000	2130	TL	2084 (97.8%)	1434 (67.3%)	1829 (85.9%)
			CB	1393 (65.4%)	566 (26.6%)	1090 (51.2%)
			CT	1575 (73.9%)	699 (32.8%)	1107 (52.0%)

Population chronology – population-level growth						
Larix sibirica	1960-1990	117	TL	113 (96.6%)	57 (48.7%)	95 (81.2%)
			CB	97 (82.9%)	35 (29.9%)	73 (62.4%)
			CT	97 (82.9%)	31 (26.5%)	71 (60.7%)
	1970-2000	94	TL	89 (94.6%)	60 (63.8%)	64 (68.1%)
			CB	64 (68.1%)	34 (36.2%)	36 (38.3%)
			CT	69 (73.4%)	38 (40.4%)	34 (36.2%)
Larix gmelinii	1960-1990	124	TL	116 (93.5%)	56 (45.2%)	103 (83.1%)
			CB	89 (71.8%)	41 (33.1%)	82 (66.1%)
			CT	105 (84.7%)	30 (24.2%)	94 (75.8%)
	1970-2000	64	TL	63 (98.4%)	41 (64.1%)	50 (78.1%)
			CB	42 (65.6%)	13 (20.3%)	35 (54.7%)
			CT	45 (70.3%)	18 (28.1%)	31 (48.4%)

* TL: T-linked method; CB: moving calendar-based method; CT: Calendar-based T-linked method.

How do these t-linked estimates compare to other estimates of growing season length or heat accumulation rather than merely monthly temperatures? Some analysis of how T-linked estimates improve upon heating degree days or growing degree days, for example, seems like it would clarify how much better this method is than other methods that attempt to avoid calendar months.

-- Thank you for your suggestion. We totally agree that using temperature thresholds to determine the thermal growing season and then calculate the growing degree days (GDD) to correlate with growth series is also an efficient and physiologically-relevant way to avoid the rigid calendar periods. Actually, the T-linked method was inspired by the GDD method.

At the same time, we found out the GDD approach might be more appropriate for detecting the overall response of tree growth to temperature during the entire growing season or estimating the forest productivity. GDD is likely limited to detect intra-annual growth-temperature responses, while the temporal flexibility is the main advantage of the T-linked method over the GDD. Using defined temperature threshold (e.g., 0°C or 5°C) to determine the thermal growing season and then calculate the GDD still means that growth series are correlated with the entire growing season conditions. GDD are highly informative for overall temperature effects on tree growth. GDD series tend to cover a long time period, e.g., May to August, which makes the GDD less able to adapt to temporal variability compared with our approach. That being said, we agree that comparing the T-linked and GDD methods would be highly useful and informative.

We extracted the timing and length of the thermal growing season for each year based on daily mean temperature series. The start of the thermal growing season was defined as the first five consecutive days with daily mean temperature at or above a series of gradient temperature thresholds (from 0°C to 10°C in steps of 0.5°C). The end of the growing season was defined as the first five consecutive days after 1st July with daily mean temperature at or below the temperature thresholds. The growing degree days (GDD), which represent the effective heat accumulation for

vegetation growth, were calculated as the sum of daily mean temperatures above the temperature threshold during the thermal growing season (Fu et al., 2014; Gao et al., 2022). GDD series were then correlated with the growth series.

The results showed that the T-linked method detected many more response signals than GDD and thermal growing season length approaches (~90% versus ~50%). More importantly, using the T-linked method provides a much more detailed information on the intra-annual variation of growth-temperature response patterns. This is particularly important for larch species, as previous studies have reported that larch species may show opposite response patterns to temperature during the phase of rebuilding their crowns (usually the early growing season) and during the active period of cambium (usually the middle growing season) (Li et al., 2019; Zhang et al., 2021; Zhou et al., 2021). This conclusion does not apply to the extremely cold subarctic regions, where low temperature plays a dominant role for tree growth. However, for other distribution areas of boreal larch, analyzing only the overall impact of climate change on tree growth may lead to the relatively weak response signals being covered by the relatively strong response signals. Effective detection of the intra-annual changes in growth-temperature response patterns is critical for fully understanding how climate change affects Eurasian boreal larch forests, especially for the larch populations at relatively lower elevations or latitudes that are more likely to show hybrid intra-annual growth-temperature response patterns.

While correlating growth series with GDD or growing season length can detect overall response patterns of tree growth to hydrothermal conditions throughout the entire growing season, some response higher frequency or variability signals may be remaining hidden. By contrast, the temporally-flexible T-linked method, clearly found emerging negative effects and diminishing benefits from warming across Eurasian boreal larch forests, reminding us that we also need to pay attention to those populations on which the net effects of warming remain positive. These points have been emphasized in the manuscript now in **Lines 383-386**, as below:

“Although cold temperature will most likely remain the main climatic limitation to most of Eurasian boreal forests for a long time, we clearly found emerging negative effects and diminishing warming-induced benefits, indicating the upcoming ‘temperature tipping point’ (Duffy et al., 2021).”

References:

- Fu, Y.H. et al., 2014. Unexpected role of winter precipitation in determining heat requirement for spring vegetation green-up at northern middle and high latitudes. *Global Change Biology*, 20(12): 3743-3755.
- Gao, S. et al., 2022. An earlier start of the thermal growing season enhances tree growth in cold humid areas but not in dry areas. *Nature Ecology & Evolution*, 6(4): 397-404.
- Li, W. et al., 2019. Diverse responses of radial growth to climate across the southern part of the Asian boreal forests in northeast China. *Forest Ecology and Management*, 458: 117759.
- Zhang, Y. et al., 2021. Higher plasticity of water uptake in spruce than larch in an alpine habitat of North-Central China. *Agricultural and Forest Meteorology*, 311: 108696.
- Zhou, P. et al., 2021. Radial growth of *Larix sibirica* was more sensitive to climate at low than high altitudes in the Altai Mountains, China. *Agricultural and Forest Meteorology*, 304-305: 108392.

I am concerned (and a little unclear) about what “warming threatened” means and how it should be interpreted. As I understand it, warming threatened means that at some point during the growing season, you identified a cluster of t-linked values that were correlated with declines in detrended ring widths regardless of how small the effect was or the nature of that cluster of t-linked values (spring, summer, long, or short). Is this correct? I think this should be made more clear in the main text. Related to this, if that is at least close to correct, the manuscript reads as if “warming threatened” (which really means growth declines somewhat with warm temperatures) is driving a perception of mortality or unsuitable habitat when looking at the maps and projections. Also, your work is on detrended growth which is often only capturing about 20% of the actual width of the ring. I suggest further clarifying (or redefining) what it means to be warming threatened and potentially setting a higher bar for what constitutes as “threatened” or just avoiding that term and sticking more to the data about correlations and growth.

-- Thank you for your comment. We agree with the reviewer’s assessment that the term “warming threatened” for populations showing some negative responses to temperature may be confusing. We have changed it throughout the manuscript to “populations showing negative responses to temperature” as suggested by the reviewer.

Lines 603-614, Methods section: “We divided the 30-yr population chronologies sensitive to temperature into two groups based on whether they showed negative responses to temperature: (i) positively-responding group, composed of chronologies showing significant positive responses to temperature but no negative responses; and (ii) negatively-responding group, composed of those showing significant negative responses to temperature, regardless of showing or not showing positive responses. The positively-responding group refers to populations where growth was expected to consistently benefit from increasing temperature, while the negatively-responding group refers to populations where growth would be threatened by high temperatures during portions of the growing season, regardless of the net effects of warming on growth. We located all 30-yr population chronologies in climate space based on their local climatic conditions over the corresponding time periods to demonstrate the climatic differentiation of negative and positive population-level temperature sensitivity.”

I am a little concerned about the use of area in some of these figures (ex figures 5 and 6) when we know the distribution of larch is not continuous across this area. Are there no good models available for larch realized distributions within this space that you can rely on? Or at the very least, compare with your area-based estimates?

-- Thank you for your comment. The distribution areas of the two larch species we currently used were digitized and merged from multiple atlases (Fang, Wang, & Tang, 2011; Osawa et al., 2010; Sokolov, Svjseva, & Kubli, 1977), which draw the distribution areas of the two larch species in different countries according to both the realized and theoretical distribution areas. Therefore, we actually are using the potential distributions of Eurasian boreal larch based on existing data and expert assessment, so even if they are not totally present there, our models do apply, since these areas can potentially sustain these species.

However, as the reviewer said, the distribution of boreal larch is not continuous across this area. We agree with the reviewer that the direct comparison of area percentages is not appropriate. Also, we did not consider the tree density. However, we did not find suitable realized distribution areas for the two larch species. Therefore, in the revision, we have removed the comparisons of area percentages and we also deleted the Figure 6 in the initial submission. We have also reduced the contents of the initial Figure 5 (which is Figure 4 in our revision, please see below) in the main text, and put these reduced contents to the supplementary information (supplementary Fig. 9) for reference. We retained the use of the theoretical distribution area of boreal larch for representing the projected response pattern of larch growth to temperature under different projection scenarios. Considering that this area is the theoretical distribution area of boreal larch and the possible changes of the realized distribution driven by climate change, we consider this reservation to be meaningful.

Fig. 4| Projected distributions of the positively-responding and negatively-responding regions under Shared Socio-economic Pathways (SSP) 2-45 and 5-85. a-d. Results of Siberian larch and Dahurian larch identified by probability thresholds (P_0) of 0.50 and 0.95, respectively. Light yellow to dark green represents the decreasing proportion of the climate projections from 25 GCMs that identified the negatively-responding regions under corresponding projection scenarios; black lines represent the baseline (1970-2000) boundaries between the positively-responding and negatively-responding regions with corresponding probability thresholds. Grey shadows represent the distribution areas where projected climatic conditions falling outside the baseline climate space of boreal larch.

Minor comments:

Line 37 and 41: I am not sure “expand” and “retreat” are the best terms here. This phrasing invokes the population moving instead of existing populations having a change in their growth-climate

relationships.

-- Thank you. We agree these terms may bring these connotations that are not really reflective of our results. We have rewritten the abstract, and the new text in our revision reads:

Lines 32-34: *“Models quantifying these results project that the negative responses of growth to temperature will spread northward and upward throughout this century.”*

Note that the abstract has been shortened to 150 words to meet the journal's requirement, which brings about other changes. See our new abstract below:

“Larch, a widely distributed tree in boreal Eurasia, is experiencing rapid warming across much of its distribution. A comprehensive assessment of growth to warming is needed to comprehend potential impact of climate change. Most studies, relying on rigid calendar-based temperature series, have detected monotonic responses at margins of boreal Eurasia, but not across the region. We developed a novel method for constructing temporally-flexible and physiologically-relevant temperature series to reassess growth-temperature relations of larch across boreal Eurasia. Our method appears more effective in assessing the impact of warming to growth. Our approach indicates a widespread and spatially heterogeneous growth-temperature responses that is driven by local climate. Models quantifying these results project that the negative responses of growth to temperature will spread northward and upward throughout this century. If true, the risks of warming to boreal Eurasia could be more widespread than conveyed from previous works.”

Line 139 “Orange shadow”

-- Thank you. We have corrected the text in **Line 138**.

Figure 1. Panel D seems to have site ID labels?

Figures often have scientific names but text refers to common names. Consider revising for consistency and readability.

-- Thank you. We have deleted the ID labels from this figure as suggested by the reviewer as below:

Line 229: Is “high moisture deficits” the correct way to phrase this? Isn’t it really showing the proportional area that is more generally threatened by warming? And we presume it is moisture deficit that is driving it?

-- Thank you. Yes, we presumed that negative responses of tree growth to temperature were primarily driven by warming-induced increasing moisture deficits, but we agree this presumption may be too speculative, as the reviewer points out, we have changed the sentence. The new text reads:

Lines 245-248: “Estimations using the loose probability threshold of 0.50 suggested that >50% of the distributions of Siberian larch and Dahurian larch could be **increasingly stressed at high temperature** during portions of the growing season from 1970-2000 (Fig. 3d-e).”

Figure 5. The caption seems to say the green represents the warming threatened regions but it actually looks like the opposite? If the scale is the same as Figure 6, just add that scale.

-- We have modified the ambiguous caption and description to make them clearer in the revision. The initial Figure 5 now is Figure 4 in our revision; the new caption reads:

Lines 285-292: “Fig. 4| Projected distributions of the positively-responding and negatively-responding regions under Shared Socio-economic Pathways (SSP) 2-45 and 5-85. a-d. Results of Siberian larch and Dahurian larch identified by probability thresholds (P_0) of 0.50 and 0.95, respectively. Light yellow to dark green represents the decreasing proportion of the climate projections from 25 GCMs that identified the negatively-responding regions under corresponding projection scenarios; black lines represent the baseline (1970-2000) boundaries between the positively-responding and negatively-responding regions with corresponding probability thresholds. Grey shadows represent the distribution areas where projected climatic conditions falling outside the baseline climate space of boreal larch.”

Figures 5 and 6 are pretty repetitive and I think show more detail than is needed for the main text. It is hard to pull out interesting patterns other than “decreasing proportion of green” and “a lot vs. a little” between the different scenarios. Consider reducing the data presented here and perhaps combining figures 5 and 6.

-- Thank you for your comment. As described above, in the revision, we have removed the comparisons of area percentages and deleted the initial Figure 6. We have also reduced the contents of the initial Figure 5 in the main text, which now is **Figure 4** (please see above), and put these reduced contents to the supplementary information as Supplementary Fig. 9.

Line 310-312: rephrase. Are you arguing they are more ecologically relevant than temperate forests?

-- We apologize for the confusion. We are definitely not arguing that boreal forests are more ecologically relevant than temperate forests. What we intended to say is that Eurasian boreal forests have equal ecological relevance compared to those in Western Europe and North America,

yet, they have received much less scientific attention. We have rephrased the text for clarity, which now reads:

Lines 294-296, Discussion section: “Despite the wide distribution and great ecological relevance of Eurasian boreal forests (Watts et al., 2023), they have received much less scientific attention than boreal forests in western Europe and North America.”

Line 345: clarify where this idea of monolithic response comes from and what you mean? I think instead your work points to a more intuitive and physiologically relevant response (once you better articulate the physiological relevance, see below).

-- Thank you for your comment. What we intended to convey here was that multiple previous studies (Lloyd and Bunn, 2007; Myneni et al., 1997; Piao et al., 2014; Zhu et al., 2016) have reported that vegetation growth across the vast boreal Eurasia show general and monotonic positive response to increasing temperatures, whereas using more flexible and physiologically-relevant temperature series, we found undetected negative responses of tree growth to increasing temperature. Our findings are in line with multiple recent studies which report weakening low-temperature control and emerging negative impacts of warming, and project the reversal of warming-enhanced vegetation productivity in boreal regions (Piao et al., 2014; Wang et al., 2018; Zhang et al., 2022). In the revision, we have revised this sentence and added citations to make clearer as below:

Lines 342-344, Discussion section: “Our findings also challenge the idea of a monolithic positive response pattern to warming across boreal forests (Myneni et al., 1997; Zhu et al., 2016) and warn of the risks behind the warming-induced benefits over Eurasian boreal forests.”

References:

- Lloyd, A.H. and Bunn, A.G., 2007. Responses of the circumpolar boreal forest to 20th century climate variability. *Environmental Research Letters*, 2(4): 045013.
- Myneni, R.B., Keeling, C.D., Tucker, C.J., Asrar, G. and Nemani, R.R., 1997. Increased plant growth in the northern high latitudes from 1981 to 1991. *Nature*, 386(6626): 698-702.
- Piao, S. et al., 2014. Evidence for a weakening relationship between interannual temperature variability and northern vegetation activity. *Nature Communications*, 5(1): 5018.
- Wang, T. et al., 2018. Emerging negative impact of warming on summer carbon uptake in northern ecosystems. *Nature Communications*, 9(1): 5391.
- Zhang, Y. et al., 2022. Future reversal of warming-enhanced vegetation productivity in the Northern Hemisphere. *Nature Climate Change*, 12(6): 581-586.
- Zhu, Z. et al., 2016. Greening of the Earth and its drivers. *Nature Climate Change*, 6(8): 791-796.

Line 399: this paragraph could be removed or should provide more concrete next steps for “complimenting radial growth” and relevant citations. For example, what type of models are needed? Here you use detrended radial growth – is that the way forward or do we need process based models etc?

-- Thank you. We have rephrased this sentence and added the relevant citation for it in the revision

as below:

Lines 405-408, Discussion section: “Complementing tree-ring data with other multi-scale ecological information from forest inventories or remotely-sensed observations will allow us to better understand the relevance and consequences of the growth-temperature response patterns for the real functioning of the forests (Babst et al., 2018).”

Line 420-424: I would argue your method as implications beyond just boreal larch populations and think you should broaden this statement here.

-- We thank the reviewer for the recognition of our work, we have revised the text as suggested by the reviewer to broaden the statement to boreal ecosystems in the revision. The new text reads:

Lines 424-427, Discussion section: “Our study also indicates that using calendar-based approaches likely underestimate the temperature sensitivity of boreal forests. This level of poor estimation likely has important consequences for the conservation and survival of boreal ecosystems under rapidly changing climate.”

Line 454-457: unclear how these met station data were used to generate climate data for each population. I see it later on line 533 but a half sentence here would help too.

-- Thank you for your suggestion. We have added new text here describing how we match the tree-ring data with the climate data from meteorological stations in the revision.

Lines 494-495, Methods section: “...matched with the sampling populations according to the principle of proximity.”

Line 557-567: It is also quite common to have lagged effects: climate from one year impacting growth the following year. Why was that not considered in this study?

-- We fully agree with the reviewer on the importance of lagged effects of climate change on tree growth. However, we decided to not include these in our study for the following reasons:

(i) The focus of this study is to investigate the spatial heterogeneity of growth-temperature response pattern during the growing season across Eurasian boreal larch forests by optimizing the construction method of temperature time series correlated with tree-ring growth series. One of the key points we aim to focus is that avoiding rigid calendar periods can better match with tree growth mechanisms. Expanding this approach to include previous growing seasons would overcomplicate and dilute this goal and may represent a number of very different conceptual and methodological challenges that are best suited for further studies, once the set value of the T-linked approached is establish for understanding current growing season growth.

(ii) For boreal regions, the lagged effects of drought or precipitation on tree growth seems to be more pronounced than that of temperature (Dial et al., 2022; Kannenberg et al., 2020; Vaganov et

al., 1999).

Also, previous studies have reported that the lagged effects seem to be more pronounced on evergreen trees than on deciduous trees (Anderegg et al., 2015; DeSoto et al., 2020; Wolfe et al., 2016). Given this, we considered that the limitation of not considering the lagged effects to be less influential in the boreal conifer forests we focus on.

Nonetheless, despite not being the core point of this study, we agree on the fundamental point of the reviewer and we are working towards continuing advancing to introduce this effect in the future. In the revised text, we have made clearer the limitation of our study in this regard, in the Discussion section:

Lines 408-412, Discussion section: *“Incorporating the ‘legacy effects’ of climate (Kannenberg et al., 2020), something not entirely incorporated in vegetation models, but omnipresent in trees, into analysis is also necessary for fully assessing the impacts of warming on forests.”*

References:

- Anderegg, W.R.L. et al., 2015. Pervasive drought legacies in forest ecosystems and their implications for carbon cycle models. *Science*, 349(6247): 528-532.
- DeSoto, L. et al., 2020. Low growth resilience to drought is related to future mortality risk in trees. *Nature Communications*, 11(1): 545.
- Dial, R.J., Maher, C.T., Hewitt, R.E. and Sullivan, P.F., 2022. Sufficient conditions for rapid range expansion of a boreal conifer. *Nature*, 608(7923): 546-551.
- Kannenberg, S.A., Schwalm, C.R. and Anderegg, W.R.L., 2020. Ghosts of the past: how drought legacy effects shape forest functioning and carbon cycling. *Ecology Letters*, 23(5): 891-901.
- Vaganov, E.A., Hughes, M.K., Kirilyanov, A.V., Schweingruber, F.H. and Silkin, P.P., 1999. Influence of snowfall and melt timing on tree growth in subarctic Eurasia. *Nature*, 400(6740): 149-151.
- Wolfe, B.T., Sperry, J.S. and Kursar, T.A., 2016. Does leaf shedding protect stems from cavitation during seasonal droughts? A test of the hydraulic fuse hypothesis. *New Phytologist*, 212(4): 1007-1018.

Line 557-564: *Is there no effect of cold temperatures that should be considered?*

-- Thank you for the question. Cold temperature is the dominant limiting factor for tree growth in a large part of Eurasian boreal larch forests, it is therefore also a factor that must be considered in the study of growth-temperature response patterns in this area. There exists a relation of "as one falls, another rises" between the positive and negative effects of temperature on trees, which can be considered as two sides of the same coin. Showing positive responses to temperature can be considered as tree growth being limited by cold temperature, or as tree growth would benefit from warming. Therefore, the positive correlations that we calculated and found are the impact of cold temperatures, and they have been quantified and considered in our study. We also stated that *“Although cold temperature will most likely remain the main climatic limitation to most of Eurasian boreal forests for a long time, we clearly found emerging negative effects and diminishing warming-induced benefits, indicating the upcoming ‘temperature tipping point’ (Duffy et al., 2021).”* in the

Discussion section (Lines 383-386). We have revised this paragraph following the reviewer's previous suggestion to make it reflects the growth-temperature response patterns more directly and accurately.

The use of red and blue seems counterintuitive to me. Often red indicates threatened situations but here you use them opposite. Perhaps a different color scheme makes more sense?

-- Thank you. We have replotted the figures and reversed the color scheme, for example:

Fig. 3| Baseline (1979-2000) probability of showing negative growth-temperature responses across Eurasian boreal larch forests. a-b. Circles represent the sampling populations. The left and right halves of each circle are colored according to the population-level growth-temperature response patterns of the represented population during 1960-1990 and 1970-2000, respectively, where red and blue represent positively-responding and negatively-responding, dark grey represents no significant response, and no color represents lack of growth data or climate data. Red-white-blue gradient across the species distributions represents the estimated probability of showing negative responses decreasing from 1 to 0, species-specific probability functions were labeled on each plot; grey lines represent the 0.50, 0.75 and 0.95 iso-probability lines. Inset **c** shows dense sampling populations in the west Altai Mountains. Insets **d-e** display the area percentage histograms of positively-responding and negatively-responding regions in species distribution identified by three probability thresholds. **f-g.** Elevation profiles of the positively-responding (blue) and negatively-responding regions (red) identified by the probability threshold of 0.50 at a latitudinal resolution of 2.5'. Shadows and lines are related to the range and average of elevation, respectively. Green shadow and line represent the range and average of elevation of the 0.50 iso-probability line.

At many points you refer to the t-linked method as “physiologically-informed” however that physiology was never discussed beyond just leaf phenology and short growing seasons. Is that all you mean? If so, this phrase seems to be a bit of a stretch.

-- Thank you for your comment. We consider that the flexibility of the T-linked method allows it to capture much more realistically the species physiology, but we agree with the reviewer that this is not actually the best possible expression and it can induce to error. Therefore, we have replaced “physiologically-informed” with “physiologically-relevant”, which we hope captures the meaning of the short and highly variable growing season conditions in boreal regions, which are difficult to capture using rigid calendar-based periods.

With the development of this new t-linked approach, why are you not also considering a more robust precipitation metric? There are obviously water balance and drought indices that do this but could your new method provide an alternative way to think about this?

-- Thank you for your suggestion. Precipitation is obviously an important indicator. We have also tried to aggregate the precipitation over the T-linked periods to generate T-linked precipitation series. However, for the following three reasons, we have not taken precipitation as the climate factor with the highest priority for this study.

(i) The T-linked method is based on continuous daily records of climatic factor, however, daily records for precipitation are severely missing across the study area, limiting the attempt to construct T-linked precipitation series. The T-linked method can be used in future research which with well-recorded daily precipitation.

(ii) In addition, due to the extreme cold temperature in boreal regions, the effects of immediate precipitation on tree growth are generally lagged. The moisture demand of tree growth during the growing season is more supplied by snowpack meltwater, which is directly controlled by temperature to a large extent (Dial et al., 2022; Sniderhan and Baltzer, 2016; Vaganov et al., 1999). Therefore, it may not be appropriate to correlate tree growth with precipitation during temporally-flexible T-linked periods.

(iii) It is well known that severe cold is the main limiting factor to forest productivity for most of our study area. We also can see that boreal Eurasia has experienced rapid warming above the global average over the past half century, while precipitation did not show a significant trend. Previous studies have reported that the warming-induced alleviation of heat deficit has promoted productivity in our study area, but the opposite is true in the relatively warmer and drier south, where warming-induced increases in forest mortality have been reported. In any case, temperature seems to be a more prioritized climatic factor in the context of global warming for boreal Eurasia.

Nonetheless, we considered the precipitation in subsequent analyses of this study, we established the temperature-precipitation climate space, and established the models to quantify the relationship between growth-temperature response pattern and local mean annual temperature and mean annual precipitation. We also look forward to finding an appropriate way to incorporate precipitation

or water conditions into T-linked method in future studies.

References:

- Dial, R.J., Maher, C.T., Hewitt, R.E. and Sullivan, P.F., 2022. Sufficient conditions for rapid range expansion of a boreal conifer. *Nature*, 608(7923): 546-551.
- Sniderhan, A.E. and Baltzer, J.L., 2016. Growth dynamics of black spruce (*Picea mariana*) in a rapidly thawing discontinuous permafrost peatland. *Journal of Geophysical Research: Biogeosciences*, 121(12): 2988-3000.
- Vaganov, E.A., Hughes, M.K., Kirilyanov, A.V., Schweingruber, F.H. and Silkin, P.P., 1999. Influence of snowfall and melt timing on tree growth in subarctic Eurasia. *Nature*, 400(6740): 149-151.

Reviewer #3 (Remarks to the Author):

Summary

This manuscript focused on the boreal larch forests of Eurasia. In these forests, the relationships between tree growth and climate were studied. To carry out this analysis, the authors developed a new method for correlating tree growth with growing season temperature. These new correlations revealed that a much higher proportion of larch forests may be responding negatively to growing season temperature than was previously thought. Using the observed correlations, the authors projected how larch forest tree growth would respond to future climate change. A main conclusion was that increased temperatures pose a significant and overlooked risk to Eurasian larch forests.

I think this is an important study with potentially strong impact. My opinion is that larch forests are probably understudied, and this work should stimulate interest and inspire new work. The T-based method is novel and interesting, though I think it requires better motivation and more validation, as described below.

We thank the reviewer for the detailed and constructive comments on our manuscript. We would also like to thank the reviewer for the recognition of our work. The reviewer raised several major concerns and a series of specific comments, which are very constructive for improving our work, and we have responded to these concerns in the revision and in our responses to the reviewer's specific comments.

The reviewer's primary concerns were briefly summarized as 1) the potential spurious correlation of the T-linked method, 2) clarification for the motivation of the study. 3) lack of mechanisms and caveats in the Discussion section. For the first concern, as suggested by the reviewer, we conducted 10-fold cross-validation analysis on the logistic models estimated based on the correlation results using T-linked method to test the robustness of our results. For the second concern, we have substantially revised the text and supplemented more explanations and supports for our study motivation. For the third concern, we have substantially revised the Discussion section as suggested. This section makes now hopefully clear the potential caveats to our analyses and adds depth to the discussions of mechanisms by which temperature affects tree growth.

In response to the concerns and comments, we have performed additional analyses and substantially revised the manuscript. These supplementary analyzes and corresponding results have been described and provided in detail below. Overall, the reviewer's comments led to a much-improved manuscript.

Please find below our point-by-point responses to each of the reviewer's comments.

Motivation for the Study

I reacted with skepticism to the argument in ll. 99-120, which is one of the major motivations for this study. It would help if the authors could do more to convince me that “the same calendar period can correspond to very different growth phases from year to year”. What calendar period is typically used? What growth phases are the authors talking about? Can the authors provide a specific example to illustrate the point?

-- Thank you very much, indeed this is a critical point to make clear in our work. For detecting growth responses to temperature, tree ring-based studies typically correlate annual tree-ring growth series with temperature series using fixed calendar periods (weeks, months, and/or seasons). For example, growth series is correlated with time series of April to September temperatures. By contrast, xylem phenology is known to be mainly regulated by temperature cues (Rossi et al., 2016; Rossi et al., 2008; Zhang et al., 2018), rather than fixed calendar dates. For instance, Zhang et al. (2018) suggested that the onset of wood formation of *Juniperus przewalskii* corresponds to a change of 14.1 days C⁻¹. For forests under continuous climatic stress or in regions with consistent and long growing seasons, monthly or seasonal temperature series are relatively effective to inform growth-temperature relations. However, for forests with comparably short and variable growth-temperature response windows, fixed-date/calendar-based approaches risk missing significant growth-climate responses (Vaganov et al., 1999).

This is indeed the case for boreal Eurasia, where seasonal dynamics of temperature vary to a great extent from year to year. For example, the date after which temperature stays consistently above 0 °C can differ by up to 25-30 days (Vaganov et al., 1999). Boreal growing seasons, especially near the Arctic Circle, are often shorter than two months and their starting and ending dates vary greatly between years (Seftigen et al., 2018). In addition, larches are deciduous conifers that need to renew their needles at the beginning of each growing season, making the initial period critical for the species' annual growth. Leafing limits xylem formation in both rate and duration (Rossi et al., 2009), further compressing growth response windows.

Therefore, we argue that correlating larch tree-ring parameters with fixed-date monthly or seasonal temperatures series will be less than ideal outcomes where, in some years, tree-ring data is compared with the temperature of a period when there is no growth and, for other years, when growth is in an active or past-peak phase (Vaganov et al., 1999). Tracheid development curves for Dahurian larch in Tura support this finding, showing sharp interannual differences in growth and end of tracheid development separated by at least half a month later between years (August 5th vs. July 15th, Osawa et al., 2010). Correlating annual radial growth series of these trees with July temperatures would draw inconsistent outcomes between years. Other work has shown consistent patterns in other coniferous species (*Juniperus przewalskii*, Zhang et al., 2018).

We hope this clarifies our position when we state that “the same calendar period can correspond to very different growth phases from year to year due to the great inter-annual variations in seasonal dynamics of temperature in boreal regions, therefore, correlating annual growth with temperature series for fixed calendar periods may result in misleading estimations of growth-temperature

response.” In the revised version, we have rewritten this part to make express our motivations and justifications to develop the T-linked method as a way to address this shortcoming of common approaches more directly and accurately. The new text reads:

Lines 92-113, Introduction section: “*Variation in tree growth is usually related to climatic variation. However, identifying the climate factors and time periods over which climate affects tree growth has been problematic (van de Pol et al., 2016). General methods for detecting growth-temperature responses typically correlate annual tree growth with temperature series for rigid calendar periods (months, seasons, and/or annual), meaning that choice of time periods is fixed, arbitrary, and might not follow what the physiology of trees require for growth. Since xylem phenology is mainly regulated by temperature cues (Huang et al., 2023; Rossi et al., 2016; Rossi et al., 2008), the same calendar period can correspond to very different growth phases from year to year due to the great inter-annual variations in seasonal dynamics of temperature in boreal regions (Osawa et al., 2010; Vaganov et al., 1999; Zhang et al., 2018), therefore, correlating annual growth with temperatures for fixed periods may result in underestimations of growth-temperature response. For trees living under continuous climatic stress or in regions with consistent growing seasons, monthly or seasonal temperature series can be relatively effective to investigate growth-temperature relations. However, for trees living in regions with comparably variable climates, it is not likely appropriate to conclude that trees are insensitive to temperature when climate series are based solely on a human calendar. Boreal growing seasons, especially near the Arctic Circle, are often shorter than two months and their starting and ending dates vary greatly between years (Osawa et al., 2010; Seftigen et al., 2018). In addition, larches dominant in boreal Eurasia are deciduous conifers that need to renew their needles at the beginning of each growing season. This trait limits xylem formation in both rate and duration (Rossi et al., 2009), further compressing growth response windows and confusing the detection of response signals using calendar-based approach.*”

References:

- Carrer, M., Castagneri, D., Prendin, A.L., Petit, G. and von Arx, G., 2017. Retrospective analysis of wood anatomical traits reveals a recent extension in tree cambial activity in two high-elevation conifers. *Frontiers in Plant Science*, 8: 737.
- Fonti, P. et al., 2013. Temperature-induced responses of xylem structure of *Larix sibirica* (Pinaceae) from the Russian Altay. *American Journal of Botany*, 100(7): 1332-1343.
- Jevšenak, J., 2019. Daily climate data reveal stronger climate-growth relationships for an extended European tree-ring network. *Quaternary Science Reviews*, 221: 105868.
- Jevšenak, J. and Levanič, T., 2018. dendroTools: R package for studying linear and nonlinear responses between tree-rings and daily environmental data. *Dendrochronologia*, 48: 32-39.
- Osawa, A., Zyryanova, O.A., Matsuura, Y., Kajimoto, T. and Wein, R.W., 2010. Permafrost ecosystems: Siberian larch forests, 209. Springer Science & Business Media, New York.
- Rossi, S. et al., 2016. Pattern of xylem phenology in conifers of cold ecosystems at the Northern Hemisphere. *Global Change Biology*, 22(11): 3804-3813.
- Rossi, S. et al., 2008. Critical temperatures for xylogenesis in conifers of cold climates. *Global Ecology and Biogeography*, 17(6): 696-707.
- Rossi, S., Rathgeber, C.B.K. and Deslauriers, A., 2009. Comparing needle and shoot phenology with xylem development on three conifer species in Italy. *Annals of Forest Science*, 66(2): 206-

206.

- Seftigen, K., Frank, D.C., Björklund, J., Babst, F. and Poulter, B., 2018. The climatic drivers of normalized difference vegetation index and tree-ring-based estimates of forest productivity are spatially coherent but temporally decoupled in Northern Hemispheric forests. *Global Ecology and Biogeography*, 27(11): 1352-1365.
- Vaganov, E.A., Hughes, M.K., Kirdyanov, A.V., Schweingruber, F.H. and Silkin, P.P., 1999. Influence of snowfall and melt timing on tree growth in subarctic Eurasia. *Nature*, 400(6740): 149-151.
- van de Pol, M. et al., 2016. Identifying the best climatic predictors in ecology and evolution. *Methods in Ecology and Evolution*, 7(10): 1246-1257.
- Zhang, J. et al., 2018. Cambial phenology in *Juniperus przewalskii* along different altitudinal gradients in a cold and arid region. *Tree Physiology*, 38(6): 840.

The authors justify their study, in part, with the sentence on ll. 67-69: "The wide distributions would imply high spatial heterogeneity of growth responses...". More justification for this statement would be helpful. The authors do cite a book, but I don't have access to that book, so it is difficult for me to evaluate the authors' claim.

-- Thank you for your comment. We now include new references (Babst et al., 2019; Babst et al., 2013; Charney et al., 2016; Hellmann et al., 2016) to support this claim (**Line 58**). These studies investigate responses of tree growth to climate over continental to global spatial scales, and demonstrate highly spatially varying growth-climate response patterns. They argue that wide distributions imply highly spatially varying abiotic environmental conditions, thus varying response pattern to climate across a tree species' populations (Babst et al., 2018).

References:

- Babst, F. et al., 2013. Site- and species-specific responses of forest growth to climate across the European continent. *Global Ecology and Biogeography*, 22(6): 706-717.
- Babst, F. et al., 2018. When tree rings go global: Challenges and opportunities for retro- and prospective insight. *Quaternary Science Reviews*, 197: 1-20.
- Babst, F. et al., 2019. Twentieth century redistribution in climatic drivers of global tree growth. *Science Advances*, 5(1): eaat4313.
- Charney, N.D. et al., 2016. Observed forest sensitivity to climate implies large changes in 21st century North American forest growth. *Ecology Letters*, 19(9): 1119-28.
- Hellmann, L. et al., 2016. Diverse growth trends and climate responses across Eurasia's boreal forest. *Environmental Research Letters*, 11(7): 1-12.

Further justification for the study comes in ll. 91-96, where the authors cite a contrast between tree ring and satellite data. I think this contrast is overplayed. The tree ring datasets provide information on tree diameter growth. The satellite data definitely do NOT provide information on tree diameter growth. For example, Berner and Goetz study vegetation greenness. Thus, I don't see how direct comparisons and contrasts are possible.

-- Thank you. We understand that both the satellite-derived data and tree-ring data are commonly-used tree growth proxies, while the former represents the canopy growth of trees and the latter

represents the diameter growth of trees. Photosynthetic activity determines the amount of carbohydrates available for physiological processes, including wood formation and canopy formation, which is the physiological basis of the relationship between radial growth and canopy growth (Decuyper et al., 2020; Wen et al., 2022). Climate should affect both the canopy and radial growth of trees. However, radial and canopy growth of trees are often inconsistently related (Brehaut and Danby, 2018; Vicente-Serrano et al., 2016; Wen et al., 2022). In our study area, previous studies based on remote sensing data suggest that warming significantly impacts vegetation over large portions of boreal Eurasia (Peng et al., 2013; Piao et al., 2017), while tree-ring-based studies suggest that less than 50% of populations located in boreal Eurasia sensitive to temperature (Hellmann et al., 2016; Tei et al., 2017). The inconsistency in temperature sensitivity between radial growth and canopy growth can be attributed to the different demands of wood formation and leaf activity for climate conditions, where higher temperatures are required for wood formation than for leaf activity, compressing response windows of radial growth to temperature (Körner, 1998; Rossi et al., 2011). We argue that rather than radial growth being insensitive to temperature, its relatively weak temperature sensitivity has not been detected due to methodological limitations linked to rigid calendar-based approaches.

We consider that our T-linked method can potentially explain the seemingly inconsistent results shown by tree-ring-based radial growth and satellite-based canopy growth. We have rewritten this part to make these points clearer and have included new references to support them, which now reads:

Lines 84-89, Introduction section: “*By contrast, studies using satellite-derived tree canopy growth data suggests that warming significantly impacts a large portion of Eurasian boreal forests (Peng et al., 2013; Piao et al., 2014). Clearly, there is an important gap in knowledge between these two approaches over a biome that is important to the global carbon cycle, which could be attributed to the different requirements of wood formation and leaf activity for temperature (Körner, 1998; Rossi et al., 2011).*”

References:

- Brehaut, L. and Danby, R.K., 2018. Inconsistent relationships between annual tree ring-widths and satellite-measured NDVI in a mountainous subarctic environment. *Ecological Indicators*, 91: 698-711.
- Decuyper, M. et al., 2020. Spatio-temporal assessment of beech growth in relation to climate extremes in Slovenia – An integrated approach using remote sensing and tree-ring data. *Agricultural and Forest Meteorology*, 287: 107925.
- Hellmann, L. et al., 2016. Diverse growth trends and climate responses across Eurasia's boreal forest. *Environmental Research Letters*, 11(7): 1-12.
- Körner, C., 1998. A re-assessment of high elevation treeline positions and their explanation. *Oecologia*, 115(4): 445-459.
- Peng, S. et al., 2013. Asymmetric effects of daytime and night-time warming on Northern Hemisphere vegetation. *Nature*, 501(7465): 88-92.
- Piao, S. et al., 2017. Weakening temperature control on the interannual variations of spring carbon uptake across northern lands. *Nature Climate Change*, 7(5): 359.

- Rossi, S., Morin, H., Deslauriers, A. and Plourde, P.-Y., 2011. Predicting xylem phenology in black spruce under climate warming. *Global Change Biology*, 17(1): 614-625.
- Tei, S. et al., 2017. Tree-ring analysis and modeling approaches yield contrary response of circumboreal forest productivity to climate change. *Global Change Biology*, 23(12): 5179-5188.
- Vicente-Serrano, S.M. et al., 2016. Diverse relationships between forest growth and the Normalized Difference Vegetation Index at a global scale. *Remote Sensing of Environment*, 187(C): 14-29.
- Wen, Y., Jiang, Y., Jiao, L., Hou, C. and Xu, H., 2022. Inconsistent relationships between tree ring width and normalized difference vegetation index in montane evergreen coniferous forests in arid regions. *Trees*, 36(1): 379-391.

Validation and Assessment of Results

Taking the results at face value, the T-linked approach seems to be an important advance over the conventional calendar-linked approach. However, I do worry about the possibility of spurious correlations. As I understand it, the T-linked approach involves the determination of two dates based on two temperature values (ll. 487-489). The temperature in this correlation window is then correlated to growth, and statistical significance is assessed. This procedure is carried out many times, for many correlation windows. Because many correlation windows are tested, it seems inevitable to me that false positives will creep in to the analysis. To make a more convincing case that the analysis is robust, the authors might consider doing some cross-validation or other sort of out-of-sample testing. I think that such out-of-sample testing is especially important given that the authors are extrapolating their correlations to projected climates (Fig. 5).

-- Thank you for your suggestion. We understand the concern related with the high number of correlations. In the initial submission, we attempted to limit the influence of potential spurious correlations by raising the identification criteria for temperature sensitivity, that is, only the growth series showing significant correlations with >3 adjacent T-linked temperature series were identified as sensitive to temperature. The probability of finding spurious significant correlations with >3 adjacent T-linked temperature series is much lower, as these are likely randomly distributed. Raising the identification criteria for temperature sensitivity can greatly reduce the influences of the randomly-distributed spurious correlations on our main results.

We fully agree with the reviewer that we need to conduct cross-validation analysis on our results. We performed a 10-fold cross-validation to test the quality and accuracy of the estimated logistic models (Hastie et al., 2001) (Lines 657-665). The observations for parameter estimation were divided into 10 subsets that were roughly of equal size. Nine of these subsets were used as training group to estimate a cross-validated model while the 10th subset was used for testing. The estimated error, which is used to determine classification accuracy for a logistic model, was calculated by applying the cross-validated model to the test group. This procedure was executed 10 times, with each subset taking turns as the test group. The resulting 10 classification accuracies were averaged as an estimate of the overall predictive performance of the final model. The results show that the logistic models we established for both larch species have fairly high cross-validation accuracy and quality (observed accuracy = 0.862±0.064, Cohen's κ = 0.713±0.132 for Siberian

larch; observed accuracy = 0.885 ± 0.056 , Cohen's $\kappa = 0.706 \pm 0.131$ for Dahurian larch) (Lines 222-225). The results of 10-fold cross-validation indicated high robustness and accuracy of the models built based on the different subsets of the correlation results using T-linked method. We also found similarly strong validation of the estimated quantitative relationships between growth-temperature response pattern and local climate conditions are robust and physiologically reasonable.

We hope that the reviewer's concern is addressed through the supplementary analyses described above. We added the supplementary analyses as independent parts in the revision (Lines 222-225, 657-665), and added caveats in the Discussion section (Lines 310-316). The new text reads as below:

Lines 222-225, Results section: *"Our 10-fold cross-validation analysis indicated that the estimated models were robust and of fairly-high accuracy and quality (Siberian larch: observed accuracy = 0.862 ± 0.064 , Cohen's $\kappa = 0.713 \pm 0.132$; Dahurian larch: observed accuracy = 0.885 ± 0.056 , Cohen's $\kappa = 0.706 \pm 0.131$)."*

Lines 310-316, Discussion section: *"While there are potentially serious issues with conducting such a large number of calculations, we tested the risk of potential spurious correlations using the T-linked method through a fair comparison to calendar-based methods and suggested that the effectiveness of our method was not attributed to the increased number of calculations. Furthermore, the cross-validation analyses we conducted demonstrated fairly-high accuracy and robustness of our estimated models, suggesting limited influences of spurious correlations on the main results."*

Lines 657-665, Methods section: *"We tested the quality and accuracy of the estimated logistic models using 10-fold cross-validation analysis (Hastie et al., 2001). The observations for parameter estimation were divided into 10 subsets that were roughly of equal size. Nine of these subsets were used as training group to estimate a cross-validated model while the 10th subset was used for testing. We calculated the classification accuracy and the Cohen's kappa (κ) of the cross-validated model applied to the test group to measure the classification performance. This procedure was executed 10 times, with each subset taking turns as the test group. The resulting 10 classification accuracies and 10 Cohen's κ were averaged as estimates of the overall predictive performance of the final model"*

Another issue with the analysis pertains to the effect sizes. The Results showed the fraction of trees and populations with either a statistically significant temperature sensitivity (Table 1). However, I did not see the authors report the actual effect sizes anywhere. This absence is important. One could get statistically significant correlations ($p < 0.05$) even if the correlation coefficient itself is very small. Thus, there is the possibility of obtaining statistically significant but biologically meaningless correlations. To convince me this is not the case, it would be helpful to provide information on the correlation coefficients obtained from correlations between tree diameter growth and correlation window temperature.

-- Thank you for your comment. Since we correlated each growth series with a batch of T-linked

temperature series, we obtained many growth-temperature correlation coefficients for each sampling population, making it unpractical to provide all correlation coefficients directly. Therefore, we established four metrics derived from these correlations to measure the effect size of temperature on each population and the growth-temperature correlation of each population. We have rewritten the descriptions of the correlation-derived metrics in more detail as a separate subsection in the Methods section of the main text. See Lines 575-602, also briefly described below:

We established four metrics based on the growth-temperature correlation results to measure the size of temperature effects on tree growth and to characterize the structure of growth-temperature response pattern for each population from two dimensions:

(i) Affecting scope, i.e., what is the proportion of individual trees (individual tree chronologies or single standardized growth series) in a population that show a significant relation to temperature. We calculated the percentage of individuals in each population showing significant negative or positive correlations with each T-linked temperature series from the nearest meteorological station. The maximum value of these percentages was used as the first metric, that is, the maximum proportion of individuals in a population that show negative or positive sensitivity to temperature, representing the maximum scope of negative or positive temperature effects on each population.

(ii) Affecting duration, i.e., how long the growth response to temperature lasts. As the time period used to construct the same T-linked temperature series varies from year to year, we calculated the mean length (in the number of days) of the varying T-linked periods over corresponding analysis period for each T-linked temperature series. We calculated the average mean lengths of the T-linked temperature series which were significantly correlated with both population- and individual-level growth series. We suspect that the metrics established based on this attribute can be interpreted as a way of describing the mean duration of negative or positive responses of growth to temperature for a population.

We believe that these four metrics are meaningful and can well describe the size of the temperature effects on tree growth, and our results showed that these four metrics are all sensitive to the local climates of the population. Additionally, we also calculated the mean coefficient of all significant correlations between each population chronology and the T-linked temperature series from the nearest meteorological station to measure the size of temperature effects on tree growth. The four metrics and the mean correlation coefficient were correlated with the local climates (Lines 174-196).

Presentation

Lines 121-122 strike me as an unhelpful generalization. What specifically do the authors mean? Are there references to back up this point?

-- Thank you. We have deleted this sentence in the revision.

I struggled with the data presented in Table 1 and the related discussion in the text. For example, the first line in Table 1 lists the “Sensitive chron. pct.” as 57.6%. I further expected that all trees showing sensitivity would either have a positive sensitivity or a negative sensitivity. That is, I expected that “Sensitive chron. pct.” = “Positive sensitive chron pct” + “Negative sensitive chron pct”. But this is clearly not the case. The authors should better explain what is going on here.

-- We correlated each growth series (tree-ring chronology) with all T-linked temperature series from the nearest meteorological station, therefore, it is possible for the same growth series to show both negative and positive sensitivity to temperature, as shown in Supplementary Fig. 18 below. This is the reason why “Sensitive chron. pct.” > “Positive sensitive chron. pct.” + “Negative sensitive chron. pct.” in Table 1.

Tree growth can show opposite responses to temperature at different stages of the growing season. For example, previous studies have reported that larch species may show opposite responses to temperature during the phase of rebuilding their crowns (usually the early growing season) and during the active period of cambium (usually the middle growing season) (Li et al., 2019; Zhang et al., 2021; Zhou et al., 2021).

Supplementary Fig. 18| Correlation results between population-level growth series and the qualified T-linked temperature time series. The blue-white-red gradient represents correlation coefficients from positive to zero to negative. Label of ‘x’ denotes significant correlations ($p < 0.05$).

References:

Li, W. et al., 2019. Diverse responses of radial growth to climate across the southern part of the Asian boreal forests in northeast China. *Forest Ecology and Management*, 458: 117759.
 Zhang, Y. et al., 2021. Higher plasticity of water uptake in spruce than larch in an alpine habitat of North-Central China. *Agricultural and Forest Meteorology*, 311: 108696.

Zhou, P. et al., 2021. Radial growth of *Larix sibirica* was more sensitive to climate at low than high altitudes in the Altai Mountains, China. *Agricultural and Forest Meteorology*, 304-305: 108392.

Figure 2b: The various p -thresholds are confusing. Why not choose a single threshold? Or, even better, report the exact p value in each case.

-- Thank you for your comment. We agree and modified this figure to provide the exact p values as below. In the revision, this figure is moved to the Supplementary Information as Supplementary Fig. 7.

Supplementary Fig. 7 | Probability distributions of climatic factors and geographical coordinates grouped by populations showing positive (red) and negative (blue) temperature sensitivity. The distribution functions were estimated by kernel density; p -values of Student's t -test are noted on each panel; vertical dashed lines represent the average values of each factor.

Comparison of the 1960-1990 and 1970-2000 results is not that compelling because those two time periods mostly overlap. Why not compare 1960-1980 and 1980-2000?

-- Thank you for this question. We chose these time periods based on the time span of the tree-ring data we collected and the climatological normals we used.

In this study, we determined the local climatic conditions of each sampling population using the WorldClim datasets, which currently provides climate layers at a high spatial resolution of 2.5' that is aggregated over two time periods, 1960-1990 and 1970-2000. The high spatial resolution of this dataset helps us distinguish the local climatic conditions of the intensive sampling populations, which is critical for accurately quantifying the relationship between growth-temperature response pattern and local climate.

Therefore, we decided to use the two 30-year periods following a widely used value in climatology, which define climatological normal using 30-year historical averages (WMO Guidelines on the Calculation of Climate Normals, 2017). 20-year periods were considered to likely not be stable enough to provide meaningful results at our scale.

Also, the two time periods used fit well with the time span of our tree-ring data. As we point out in

Lines 556-561, “In terms of growth data, specifically, 258 of the 260 populations were sampled in or after 1990, 68.5% of the populations north of 60°N were sampled before 1995, and 77.9% of the populations south of 60°N were sampled after 1999 (Supplementary Fig. 14). Based on these, we conducted growth-temperature correlation analyses in parallel for both 1960-1990 and 1970-2000 to maximize the matching of growth data and local climatic conditions.”

Therefore, growth-temperature correlation analyses in parallel for both 1960-1990 and 1970-2000 will maximize the matching of our growth data and local climatic conditions. We do, however, look forward to the update and enrichment of tree-ring data across high latitudes in Eurasia so that we can better study the continuous changes of growth-climate response patterns over time.

We have written the manuscript to include this point within the Discussion section in Lines 398-400: “Understanding of Eurasian boreal larch forests can be further improved by increasing the density and spatial-temporal coverage of tree-ring collections (Zhao et al., 2019).”

Mechanisms and Caveats

The Discussion section of the paper struck me as narrow. I felt like the authors could have done more to discuss the mechanisms for the changes and the caveats to their analysis. For example, this study analyzed MAP but not VPD. I think this choice could be controversial. A number of recent studies argue for the importance of VPD in controlling boreal tree growth and mortality. Also, the correlations between temperature and tree growth may now be controlled (or could possibly be controlled in the future) by many mechanisms, like permafrost loss, increases in nitrogen mineralization, etc. I was surprised to see no discussion of these points. Neither was there much discussion of the mechanisms behind the observed north-south variation in the correlations. How might these spatial gradients be impacted by gradients in stand density, soils, etc.? Also, the climate projections were presented with few caveats. It is not at all clear to me that climate models are able to correctly simulate the observed interannual variation in boreal Eurasian climate, which is the underlying motivation for this study.

-- Thank you for your suggestion, we have substantially revised the Discussion section as suggested and include these points. This section makes now hopefully clear the potential caveats to our analyses in this regard and adds depth to the discussions of mechanisms by which temperature affects tree growth:

(1) We agree with the reviewer that VPD likely play an important role in controlling boreal tree growth and mortality (Berner and Goetz, 2022). VPD, though less available, may be a more effective index to measure regional moisture conditions for tree growth for cold boreal regions. In our analysis, we refer to Whittaker's biomes to use 30-year mean annual temperature and mean annual precipitation to establish climate space for displaying the climatic heterogeneity in growth-temperature response patterns of boreal larch. Nonetheless, we agree with the reviewer that incorporating VPD as a key climate index will be a great addition in future development of the model. We discuss this points in the revised Discussion section in Lines 410-413: “... and comprehensive

climatic variables such as **vapor pressure deficit** (Berner and Goetz, 2022) into analysis is also necessary for fully assessing the impacts of warming on forests. Introducing these factors is the natural next step to understand the consequences of rapid warming for the future of Eurasian boreal forests.”

(2) We have added the following points discussing on nitrogen mineralization and permafrost loss in **Lines 330-344**: “Rapid warming has led to radical changes in abiotic conditions across global boreal regions (Gauthier et al., 2015), affecting forest growth through multiple pathways, including rapid thawing of the permafrost, changed nutrient availability, and drastically altered hydrothermal conditions (Lim et al., 2018; Osawa et al., 2010; Sniderhan and Baltzer, 2016). Increasing soil temperature is expected to promote nitrogen mineralization through stimulating the humus decomposition (Lim et al., 2018; Rustad et al., 2001), consequently alleviating the prevailing nitrogen limitation in boreal forests (Du et al., 2020). Warming also extends the growing season (Gao et al., 2022) and ensures the snowpack meltwater for moisture demand during the growing season (Vaganov et al., 1999). Taken together, increasing temperature is expected to enhance boreal forest productivity (Piao et al., 2014; Zhu et al., 2016) and advance boreal forest into Arctic tundra (Berner and Goetz, 2022; Dial et al., 2022). However, previous studies indicate that such beneficial effects may be transient (D’Orangeville et al., 2018; Lim et al., 2018). Our findings also challenge the idea of a monolithic positive response pattern to warming across boreal forests (Myneni et al., 1997; Zhu et al., 2016) and warn of the risks behind the warming-induced benefits over Eurasian boreal forests.”

(3) Discuss how temperature affects tree growth. **Lines 386-388** now reads: “This shift could be attributed to increased evapotranspiration demand exceeding available moisture as temperature increases (Gauthier et al., 2015; Sniderhan and Baltzer, 2016).”

(4) Clarify the risk of potential spurious correlations within the T-linked method in **Lines 310-316**: “While there are potentially serious issues with conducting such a large number of calculations, we tested the risk of potential spurious correlations using the T-linked method through a fair comparison to calendar-based methods and suggested that the effectiveness of our method was not attributed to the increased number of calculations. Furthermore, the cross-validation analyses we conducted demonstrated fairly-high accuracy and robustness of our estimated models, suggesting limited influences of spurious correlations on the main results.”

(5) Make clear the uncertainties in extrapolating the models to climate projections. Added in **Lines 357-361**: “There are substantial uncertainties in extrapolating the models estimated based on current growth-temperature relations to climate projections. However, the consistent qualitative spatial trends revealed here suggests manager and policymakers need to consider the possibility of emerging negative impacts of warming across much more of the Eurasian boreal forests than previously estimated.”

(6) Introduce a sentence to make this caveat clear in **Lines 395-398**: “Interactions among nutrients, climate and forest growth are complex and nuanced (Dial et al., 2022; Lim et al., 2018). Further research is needed to estimate and verify the long-term implications of extensive changing climate

sensitivity in boreal Eurasia, including but not limited to mortality and demographic trends.”

(7) Explicitly addressing limitations and potential sensitivity bias of current tree-ring networks. **Lines 398-405:** *“Understanding of Eurasian boreal larch forests can be further improved by increasing the density and spatial-temporal coverage of tree-ring collections (Zhao et al., 2019). Central Siberia is noticeably bereft of populations for study. The existing tree-ring network also could be oversensitive to climate due to the current sampling bias (Babst et al., 2018; Klesse et al., 2018b), though see Nehrbass-Ahles et al. (2014) as a counterpoint that it might not be an overestimate. Such lack of spatial representation and potential climate sensitivity bias may result in contrasting response patterns detected with the actual situation.”*

References:

Berner, L.T. and Goetz, S.J., 2022. Satellite observations document trends consistent with a boreal forest biome shift. *Global Change Biology*, 28(10): 3275-3292.

REVIEWERS' COMMENTS

Reviewer #1 (Remarks to the Author):

Dear authors,

I enjoyed reading your revised manuscript, Reassessment of growth-climate relations indicates the potential for decline across Eurasian boreal larch forests. I have personally never seen such a comprehensive and effective effort by authors to revise their manuscript. I'm impressed and very satisfied with your work. The extensive revisions have strengthened an already very good paper into one that is even more robust and clear.

I only noticed a couple of minor things that need attention:

P13, lines 272-273: Red and blue are reversed here - red = negative responding, blue = positive responding.

P14, lines 285-292: The color scale could be a bit more clear. Green = less likely to be negative responding, correct? Also, the black line is very hard to see in the figures. Could the line be thicker, or perhaps a color? Maybe blue to tie in with the color scale used above?

Reviewer #2 (Remarks to the Author):

I previously reviewed this manuscript as Reviewer #2. Overall, I think the authors did a great job addressing my concerns with this manuscript. I have only a few small suggestions on phrasing.

Abstract: Should be "warming on growth"

Line 92: "Variation in tree growth is usually related to climatic variation" could be interpreted in a number of ways. There are lots of things that lead to variation in tree growth. I think here, you mean that studies often focus in on linking variation to climate though. Rephrase for clarity.

Line 192-196: these sentences are confusing to me. Please clarify.

Line 610. Still not happy with the use of the term "threatened" here. Please replace with something more like "where growth would decline with high temperatures"

REVIEWERS' COMMENTS

Reviewer #1 (Remarks to the Author):

Dear authors,

I enjoyed reading your revised manuscript, Reassessment of growth-climate relations indicates the potential for decline across Eurasian boreal larch forests. I have personally never seen such a comprehensive and effective effort by authors to revise their manuscript. I'm impressed and very satisfied with your work. The extensive revisions have strengthened an already very good paper into one that is even more robust and clear.

We sincerely thank you for your recognition of our work, and for your constructive and detailed comments, which have led to a much-improved manuscript.

I only noticed a couple of minor things that need attention:

P13, lines 272-273: Red and blue are reversed here - red = negative responding, blue = positive responding.

--Thank you, we have corrected the text.

P14, lines 285-292: The color scale could be a bit more clear. Green = less likely to be negative responding, correct? Also, the black line is very hard to see in the figures. Could the line be thicker, or perhaps a color? Maybe blue to tie in with the color scale used above?

--Thank you, we have replotted this figure as suggested to make it clearer.

Reviewer #2 (Remarks to the Author):

I previously reviewed this manuscript as Reviewer #2. Overall, I think the authors did a great job addressing my concerns with this manuscript. I have only a few small suggestions on phrasing.

We sincerely thank you for your recognition of our work, and for your constructive and detailed comments, which have led to a much-improved manuscript.

Abstract: Should be "warming on growth"

--Thank you, we have corrected the text.

Line 92: "Variation in tree growth is usually related to climatic variation" could be interpreted in a number of ways. There are lots of things that lead to variation in tree growth. I think here, you mean that studies often focus in on linking variation to climate though. Rephrase for clarity.

--Thank you, we have rephrased this sentence as following:

Line 84: "Many studies focus on the variation in tree growth related to climatic variations."

Line 192-196: these sentences are confusing to me. Please clarify.

--Thank you, we have rephrased this sentence as following:

Lines 176-180: "Populations with opposing temperature sensitivities were significantly clustered along local climatic conditions (two-sided t-test, $p < 0.01$). Interestingly, the latitudinal cluster prevailed over the longitudinal cluster (Supplementary Fig. 7), which reinforces the idea of local temperature in determining the growth-temperature response pattern."

Line 610. Still not happy with the use of the term "threatened" here. Please replace with something more like "where growth would decline with high temperatures"

--Thank you, we have corrected the text as suggested (on Line 539).